# VISUAL DESCRIPTION GROUNDING REDUCES HALLUCINATIONS AND BOOSTS REASONING IN LVLMS

**Sreyan Ghosh**[1*]**, Chandra Kiran Reddy Evuru**[1*]**, Sonal Kumar**[1*]**, Utkarsh Tyagi**[1]**,
Oriol Nieto**[2]**, Zeyu Jin**[2]**, Dinesh Manocha**[1]
[1]University of Maryland, College Park, USA, [2]Adobe, USA
{sreyang, sonalkum}@umd.edu *equal technical contribution.

Project: `https://sreyan88.github.io/VDGD/`

## ABSTRACT

Large Vision-Language Models (LVLMs) often produce responses that misalign with factual information, a phenomenon known as hallucinations. While hallucinations are well-studied, the exact causes behind them remain underexplored. In this paper, we first investigate the root causes of hallucinations in LVLMs. Our findings reveal that existing mitigation techniques primarily reduce hallucinations for visual recognition prompts—those that require simple descriptions of visual elements—but fail for cognitive prompts that demand deliberate reasoning. We identify the core issue as a lack of true visual perception in LVLMs: although they can accurately recognize visual elements, they struggle to fully interpret these elements in the context of the input prompt and effectively link this recognition to their internal knowledge, which is critical for reasoning. To address this gap, we introduce **Visual Description Grounded Decoding** (VDGD), a simple, robust, and *training-free* method designed to enhance visual perception and improve reasoning capabilities in LVLMs. VDGD works by first generating a detailed description of the image and appending it as a prefix to the instruction. During response generation, tokens are sampled based on their KL divergence to the description, favoring candidates with lower divergence. Experimental results on multiple visual reasoning benchmarks and LVLMs demonstrate that VDGD consistently outperforms existing baselines 2% - 33%. Finally, we introduce VaLLu, a benchmark designed for comprehensive evaluation of the cognitive capabilities of LVLMs.

## 1 INTRODUCTION

Large Language Models (LLMs) pre-trained on web-scale text data with the next token prediction objective implicitly compress world knowledge in their parameters (Zhao et al., 2023). These models learn general-purpose representations, which can then be *aligned* with the desired response characteristics (Zhang et al., 2024b). This step generally involves fine-tuning on instruction-response pairs, also known as *instruction tuning* (IT) (Wei et al., 2022), and is followed by an optional step of *reinforcement learning from human feedback* (RLHF) (Bai et al., 2022). Such aligned LLMs can be engaged through text-based inputs, or *prompts*, directing them to perform various tasks, including those based on information retrieval and reasoning.

Recent advancements in the research community have demonstrated success in multi-modal alignment, whereby fine-tuning LLMs with multi-modal instruction-response pairs enables them to execute tasks that conform to the text prompt as well as additional multi-modal inputs (Yin et al., 2023a). Since their introduction, the research community has extensively evaluated the capabilities of Large Vision-Language Models (LVLMs) across areas such as visual recognition (Wang et al., 2023a), perception (Fu et al., 2024), and reasoning (Yue et al., 2024). Researchers have made significant efforts to enhance these capabilities by refining model architectures and training techniques (Zhang et al., 2024a) or by implementing strategies to mitigate hallucinations [1] (Huang et al., 2023a).

Among these approaches, *training-free* hallucination mitigation techniques stand out as a cost-effective and accessible method to enhance LVLM performance without the need for extensive

---

[1]Hallucination refers to the mismatch between factual content and the model's generated responses. Mitigation aims to reduce these discrepancies for more accurate outputs and improve task performance.

computational resources. However, current hallucination mitigation techniques primarily focus on reducing hallucinations and improving the performance of visual recognition tasks (e.g., describing a scene, OCR, etc). Our extensive experiments show that these techniques fall short when applied to cognitive prompts requiring reasoning or knowledge extraction, indicating a significant gap in their ability to address hallucinations in more complex, reasoning-intensive scenarios.

To this end, we propose **Visual Description Grounded Decoding** (VDGD), a simple and novel *training-free* technique designed to improve LVLM reasoning by reducing hallucinations for cognitive prompts. Much like humans, who often write down their observations of an image in their own words and refer back to them while tackling complex reasoning tasks, we hypothesize that grounding the LVLM's response generation in an explicit visual description can significantly aid its reasoning capabilities. Specifically, we first generate a description of the input image. Next, at every decoding step during auto-regressive response generation, we calculate the Kullback–Leibler divergence (KLD) between the top-$k$ tokens in the current logit and all the logits of the image description. Finally, to predict the next word, we sample from the plausible candidates according to their KLD to the description, where lower KLD is given higher preference. By continuously referencing its descriptive summary of the visual content, the LVLM maintains alignment between the visual input and its generated responses, thereby reducing hallucinations. To summarize, our contributions are as follows:

1. Through extensive experiments, we categorize hallucinations in LVLMs and show that existing mitigation techniques work well for visual recognition prompts but fail for reasoning-intensive cognitive prompts.

2. Inspired by this, we identify a critical visual perception gap in LVLMs: while they can recognize visual elements and reason, they struggle to contextualize visual information with the prompt, leading to hallucinations in cognitive reasoning tasks.

3. We introduce **VDGD**, a novel and training-free method that enhances LVLM reasoning by grounding response generation in visual descriptions. We evaluate VDGD on 8 benchmarks with prompts that require deliberate reasoning and how VDGD significantly outperforms existing techniques, improving performance by 2% - 33% by reducing hallucinations.

4. Finally, we introduce **VaLLu**, a meticulously curated benchmark designed to evaluate the cognitive capabilities of LVLMs.

## 2 RELATED WORK

**Large Vision-Language Models** In recent years, Large Vision-Language Models (LVLMs) have seen rapid advancements with the release of numerous new models. These developments have been driven by innovations in model architectures (Liu et al., 2023b; Ye et al., 2023; Wang et al., 2023b), training methods (Liu et al., 2023b), alignment techniques (Chen et al., 2024b), and data augmentation techniques (Liu et al., 2023b; Wang et al., 2024b). To assess the capabilities of these models, the research community has introduced a variety of benchmarks specifically designed to evaluate different aspects, including visual recognition and understanding in diverse scenarios (Liu et al., 2023a; Kim et al., 2022), reasoning (Lu et al., 2023; Yue et al., 2024; Bai et al., 2024; Liu et al., 2024b), knowledge-based information retrieval (Hu et al., 2023), and other specialized tasks.

**Reducing Hallucinations to Improve Response Quality:** Hallucination mitigation broadly aims to enhance the accuracy of model responses. While numerous studies have proposed methods to reduce hallucinations and improve response accuracy (Sicong Leng et al., 2023; Huang et al., 2023b; Yin et al., 2023b; Zhou et al., 2023b) and evaluated their effectiveness (Wang et al., 2023a; Guan et al., 2023), most focus on mitigating object hallucinations in visual recognition tasks. While these efforts are valuable, they fall short of assessing the true reasoning capabilities of LVLMs. Moreover, despite the growing focus on evaluation and mitigation, the underlying causes of hallucinations remain largely unexplored. Recent methods like Compositional Chain-of-Thought Prompting (CCoT)(Mitra et al., 2024), Visual Table(Zhong et al., 2024), and Visual Evidence Prompting (Li et al., 2024) introduce structured intermediate representations (e.g., scene graphs or hierarchical descriptions) to improve reasoning. While these methods show strong performance on real-world scenes, they rely heavily on explicit object-centric representations and often struggle with abstract or non-real-world inputs (e.g., charts or mathematical graphs). Additionally, methods like Visual Table are not training-free and require substantial computational resources. In contrast, VDGD is lightweight, training-free, and generalizes effectively to diverse inputs, including abstract datasets like MMMU and MathVista.

## 3 HOW CLOSE ARE WE TO RELIABLE LVLM RESPONSES?

**Experimental Setup for Analysis.** We evaluate six LVLMs—LLaVA-v1, LLaVA-v1.5, LLaVA-v1.6, mPLUG-Owl2, InternLM-X, and CogVLM—all built on a 7B parameter language model. The models are tested across seven benchmarks: AMBER (visual recognition), SynthDoG (OCR), MMMU (expert-level reasoning), MathVista and MATH-Vision (mathematical reasoning), MMC (chart understanding), and MME and HallusionBench. To address potential limitations in existing evaluation metrics, which may not fully capture reasoning skills and often rely on string-matching techniques, we employ a combination of expert human evaluation and LLM-as-a-Judge (using GPT-4-turbo-2024-04-09). This paradigm has been extensively followed by prior studies in LLM evaluation (Zhou et al., 2023a; Ghosh et al., 2024; Zheng et al., 2023) and hallucination evaluation (Sicong Leng et al., 2023; Yu et al., 2023; Liu et al., 2023c). We devise an evaluation prompt that penalizes hallucinations and assigns scores from 1 to 5 based on factual correctness (see Appendix G). This approach also helps us standardize scoring across benchmarks and has a higher correlation ($\approx$0.97%) to expert human evaluations (see Section D). Quantitative results for all graphs in the paper are in Appendix L.

**Preliminaries.** As LVLMs advance and become more accessible, their application extends beyond merely generating image descriptions. Thus, to better understand visual perception and reasoning, we break down information processing by an LVLM into its constituent steps:

$$\text{Visual Recognition} \xrightarrow[\text{Perception}]{} \text{Knowledge Extraction (optional)} \rightarrow \text{Reasoning (optional)}$$

Figure 1 demonstrates that different prompts applied to a single image can assess various LVLM skills. Visual Recognition (VR) focuses on identifying visual elements and their relationships within the image, such as describing objects or specific details. Visual Perception (VP) extends beyond VR by interpreting and contextualizing these elements within the broader scene (Fu et al., 2024), essential for tasks requiring more than basic recognition. Prompts that demand knowledge-specific insights (also known as information-seeking prompts) engage in

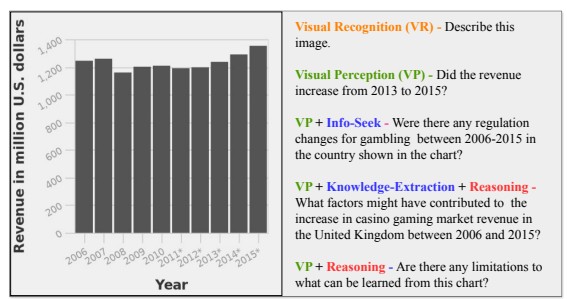

Figure 1: Depending on the text instruction, an LVLM might be assessed on one or more different capabilities.

knowledge extraction (KE) learned from pre-training. Finally, prompts requiring reasoning involve combining visual data, textual prompts, and extracted knowledge to generate a response. While recognition depends on the vision encoder, knowledge extraction, and reasoning rely on the foundational language model. VP, implicitly learned during alignment (Ghosh et al., 2024; Zhou et al., 2023a), bridges VR and higher cognitive functions, facilitating comprehensive understanding of the image in context to the prompt (Barrow et al., 1978; Torralba & Oliva, 2002; Hartley & Zisserman, 2003).

### 3.1 DO HALLUCINATION MITIGATION TECHNIQUES IMPROVE COGNITIVE ABILITIES?

Fig. 2 (left) compares the performance of LVLMs across 5 benchmarks. As we clearly see, while newer models that were trained using improved architectures and alignment training data boosted performance on the AMBER benchmark (for recognition), performance on other reasoning and information-seeking benchmarks remained stagnant.

Fig. 2 (right) compares the performance of LLaVA-1.5 across the same 5 benchmarks when employed with explicit hallucination mitigation techniques, with strategies ranging from improving model

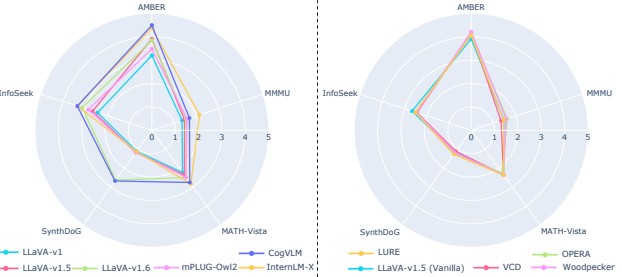

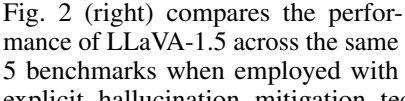

Figure 2: (Left) Performance comparison of different LVLMs on various benchmarks. (Right) Performance comparison of different hallucination mitigation techniques applied to LLaVA-1.5.

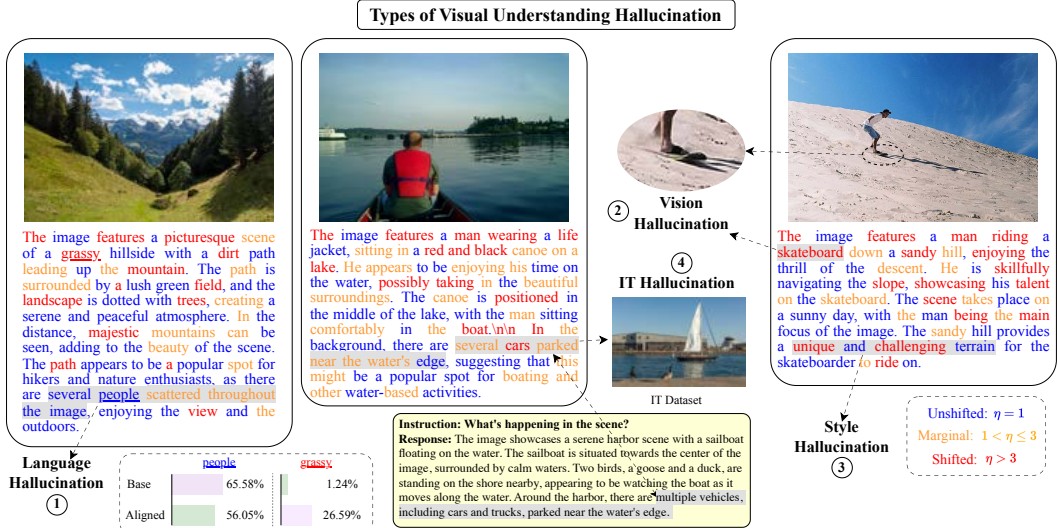

Figure 4: **Types of Visual Recognition Hallucinations.** We define Algo. 1 to divide VR hallucinations into 4 different categories automatically: Language, Vision, Style, and IT (explained further in Sec. 3.3). While language and vision hallucinations have been explored earlier, and methods to alleviate them have been proposed, we show for the first time that Style and IT hallucinations exist and existing methods fail to alleviate them.

architectures to explicit response correction strategies, namely, Visual Contrastive Decoding (VCD) (Sicong Leng et al., 2023), OPERA (Huang et al., 2023b), Woodpecker (Yin et al., 2023b) and LURE (Zhou et al., 2023b). As we clearly see, while all these methods boost performance on AMBER, performance on other reasoning and information-seeking benchmarks remains stagnant.

## 3.2 DO MITIGATION TECHNIQUES GENERALIZE BEYOND REAL-WORLD SCENES?

As LVLMs are now being employed on a wide range of tasks, considerable research efforts are being made to evaluate the efficacy of LVLMs to reason on images beyond real-world scenes, i.e., charts (Liu et al., 2023a) or math problems (Lu et al., 2024). *However, how often do LVLMs hallucinate while describing images beyond real-world scenes?* Fig 3 (left) compares the performance of various LVLMs across 5 benchmarks. MMMU and MathVista include images with mathematical figures and MMC includes charts from diverse sources. On the other hand, Synth-

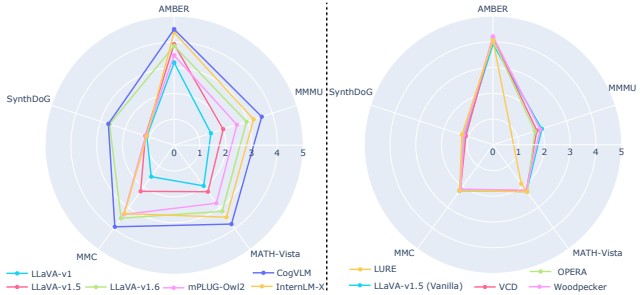

Figure 3: (Left) Performance comparison of different LVLMs on various benchmarks when prompted to only describe the image. (Right) Performance comparison of different hallucination mitigation techniques applied to LLaVA-1.5.

DoG prompts the model for OCR and AMBER for image descriptions for real-world scenes. As we clearly see, compared to AMBER, models hallucinate more often when describing visuals beyond real-world scenes. Fig 3 (right) further shows that hallucination mitigation techniques improve performance on AMBER but rarely other benchmarks.

## 3.3 ARE ALL HALLUCINATIONS EQUAL?

**Overview.** In our detailed analysis of model responses on the AMBER dataset, we identified intriguing patterns in which hallucinations occur. Although the community has primarily focused on the broader topic of *object hallucinations* (Liu et al., 2024b), our findings reveal that visual recognition hallucinations can be classified into several subcategories based on their specific occurrence patterns. Specifically, we perform the token-distribution analysis proposed by (Lin et al., 2023).

For a given instruction-response-image triplet, the instruction $i = \{i_1, i_2, \cdots\}$ and the image $v$ are first input to the aligned (or fine-tuned) model to obtain its response $r = \{r_1, r_2, \cdots\}$ via greedy decoding. Next, for each position $t$ in the response, a 'context' at this position is defined as to be $x_t = i + \{r_1, \cdots, r_{t-1}\}$. This "context" is then input to the base model (model from which the aligned model was instruction-tuned) to obtain its probability distribution for predicting the next token at position $t$, $P_{base}$. We then define **Base Rank** as follows: Rank in $P_{base}$ of the token at $t$ with the maximum $P_{align}$ value. With the base rank denoted as $\eta$, the unshifted, marginal and shifted tokens are defined as when $(\eta = 0)$, $(0 < \eta \leq 2)$ and $(\eta > 2)$ respectively. Next, we use our judge (GPT-4) to extract the exact hallucinated phrases based on 3 types: object, relation, and action hallucinations. Finally, based on the Base Rank formulation and the hallucinated phrases, we propose Algorithm 1 to categorize identified hallucinations into 4 different categories based on their cause. The algorithm is further verbally described in Appendix K.4.

---

**Algorithm 1** Categorizing Visual Hallucinations

Given: Model Response $R$, IT dataset $D_{IT}$, Aligned LVLM $M_{aligned}$, base LLM $M_{base}$,
Obtain Top-$k$ similar instances $S_R$ to $R$ from $D_{IT}$ using CLIP
Extract visual elements $V$ from $R$.
Extract hallucinated phrases $P$ from $R$ using LLM judge divided into Object, Relation, and Action hallucinations
**procedure** TOKENANALYSIS($R$, $M_{aligned}$, $M_{base}$)
   **for** each word $p$ in $P$ **do**
      **if** $p$ in $V$ **then**      ▷ Consider only 1st token of $p$
         Calculate Base-Rank **BR** of $p$    ▷ Defined in Section 3.3
         **if BR** $> 0$ **then**    ▷ marginal or shifted token
            **if** $p$ in $S_R$ **then**
               $p$ is an IT Hallucination
            **else**
               **if** $p ==$ Relation & $p\; != $ Object **then**
                  $p$ is a Style Hallucination
               **else**
                  $p$ is a Vision Hallucination
               **end if**
            **end if**
         **else**        ▷ unshifted token
            $p$ is a Language Hallucination
         **end if**
      **end if**
   **end for**
**end procedure**

---

**(1) Language Hallucinations**: These are hallucinations that are caused when LVLMs over-rely on the language priors (or prior tokens in the response) rather than the input image. We attribute all hallucinated tokens that are unshifted as language hallucinations.

**(2) Vision Hallucinations**: Unlike language hallucinations, the tokens for vision hallucinations are generated by attending to the input image and are shifted or marginal tokens. These are hallucinations caused when the model fails to accurately recognize visual elements in an input image or inaccurate reasoning or knowledge extraction, among other factors.

**(3) 1.3 Hallucinations.** These are hallucinations caused by *style imitation*. Open-source VLLMs are often trained on instruction-tuning data synthesized from proprietary closed-source models. Training on such datasets makes the VLLM learn to mimic the characteristics of responses in the IT dataset (Ghosh et al., 2024; Gudibande et al., 2024) (e.g., lengthy answers, a summary line at the end of the response, etc.). However, owing to the lack of sufficient knowledge to mimic its closed-source counterparts, open-source VLLMs might hallucinate facts while responding (Ghosh et al., 2024).

**(4) Instruction Tuning (IT) Hallucinations.** These are hallucinations caused by biases learned during the instruction tuning stage. LLMs are known to mimic the responses as-sociated with the unfamiliar examples in the model's instruction-tuning data (i.e., those unfamiliar to the pre-trained base model) (Ghosh et al., 2024). Formally, this can be defined as a distribution shift. As visual instruction tuning is an extreme case of such a distribution shift, IT hallucinations are a major cause of hallucinations in LLMs.

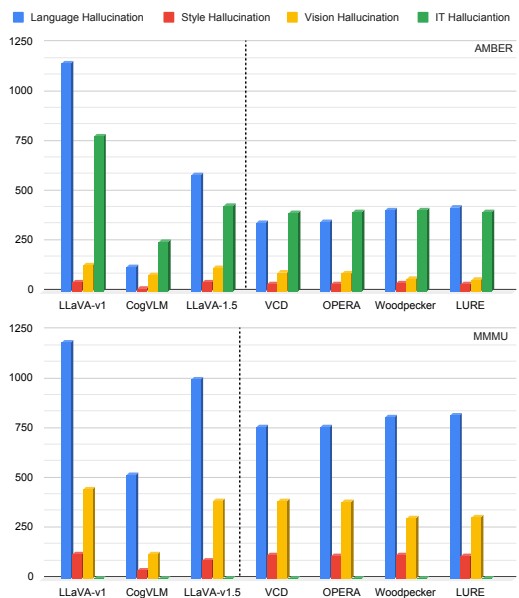

Figure 5: (Left) Frequency comparison of hallucination categories. (Right) Comparison for LLaVA-v1.5 with hallucination mitigation techniques. The top graph compares AMBER, and the bottom graph compares MMMU.

**Performance Comparison.** Fig. 5 (top;left) compares the frequency of different categories of hallucinations across various LVLMs for AMBER. Fig. 5 (top; right) displays the frequency of different hallucination categories for LLaVA-v1.5 alongside various hallucination mitigation techniques.

Fig. 5 (bottom) offers a similar comparison for MMMU. Our key findings include: **(1)** Model-based improvements and mitigation strategies reduce language and vision hallucinations, but progress in style and IT hallucinations remains limited. **(2)** Specific mitigation techniques are more effective for certain types of hallucinations. For e.g., decoding-based methods like VCD and OPERA excel at reducing language hallucinations, whereas grounding approaches like Woodpecker are more effective for vision hallucinations. Decoding-based techniques benefit from low-confidence hallucinated tokens and uncertainty encoded in logits discussed next. More discussion is in Appendix K.1. **(3)** Existing methods fail to reduce hallucinations on reasoning benchmarks like MMMU. We attribute this to the algorithmic biases of these methods that are crafted to just reduce object hallucinations.

> ***Key Takeaway:*** Current hallucination mitigation techniques mainly improve LVLMs' visual recognition skills but are less effective in improving other cognitive abilities, such as reasoning. Additionally, these improvements are largely limited to real-world scenes and specific types of hallucinations, leaving areas like non-real-world scenes largely unexplored.

## 4  THE VISUAL PERCEPTION GAP: LVLMS CAN SEE BUT NOT PERCEIVE

It is evident that current hallucination mitigation techniques primarily enhance visual recognition but fail to improve other cognitive abilities, such as reasoning or knowledge extraction. In this section, we investigate the underlying cause of this limitation. Our analysis reveals that LVLMs often rely on language priors rather than attending to the input image when generating responses to reasoning prompts. This leads us to identify a critical issue: the visual perception gap. While LVLMs can accurately recognize visual elements and possess the necessary knowledge and reasoning skills to respond factually, they struggle to perceive and interpret these elements in relation to the input prompt. This gap in perception causes hallucinations, resulting in incorrect responses.

### 4.1  VISUAL BLIND SPOTS: LVLMS IGNORE THE INPUT IMAGE FOR REASONING

With findings from previous sections, we now assess the influence of the image on the LVLM response. Precisely, we compare the Base Rank (defined in Section 3.3) of tokens across datasets. A lower value would signify that the model solely responds using language priors, while a higher value would signify that the model pays attention to the image, i.e., it responds with tokens different from what the model would have responded with only language priors. Fig. 6 plots the Base Rank of tokens against the token position in response (average across the dataset). As we see, the Base Rank for AMBER, which prompts for an image description, is higher than Math-Vision, which requires it to reason or extract knowledge based on the image. This hints to-

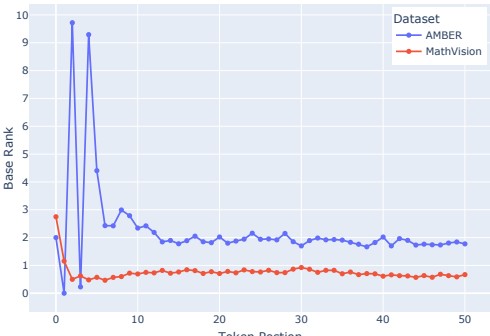

Figure 6: Base Rank Comparison between AMBER and MATH-Vision datasets as a function of token position in responses (for CogVLM).

wards an alignment issue: different from AMBER, for Math-Vision LVLMs are unable to perceive the image before responding, under-relying on visual cues and over-relying on language priors for generating responses. In Appendix K.1 we explain why decoding-based mitigation techniques discussed in Section 3.3 do not work for reaosning in cognitive prompts.

### 4.2  DISENTANGLING VISUAL RECOGNITION FROM OTHER COGNITIVE SKILLS

A key objective of alignment fine-tuning is to equip LVLMs with implicit visual perception skills. As outlined in Section 3, LVLMs should interpret images in context and respond accurately to information-seeking or reasoning prompts. To determine if hallucinations arise from the base LLM's limitations or alignment gaps, we aim to isolate visual recognition from other cognitive skills and evaluate them separately. We use GPT-4 to rephrase prompts from visual information-seeking and reasoning benchmarks, modifying them so they can be answered without the image (examples in Appendix K.7 and prompt used in Appendix G).

LVLMs are first evaluated on these text-only prompts and then on generating independent image descriptions (evaluation prompt in Appendix G). Figure 7 (left) compares the performance of LVLMs on the text-only prompts (marked with -*t*) against their original image-text versions, while Figure 7 (right) shows VR scores. Our results indicate that LVLMs excel in visual recognition and perform better on factual responses when using text-only prompts than with combined image-text inputs. This suggests that

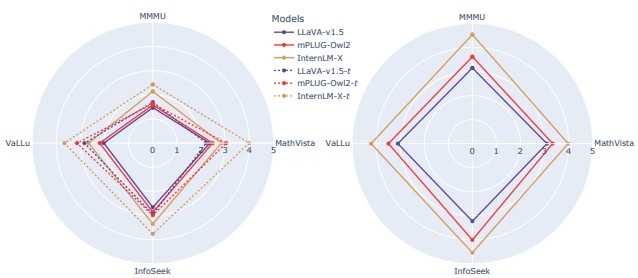

Figure 7: (Left) Performance comparison of different LVLMs when prompted w/ original prompt vs rephrased prompts w/o image (-*t*). (Right) Performance comparison of different LVLMs for their ability to generate a faithful image description.

LVLMs have the necessary knowledge and reasoning skills but struggle to integrate these with visual perception in prompts requiring both. We identify this as a perception gap: LVLMs can recognize visual elements but fail to fully perceive them in a way that connects recognition to internal knowledge, which is essential for reasoning or information extraction. This challenge likely arises due to several factors: (i) training datasets that are largely composed of image-description pairs, (ii) alignment datasets that contain strong visual cues within the natural-language prompts.

## 4.3 Uncertainty in the Logit Space

To investigate the origins of hallucinations, we analyze the logit space of hallucinated tokens. Specifically, after applying the softmax operation to the logits, we rank them by their probability scores. We then use the elbow method (detailed in Appendix K.6) to truncate the distribution, obtaining the top-*k* tokens along with their corresponding probabilities. This truncation approach, based on the elbow method, effectively identifies the tokens most likely to be selected during multinomial sampling. Table 1

| Models | Dataset | $k$ | Variance | Range |
|---|---|---|---|---|
| LLaVA-v1 | AMBER | 1.1 / 4.2 | 0.0 / 0.1 | 0.0 / 0.3 |
| LLaVA-v1.5 | AMBER | 1.0 / 3.9 | 0.0 / 0.1 | 0.0 / 0.4 |
| CogVLM | AMBER | 1.0 / 3.4 | 0.0 / 0.1 | 0.0 / 0.5 |
| LLaVA-v1 | MMMU | 1.2 / 3.7 | 0.0 / 0.1 | 0.0 / 0.4 |
| LLaVA-v1.5 | MMMU | 1.2 / 3.5 | 0.0 / 0.2 | 0.0 / 0.4 |
| CogVLM | MMMU | 1.1 / 3.2 | 0.0 / 0.2 | 0.0 / 0.5 |

Table 1: Post-truncation probability statistics using the elbow method. All values are in the format: non-/hallucinated averaged across all tokens in the dataset.

shows several statistics of the top-*k* probabilities for hallucinated tokens (we only consider the first token of each word). As evident from the value of *k*, standard deviation, and the average, the top-*k* probability space for hallucinated tokens is dominated by a set of low and equally confident tokens. Thus, each of these tokens has almost a similar chance of being sampled with multi-nomial sampling.

> ***Key Takeaway:*** LVLMs generally demonstrate accurate recognition and cognition skills to respond and reason accurately. However, they struggle to effectively link they recognize to their internal knowledge during the reasoning process, which results in inaccurate responses. This issue underlines what we identify as the visual perception gap.

## 5 Visual Description Grounding Decoding (VDGD)

**Main Motivation.** To bridge the visual perception gap and reduce hallucinations for cognitive prompts, we propose Visual Description Grounding Decoding (VDGD), a simple, effective, and *training-free* hallucination mitigation technique. During decoding, VDGD first appends the image description (generated by the LVLM itself) as a prefix to the prompt's text instruction, then grounds token generation by selecting the token that *deviates* the least from the description. Token deviation is measured using a novel formulation based on Kullback–Leibler

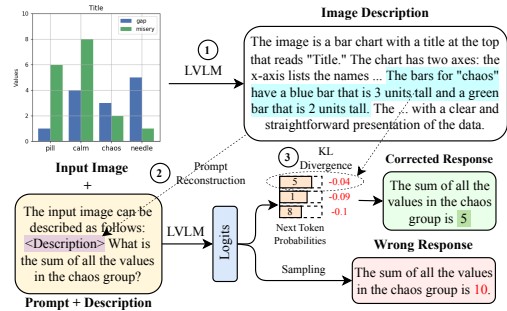

Figure 8: Illustration of our proposed VDGD method.

divergence of the token with the description generated by the LVLM itself. Much like humans, who often write down their key observations in an image and refer to them while solving complex reasoning tasks, we hypothesize that grounding the LVLM's responses in a detailed visual description can significantly enhance its reasoning process. Additionally, re-scoring the equally and low-confidence tokens (discussed in Section 1) based on deviation from the description can increase the confidence of the accurate token from several low-confidence tokens (Xu et al., 2023).

## 5.1 METHODOLOGY

Fig. 8 illustrates the VDGD methodology. For an input prompt, we first employ the LVLM to generate a description of the image accompanying the prompt (see Appendix G for the prompt). Next, we concatenate the generated description as a prefix to the original prompt.

Formally put, let the VLM, denoted as $p_{VLM}$, receive an input prompt of length $n$ symbolized as $X_{\text{pre}} = x_1 \ldots x_n$, with each $x_i$ representing a token from the vocabulary $\mathcal{V}$. The decoder's objective is to produce responses of length $m$, represented as $X_{\text{res}} = x_{n+1}, \ldots, x_{n+m}$. During the decoding phase, the model iteratively decodes one token at a time, conditioning on the tokens that have been generated so far:

$$p_{VLM}(X_{\text{resp}} \mid X_{\text{pre}}) = \prod_{i=n+1}^{n+m} p_{VLM}(x_i \mid x_{<i}) \tag{1}$$

Here, $p_{VLM}(x_i \mid x_{<i})$ denotes the probability distribution for the next token, $x_i$, given the previous tokens. Inspired by our findings in Section 4.3, we first apply a plausibility constraint to truncate the vocabulary space of the next token $x_i$ (Li et al., 2023). Essentially, this step cuts off tokens from the vocabulary $\mathcal{V}$ below the elbow and keeps only the top-$k$ plausible tokens. This is done as follows:

$$p_{\text{VLM}}(x_i \mid x_{<i}) \begin{cases} p_{\text{VLM}}(x_i \mid x_{<i}), & \text{if } p_{\text{VLM}}(x_i \mid x_{<i}) \geq \alpha \max_w p_{\text{VLM}}(w \mid x_{<i}) \\ -\infty, & \text{Otherwise} \end{cases} \tag{2}$$

where $\alpha$ is a hyper-parameter between [0,1]. Now let $\mathcal{V}_K^{x_i}$ be the set of top-$K$ most plausible tokens from the vocabulary $\mathcal{V}$ after the truncation of $x_i$. Next, for each token $w_k \in \mathcal{V}_K^{x_i}$ in the top-$K$ plausible tokens, we calculate the deviation of the token with the $n$ tokens of the prompt as follows:

$$\text{KL}_{w_k}^{x_i} = \min_{1 \leq j \leq n} \text{KL}\left(\text{one-hot}(w_k) \| p_{\text{VLM}}(\cdot \mid x_{<j})\right). \tag{3}$$

where KL is the Kullback–Leibler divergence and $\text{one-hot}(w_i) \in \mathrm{R}^{\mathcal{V}}$ is the one-hot representation of token $w_k$ where each value is 0 except the position corresponding to the token-id of $w$, which is 1. Next, we replace the value of $w_k$ in the logit with $-\text{KL}_{w_k}^{x_i}$ and apply softmax on the logits to obtain a probability distribution. Thus, tokens with larger KLD (or a large *deviation*) to the input prompt are less likely to be sampled given the softmax operation upon KL divergence. With this distribution, we can now apply any sampling technique to sample the next token.

## 5.2 EXPERIMENTAL SETUP

**Datasets and Evaluation Metrics.** For evaluation, we employ a variety of standard benchmarks focused on reasoning and information-seeking tasks. These include LLaVA-Bench, MM-Vet (Yu et al., 2023), MMBench (Liu et al., 2023d), MME (Fu et al., 2023), MathVista (test-mini subset), MathVision, and MMMU (validation set). For each benchmark, we employ our proposed GPT evaluation technique described in Section 3. Beyond being more robust to the diverse formats of model generation, we also find GPT scores to have a higher correlation to expert-human evaluation ($\approx 0.97$; Section D has more details with original benchmark evaluation metrics that has a correlation of $\approx 0.92$). We are inspired by a wealth of prior studies in hallucination evaluation (Sicong Leng et al., 2023; Yu et al., 2023; Liu et al., 2023c). All results are averaged across 3 runs.

**Baselines.** We compare VDGD against VCD, OPERA, Woodpecker, LRV, LURE, HALC (Chen et al., 2024a) and PAI (Liu et al., 2024d) hallucination mitigation techniques. Additionally, we compare two vanilla decoding methods (without any additional mitigation techniques): Vanilla-greedy with vanilla-greedy decoding and Vanilla-sampling with multinomial sampling-based decoding (top-p=0.5 and temperature=0.7). We employ greedy decoding for all methods as we find no difference in performance on sampling. For LVLMs, we employ LLaVA-v1, LLaVA-v1.5, LLaVA-v1.6, mPLUG-Owl2, InternLM-X, CogVLM, Qwen2-VL (Wang et al., 2024b) and CogVLM2 (Hong et al., 2024).

| Benchmark | Baseline | LLaVA-v1 | LLaVA-1.5 | LLaVA-1.6 | mPLUG-Owl2 | InternLM-X | CogVLM | CogVLM2 | Qwen2-VL |
|---|---|---|---|---|---|---|---|---|---|
| **MMMU** | Vanilla-greedy | 1.26 | 1.35 | 1.42 | 1.40 | 2.01 | 1.66 | 2.08 | 2.39 |
| | Vanilla-sampling | 1.27 | 1.44 | 1.40 | 1.41 | 2.05 | 1.64 | 2.10 | 2.35 |
| | VCD | 1.34 | 1.52 | 1.44 | 1.53 | 2.22 | 1.68 | 2.14 | 2.64 |
| | OPERA | 1.30 | 1.43 | 1.57 | 1.64 | 2.25 | 1.62 | 2.21 | 2.44 |
| | Woodpecker | 1.32 | 1.44 | 1.63 | 1.61 | 2.12 | 1.65 | 2.14 | 2.56 |
| | LRV | 1.29 | 1.49 | 1.61 | 1.58 | 2.08 | 1.59 | 2.18 | 2.48 |
| | LURE | 1.31 | 1.47 | 1.60 | 1.64 | 2.27 | 1.66 | 2.27 | 2.71 |
| | PAI | 1.42 | 1.39 | 1.44 | 1.56 | 2.23 | 1.65 | 2.29 | 2.64 |
| | HALC | 1.40 | 1.54 | 1.63 | 1.65 | 2.15 | 1.67 | 2.24 | 2.69 |
| | **VDGD (ours)** | **1.49 (+18%)** | **1.62 (+20%)** | **1.75 (+23%)** | **1.72 (+23%)** | **2.39 (+19%)** | **1.71 (+3%)** | **2.47 (+19%)** | **2.91 (+22%)** |
| **MathVista** | Vanilla-greedy | 1.56 | 1.65 | 2.00 | 1.92 | 2.56 | 2.15 | 2.45 | 2.97 |
| | Vanilla-sampling | 1.54 | 1.68 | 1.91 | 1.94 | 2.52 | 2.19 | 2.46 | 2.95 |
| | VCD | 1.67 | 1.68 | 2.07 | 2.11 | 2.59 | 2.53 | 2.81 | 3.19 |
| | OPERA | 1.64 | 1.83 | 2.04 | 2.04 | 2.62 | 2.30 | 2.53 | 3.22 |
| | Woodpecker | 1.72 | 2.10 | 2.19 | 2.13 | 2.43 | 2.43 | 2.59 | 3.13 |
| | LRV | 1.68 | 1.68 | 2.17 | 1.99 | 2.61 | 2.20 | 2.62 | 3.30 |
| | LURE | 1.59 | 2.03 | 2.21 | 2.07 | 2.37 | 2.45 | 2.49 | 3.13 |
| | PAI | 1.68 | 1.85 | 1.97 | 2.08 | 2.63 | 2.20 | 2.60 | 3.39 |
| | HALC | 1.71 | 2.05 | 2.17 | 1.97 | 2.54 | 2.24 | 2.77 | 3.38 |
| | **VDGD (ours)** | **1.84 (+18%)** | **2.19 (+33%)** | **2.44 (+22%)** | **2.24 (+17%)** | **2.88 (+13%)** | **2.62 (+22%)** | **2.93 (+20%)** | **3.59 (+21%)** |
| **MMBench** | Vanilla-greedy | 2.67 | 2.89 | 3.11 | 2.98 | 3.58 | 3.19 | 3.71 | 3.93 |
| | Vanilla-sampling | 2.69 | 2.92 | 3.12 | 3.02 | 3.59 | 3.22 | 3.68 | 3.95 |
| | VCD | 2.80 | 3.03 | 3.09 | 3.22 | 3.93 | 3.28 | 3.81 | 4.20 |
| | OPERA | 2.81 | 2.94 | 3.17 | 3.10 | 3.95 | 3.22 | 3.93 | 4.05 |
| | Woodpecker | 2.83 | 2.96 | 3.14 | 3.07 | 3.84 | 3.33 | 3.83 | 4.19 |
| | LRV | 2.77 | 2.99 | 3.13 | 3.23 | 3.84 | 3.25 | 3.77 | 4.05 |
| | LURE | 2.72 | 2.97 | 3.13 | 3.26 | 3.72 | 3.21 | 3.82 | 4.16 |
| | PAI | 2.75 | 3.00 | 3.04 | 3.16 | 3.94 | 3.35 | 3.97 | 4.05 |
| | HALC | 2.73 | 2.93 | 3.02 | 3.18 | 3.81 | 3.27 | 4.04 | 4.08 |
| | **VDGD (ours)** | **2.93 (+10%)** | **3.12 (+8%)** | **3.26 (+5%)** | **3.32 (+11%)** | **4.08 (+14%)** | **3.42 (+7%)** | **4.15 (+12%)** | **4.31 (+10%)** |
| **MME** | Vanilla-greedy | 3.32 | 3.54 | 3.65 | 3.49 | 3.75 | 3.57 | 3.64 | 3.86 |
| | Vanilla-sampling | 3.34 | 3.53 | 3.62 | 3.48 | 3.77 | 3.58 | 3.66 | 3.85 |
| | VCD | 3.46 | 3.61 | 3.77 | 3.59 | 3.96 | 3.67 | 3.95 | 4.21 |
| | OPERA | 3.42 | 3.59 | 3.82 | 3.54 | 3.93 | 3.60 | 3.88 | 4.09 |
| | Woodpecker | 3.37 | 3.55 | 3.76 | 3.63 | 3.82 | 3.59 | 3.92 | 4.15 |
| | LRV | 3.43 | 3.63 | 3.78 | 3.52 | 3.95 | 3.61 | 3.96 | 3.99 |
| | LURE | 3.42 | 3.62 | 3.87 | 3.67 | 3.93 | 3.72 | 3.77 | 4.10 |
| | PAI | 3.38 | 3.56 | 3.81 | 3.55 | 3.89 | 3.69 | 3.82 | 4.04 |
| | HALC | 3.47 | 3.58 | 3.83 | 3.71 | 4.01 | 3.59 | 3.80 | 4.14 |
| | **VDGD (ours)** | **3.59 (+8%)** | **3.70 (+5%)** | **3.99 (+9%)** | **3.82 (+9%)** | **4.12 (+10%)** | **3.79 (+6%)** | **4.09 (+12%)** | **4.34 (+12%)** |
| **Oven** | Vanilla-greedy | 2.44 | 2.66 | 3.13 | 2.86 | 3.35 | 3.35 | 3.48 | 3.64 |
| | Vanilla-sampling | 2.45 | 2.65 | 3.10 | 2.84 | 3.33 | 3.37 | 3.45 | 3.61 |
| | VCD | 2.20 | 2.44 | 2.96 | 2.67 | 3.10 | 3.21 | 3.39 | 3.59 |
| | OPERA | 2.50 | 2.49 | 3.20 | 2.92 | 3.44 | 3.42 | 3.53 | 3.68 |
| | Woodpecker | 2.46 | 2.46 | 3.21 | 2.90 | 3.45 | 3.45 | 3.52 | 3.70 |
| | LRV | 2.51 | 2.52 | 3.18 | 2.88 | 3.47 | 3.40 | 3.57 | 3.74 |
| | LURE | 2.49 | 2.51 | 3.18 | 2.91 | 3.42 | 3.46 | 3.55 | 3.72 |
| | PAI | 2.54 | 2.60 | 3.26 | 2.95 | 3.48 | 3.48 | 3.60 | 3.78 |
| | HALC | 2.52 | 2.57 | 3.24 | 2.94 | 3.50 | 3.45 | 3.61 | 3.80 |
| | **VDGD (ours)** | **2.78 (+14%)** | **2.84 (+7%)** | **3.49 (+12%)** | **3.15 (+10%)** | **3.68 (+10%)** | **3.59 (+7%)** | **3.75 (+8%)** | **3.92 (+7%)** |
| **LLaVA-Bench** | Vanilla-greedy | 3.01 | 3.12 | 3.23 | 3.13 | 4.12 | 3.68 | 4.15 | 4.37 |
| | Vanilla-sampling | 2.98 | 3.10 | 3.27 | 3.17 | 4.08 | 3.71 | 4.17 | 4.33 |
| | VCD | 2.93 | 3.16 | 3.18 | 3.15 | 3.89 | 3.57 | 4.13 | 4.28 |
| | OPERA | 3.08 | 3.18 | 3.30 | 3.22 | 4.16 | 3.75 | 4.22 | 4.40 |
| | Woodpecker | 3.10 | 3.22 | 3.36 | 3.23 | 4.21 | 3.73 | 4.21 | 4.39 |
| | LRV | 3.16 | 3.19 | 3.30 | 3.22 | 4.27 | 3.72 | 4.20 | 4.49 |
| | LURE | 3.18 | 3.26 | 3.35 | 3.24 | 4.25 | 3.64 | 4.27 | 4.51 |
| | PAI | 3.20 | 3.28 | 3.38 | 3.25 | 4.30 | 3.74 | 4.32 | 4.54 |
| | HALC | 3.25 | 3.27 | 3.37 | 3.22 | 4.24 | 3.76 | 4.35 | 4.53 |
| | **VDGD (ours)** | **3.36 (+12%)** | **3.47 (+11%)** | **3.52 (+9%)** | **3.38 (+8%)** | **4.45 (+8%)** | **3.96 (+8%)** | **4.50 (+8%)** | **4.78 (+9%)** |
| **MMVET** | Vanilla-greedy | 2.39 | 2.52 | 2.72 | 2.87 | 3.39 | 3.18 | 3.47 | 3.64 |
| | Vanilla-sampling | 2.34 | 2.50 | 2.70 | 2.84 | 3.32 | 3.35 | 3.42 | 3.62 |
| | VCD | 2.29 | 2.61 | 2.67 | 2.92 | 3.29 | 3.42 | 3.39 | 3.58 |
| | OPERA | 2.48 | 2.67 | 2.77 | 3.01 | 3.37 | 3.39 | 3.52 | 3.69 |
| | Woodpecker | 2.47 | 2.64 | 2.82 | 3.06 | 3.40 | 3.44 | 3.50 | 3.72 |
| | LRV | 2.58 | 2.56 | 2.79 | 2.92 | 3.38 | 3.42 | 3.51 | 3.75 |
| | LURE | 2.54 | 2.63 | 2.81 | 3.12 | 3.41 | 3.46 | 3.56 | 3.71 |
| | PAI | 2.62 | 2.60 | 2.83 | 3.18 | 3.37 | 3.49 | 3.59 | 3.78 |
| | HALC | 2.66 | 2.70 | 2.88 | 3.15 | 3.44 | 3.47 | 3.54 | 3.76 |
| | **VDGD (ours)** | **2.85 (+19%)** | **3.01 (+19%)** | **3.23 (+19%)** | **3.45 (+20%)** | **3.78 (+12%)** | **3.57 (+12%)** | **3.74 (+8%)** | **3.99 (+10%)** |
| **VaLLu** | Vanilla-greedy | 1.95 | 2.03 | 2.63 | 2.20 | 2.66 | 2.82 | 3.23 | 3.45 |
| | Vanilla-sampling | 1.86 | 2.01 | 1.64 | 2.18 | 2.70 | 2.83 | 3.21 | 3.40 |
| | VCD | 1.47 | 1.55 | 1.80 | 1.80 | 2.32 | 2.63 | 3.19 | 3.42 |
| | OPERA | 2.04 | 2.05 | 2.62 | 2.26 | 2.65 | 2.85 | 3.22 | 3.47 |
| | Woodpecker | 2.01 | 2.05 | 2.67 | 2.23 | 2.60 | 2.91 | 3.28 | 3.52 |
| | LRV | 1.98 | 2.10 | 2.65 | 2.19 | 2.59 | 2.88 | 3.32 | 3.51 |
| | LURE | 2.03 | 2.03 | 2.64 | 2.24 | 2.64 | 2.78 | 3.28 | 3.48 |
| | PAI | 2.04 | 2.23 | 2.78 | 2.34 | 2.76 | 2.86 | 3.31 | 3.54 |
| | HALC | 2.02 | 2.22 | 2.74 | 2.32 | 2.72 | 2.89 | 3.29 | 3.52 |
| | **VDGD (ours)** | **2.16 (+11%)** | **2.64 (+30%)** | **3.16 (+20%)** | **2.72 (+24%)** | **3.45 (+30%)** | **3.01 (+7%)** | **3.48 (+8%)** | **3.67 (+6%)** |

Table 2: Performance comparison of VDGD with various baselines. VDGD outperforms by 2%-33%.

**Proposed VaLLu Benchmark.** Automatic evaluation of LVLMs, with powerful closed-source models, on multiple large benchmarks can be prohibitively expensive. Thus, we propose VaLLu, a benchmark that we build by amalgamating existing benchmarks.

VaLLu comprises a total of 1500 instances, with 554 sourced from Oven, 535 from MMMU, 218 from MMC, 74 from MathVista, 60 from HallusionBench, 53 from MATH-Vision, and 14 from MME. We only include instances with instructions that require open-ended generations and do not include any binary Yes/No or Multi-choice questions. We control the size to ensure the evaluation is affordable while keeping the diversity of tasks and topics for comprehensive analysis. VaLLu is manually filtered for noisy examples commonly found in existing benchmarks (we

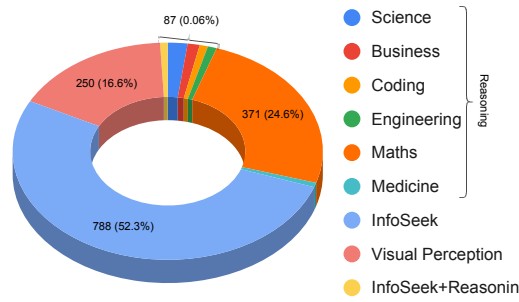

Figure 9: Distribution of task types in VaLLu.

discuss this further in Appendix K.3), annotated with additional meta-data (illustrated in Fig. 9) and paired with an expert-provided response.

## 5.3 Results and Analysis

**Quantitative results.** Table 2 compares the performance of VDGD with other baselines. As we clearly see, VDGD not only outperforms all baselines but is also the only method that alleviates and improves performance on VaLLu. Results on other full benchmarks are provided in Appendix L.6, together with scores for other aspects beyond factuality, where we show that VDGD does not compromise the depth and expressivity of responses. Ablation results in Appendix C demonstrate that (i) removing KL divergence leads to drop in performance (ii) using only the description prefix without the image leads to performance drops confirming that VDGD enhances reasoning through grounding rather than making the model rely completely on the description alone, and (iii) more advanced captioning models can significantly enhance VDGD performance, while smaller captioning models provide competitive results with the added benefit of lower computational overhead.

**Qualitative examples.** Fig. 10 compares responses from vanilla greedy decoding with VDGD. Grounding response generation results in more factual responses. Appendix B illustrates 2 failure cases of VDGD where incorrect descriptions can lead VDGD to hallucinate.

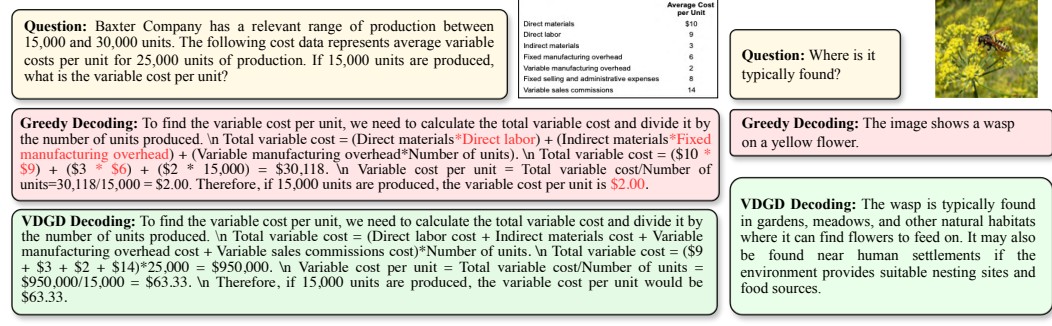

Figure 10: Illustration of responses generated with VDGD compared with greedy decoding. More in Sec. K.2.

## 6 Conclusion, Limitations and Future Work

In this paper, we conduct extensive experiments to analyze hallucinations in LVLMs. Our findings reveal that although there has been considerable progress in reducing hallucinations related to visual recognition, reducing hallucinations for cognitive prompts has been limited. Further investigation into LVLM responses to cognitive prompts reveals a visual perception gap: despite having the required capabilities, LVLMs often produce hallucinations when responding to prompts. To address this issue, we introduce VDGD, a novel and training-free method to reduce hallucinations. Our results demonstrate that VDGD is the first method to effectively reduce hallucinations in responses that necessitate additional cognitive skills beyond just visual recognition.

As part of future work, we would like to work on (1) Error accumulation in VDGD that comes from inaccurate image descriptions in the prefix and (2) the requirement of prompting the LVLM twice.

## 7 REPRODUCIBILITY STATEMENT

We provide our code here: `https://sreyan88.github.io/VDGD/`. All experimental details, including training parameters and hyper-parameters, are provided in Section 5.2. The Appendix has more comprehensive details about our experimental setup.

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

## A APPENDIX

In the supplemental material,

1. B: We illustrate failure cases for VDGD.
2. C: We provide results for ablation of VDGDs key components.
3. D: We provide correlation between GPT scores and actual metrics.
4. E: We provide a detailed explanation of computational complexity of VDGD algorithm described in section 5.
5. F: We describe the impact of the paper on society and research communities.
6. G: We provide the prompts used for object detection, description, rephrasing, noise removal and evaluations required for our experiments.
7. H: We provide a detailed description of all the datasets utilized in this paper.
8. I: We provide a detailed description of all the models we have experimented with.
9. J: We provide a detailed description of all the prior-art hallucination mitigation techniques used as baselines.
10. K: We provide details about:
    - K.1: We provide detailed explanation about Logit Space for VR and Cognitive Prompts.
    - K.2: We provide examples comparing generations from Greedy, Decoding, VCD and VDGD.
    - K.4: We provide and in-depth explanation of all the details of Algorithm 1.
    - K.6: We explain the elbow method used for logit analysis.
    - K.7: We provide examples of rephrased prompts which do not require image input.
    - K.8: We provide details of computational resources used for this paper.
11. L: We provide additional results for all the experiments described in the paper.
12. M: We provide details about computation cost for VDGD.
13. N: We provide results for VDGD with beam search.
14. O: We provide some further disccusion about related works.

## B VDGD FAILURE CASES

Fig. 11 illustrates 2 failure cases of VDGD. We show that VDGD can lead to incorrect responses due to an incorrectly generated image description. However, we also argue that as LVLMs get better at describing images, the performance of VDGD decoding will improve.

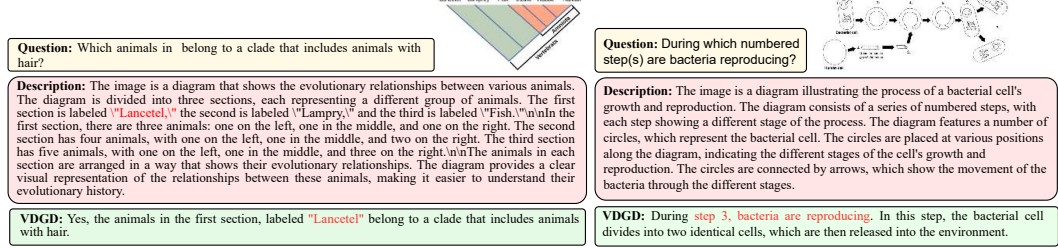

Figure 11: Illustration of failure cases of VDGD.

## C VDGD ABLATION RESULTS

To demonstrate the effectiveness of VDGD, we conduct ablation studies on its key components: (i) VDGD (- VCD): Image descriptions are generated without using Visual Contrastive Decoding (VCD), (ii) VDGD (+ GPT-4 Captions): Image descriptions are generated using GPT-4-turbo-2024-04-09 instead of the LVLM itself. (iii) VDGD (+ Ramos et al. Captions): Image descriptions are generated

using the small captioning model introduced by  Ramos et al.  instead of the LVLM itself. (iv) VDGD (- KLD): KLD-based sampling is removed, and only the description is appended, (v) VDGD (- Image): VDGD is applied without the original image input; only the image description is appended to the prompt. Results are presented on the VDGD benchmark in Table 3. Removing KL Divergence leads to a substantial performance drop. We notice a similar phenomenon by removing the input image with the steepest drop. This shows that the input image is still essential for the model.

As we can clearly see, VDGD achieves a significant performance boost when using captions generated by GPT-4, highlighting the potential of higher-quality captions to enhance VDGD's effectiveness. This is a case of implicit knowledge transfer from GPT-4 to VDGD through detailed descriptions. Additionally, VDGD delivers competitive results with a small-scale captioning model, illustrating an alternative, more computationally efficient approach to improving performance. These findings suggest that future advancements in both large and small captioning models could further enhance the performance and efficiency of VDGD.

| Benchmark | LLaVA-v1 | LLaVA-1.5 | LLaVA-1.6 | mPLUG-Owl2 | InternLM-X | CogVLM |
|---|---|---|---|---|---|---|
| Vanilla-greedy | 1.95 | 2.03 | 2.63 | 2.20 | 2.66 | 2.82 |
| **VDGD** *(ours)* | 2.16 | 2.64 | 3.16 | 2.72 | 3.45 | 3.01 |
| **VDGD (+) GPT-4 Captions** | $2.31_{\pm 0.04}$ | $2.91_{\pm 0.6}$ | $3.37_{\pm 0.02}$ | $2.97_{\pm 0.05}$ | $3.65_{\pm 0.02}$ | $3.44_{\pm 0.06}$ |
| **VDGD (+) Ramos et al. Captions** | $2.06_{\pm 0.06}$ | $2.38_{\pm 0.08}$ | $3.00_{\pm 0.03}$ | $2.43_{\pm 0.09}$ | $3.23_{\pm 0.08}$ | $2.95_{\pm 0.04}$ |
| **VDGD (-) VCD** | $2.08_{\pm 0.09}$ | $2.43_{\pm 0.15}$ | $3.01_{\pm 0.08}$ | $2.54_{\pm 0.07}$ | $3.26_{\pm 0.12}$ | $2.95_{\pm 0.06}$ |
| **VDGD (-) KLD** | $2.05_{\pm 0.10}$ | $2.30_{\pm 0.10}$ | $2.78_{\pm 0.12}$ | $2.37_{\pm 0.06}$ | $3.01_{\pm 0.02}$ | $2.87_{\pm 0.08}$ |
| **VDGD (-) Image** | $1.69_{\pm 0.07}$ | $1.95_{\pm 0.02}$ | $2.43_{\pm 0.06}$ | $1.98_{\pm 0.02}$ | $2.24_{\pm 0.04}$ | $2.50_{\pm 0.05}$ |

Table 3: Ablation results for VDGD. VDGD sees a drop in performance when the KLD decoding objective is removed and a major drop when no input image is provided.

## D    CORRELATION BETWEEN GPT SCORES AND ACTUAL METRICS

**Background of Experts in Human Evaluation.** For the expert human evaluation of model generations, we engaged a group of 5 PhD students with at least four years of research experience in computer vision. These experts were tasked with evaluating the outputs of the models for accuracy, relevance, and factual correctness. To ensure thorough evaluation, they were provided with internet access to verify factual information when necessary. The evaluation process was conducted in an anonymized format to maintain objectivity, with the experts working independently and without prior knowledge of the model specifics. We selected 500 samples or the size of the benchmark, whichever was lower. Scores presented in Table 4 are averaged across scores provided by the 5 students.

Table 4 presents scores in the format of original benchmark metrics / GPT Scores / Expert Human Scores. Remarkably, we show that our proposed GPT score has a higher correlation to Expert Human Score. Nonetheless, our GPT Score also has a high correlation with the original benchmark metrics.

| Dataset | Baseline | LLaVA-1.5 | mPLUG-Owl2 | InternLM-X | CogVLM2 | Avg Correlation |
|---|---|---|---|---|---|---|
| MME | Vanilla | 1501.45 / 3.54 / 3.91 | 1450.19 / 3.49 / 3.82 | 1712.00 / 3.75 / 4.15 | 1512.45 / 3.66 / 4.02 | 0.95 / 0.96 / 0.92 |
| | Woodpecker | 1503.84/3.55/3.94 | 1508.87/3.63/3.98 | 1746.93 / 3.82 / 4.21 | 1789.63 / 3.92 / 4.33 | |
| | VDGD | 1698.23 / 3.70 / 4.12 | 1724.27 / 3.82 /4.24 | 1852.48 / 4.12 / 4.56 | 1848.87 / 4.09 / 4.34 | |
| MMMU-Val | Vanilla | 30.30 / 1.35 / 2.02 | 32.72 / 1.40 / 2.14 | 43.35 / 2.01 / 2.70 | 44.37 / 2.08 / 2.83 | 0.99 / 0.96 / 0.96 |
| | Woodpecker | 33.12 / 1.44 / 1.83 | 36.35 / 1.61 / 1.79 | 47.58 / 2.12 / 2.60 | 48.98 / 2.14 / 2.53 | |
| | VDGD | 36.89 / 1.62 / 2.37 | 38.42 / 1.72 / 2.55 | 53.86 / 2.39 / 2.80 | 56.43 / 2.47 / 3.16 | |
| LLaVA-Bench | Vanilla | 65.42 / 3.10 / 3.49 | 69.37 / 3.17 / 3.60 | 72.21 / 4.08 / 4.22 | 73.76 / 4.17 / 4.19 | 0.96 / 0.97 / 0.95 |
| | Woodpecker | 70.48 / 3.22 / 3.52 | 70.68 / 3.23 / 3.74 | 74.87 / 4.21 / 4.48 | 74.52 / 4.21 / 4.17 | |
| | VDGD | 71.03 / 3.47 / 4.04 | 70.57 / 3.38 / 3.91 | 77.43 / 4.45 / 4.72 | 78.03 / 4.50 / 4.73 | |
| MMVET | Vanilla | 31.15 / 2.50 /2.41 | 36.3 / 2.84 / 2.99 | 51.27 / 3.32 / 3.70 | 60.42 / 3.42 / 3.55 | |
| | Woodpecker | 33.42 / 2.64 / 2.88 | 47.26 / 3.06 / 3.29 | 51.67 / 3.40 / 3.81 | 65.78 / 3.50 / 3.75 | |
| | VDGD | 45.79 / 3.01 / 3.27 | 62.41 / 3.45 / 3.90 | 74.57 / 3.78 / 4.16 | 71.98 / 3.74 / 4.10 | 0.97 / 0.98 / 0.91 |

Table 4: Scores across 4 benchmarks in Original Metric / (Our proposed) GPT Score / (Our) Expert Human Score format. We also show the person correlation averaged across datasets. The correlation scores are in Original Metric-GPT Score / GPT Score-Expert Human Score / Original Metric-Expert Human Score format. GPT Scores show a higher correlation with Expert Human Scores.

## E    VDGD COMPUTATIONAL COMPLEXITY

Given an LVLM with a vocabulary size $V$, an input instruction $I$ with a description of length $M$ tokens, at each decoding step, the complexity for equation (2) is $O(V)$ and let the number of plausible

tokens after this step be $K$, the complexity for equation (3) is $O(VMK)$. The total computational complexity of VDGD at each decoding step is $O(V) + O(VMK) \approx O(VMK)$.

## F  BROADER IMPACT

This paper addresses the broader impact of reducing hallucinations in Large Vision-Language Models (LVLMs), which are increasingly utilized in diverse applications such as automated content generation, assistive technologies, and decision-making systems. By focusing on the phenomenon of hallucination, particularly in responses to cognitive prompts, our work has the potential to enhance the reliability and safety of LVLMs in practical scenarios. The proposed Visual Description Grounded Decoding (VDGD) method, being training-free, offers an accessible and scalable solution that can be readily implemented across various models to mitigate risks associated with inaccurate or misleading outputs. Furthermore, the introduction of the VaLLu benchmark contributes to the field by providing a tool for the comprehensive evaluation of cognitive capabilities in LVLMs, promoting transparency and fostering further research in the area. Collectively, these contributions aim to improve user trust in AI systems and encourage responsible AI development that prioritizes factual accuracy and understanding.

## G  PROMPTS

For image description benchmarks, we employ the prompt illustrated in Fig. 12. For non-image description benchmarks, we modify this prompt slightly (we modify the part of the prompt for extracting hallucinated phrases), and we illustrate the same in Fig. 16. The prompt used to generate image descriptions for VDGD is illustrated in Fig. 15. *Correctness* in the prompts refers to *Factuality* in all tables in Section L. We name it so after an ablation study on the terminology that works better for prompting the judge (GPT-4). All prompt evaluations on GPT-4 are done with a temperature of 0.7.

```
# Evaluation Prompt

Please act as an impartial judge and evaluate the quality of the response provided with respect to the input image. You will rate the quality
of the response on multiple aspects, such as Helpfulness, Clarity, Correctness, Depth and Engagement. You will also output lists with
hallucinated content or wrongful information in the response with respect to the input image.

##Query: {instruction}

##Output: {answer}

## Evaluate
### Aspects
- Helpfulness: Rate the response based on how well it addresses the users query about the image and provides a relevant answer. A score of 5
indicates the answer fully aids the user, while a 1 suggests it offers little to no help.
- Clarity: Rate the response based on how well-structured it is, with ideas presented in a clear and coherent manner. A high score of 5 means
the answer is clear and logically structured, while a 1 suggests a disjointed or confusing reply.
 - Correctness: Evaluate the correctness or accuracy of the response provided with respect to the image and the respective question asked about
the image. A perfect 5 indicates the response is entirely correct and accurate, while a 1 suggests it has significant errors or has not
provided an answer to the question asked at all.
 - Depth: Determine the level of detail and thoroughness in the response. A score of 5 means the answer delves deeply into the aspects of the
input image for answering the question, while a 1 indicates it barely scratches the surface.

### Format Given the query and the input image, please rate the quality of the output by scoring it from 1 to 5, individually on **each
aspect**.
- 1: strongly disagree
- 2: disagree
- 3: neutral
- 4: agree
- 5: strongly agree

 You are also asked to output the hallucinated content or wrongful information in the response with respect to the input image with respect to
3 different aspects:
### Aspects
- Object Hallucination: Objects in the response that are not present in the image.
- Action/Verb Hallucination: Actions or verbs in the response that cannot be perceived from the image. For e.g., if the person is just walking
and the response contains that the person is jumping.
- Relation Hallucination: Object-to-object spatial relationships in the response that are not present in the image or cannot be seen or
perceived from the image. For example, if a book is on the left of a glass but the response contains it is on the right.

Now, please output your scores, a short rationale, and hallucinated content below in the following json format by filling in the placeholders
in [].
  { 'helpfulness': { 'reason': '[your rationale]', 'score': '[score from 1 to 5]' }, 'clarity': { 'reason': '[your rationale]', 'score':
'[score from 1 to 5]' }, 'correctness': { 'reason': '[your rationale]', 'score': '[score from 1 to 5]' }, 'depth': { 'reason': '[your
rationale]', 'score': '[score from 1 to 5]' }, 'engagement': { 'reason': '[your rationale]', 'score': '[score from 1 to 5]' }, 'object
hallucinations': {'reason': '[elaborate reasoning of hallucination]', 'tokens': '[comma separated list of exact phrases from the response that
are hallucinated]'}, 'action/verb hallucinations': {'reason': '[elaborate reasoning of hallucination]', 'tokens': '[comma separated list of
exact phrases from the response that are hallucinated]'}, 'relation hallucinations': {'reason': '[elaborate reasoning of hallucination]',
'tokens': '[comma separated list of exact phrases from the response that are hallucinated]'} }

Only return the json and nothing else.
```

Figure 12: Prompt used to evaluate image description benchmarks using GPT-4-Turbo as described in Section 3.2.

```
# Rephrase Prompt

I will provide you with an image and an instruction that I aim to ask an AI agent to obtain a response from it. Referring to the input image,
rephrase the instruction such that the agent is asked to refer more to the image for solving the instruction to output a faithful response
(currently it is looking more at the instruction). Instruction: {inst}

Please output a json with a single key called 'rephrased': and the output as the rephrased instruction.
```

Figure 13: Prompt used to evaluate the performance of LLMs on rephrased instructions using GPT-4-Turbo as described in Section 4.2.

```
# LLaMA3 Prompt

I will provide you with a response from an AI agent which has been asked to describe an image. Please identify all the phrases that in the
image description that constitute the image. These phrases might be foreground and background objects, adverbial phrases, etc. Return them as
comma separated values. There should not be any additional information other than these values in the output. The response is as follows:
{response}
```

Figure 14: Prompt used to identify the visual elements in images using Llama3 for the algorithm described in Section 3.3.

```
# Image Description Prompt

<image>
I have been given this image to complete the task described as: {inst}.

To help me complete the task, describe the given image in detail. In case of real-world scenes, please include all foreground and background
objects in the description, their properties (like color, shape, etc.), their relations with other objects, their count, and all other
components in the image. In case of non-real-world scenes, like charts, graphs, tables, etc., please describe the table, mention all numbers
(if any), mention the written text, and all other details.
```

Figure 15: Prompt used to describe images using LVLMs for VDGD as described in Section 5.

```
# Maths Evaluation Prompt

Please act as an impartial judge and evaluate the quality of the response provided with respect to the input image. You will rate the quality
of the response on multiple aspects, such as Helpfulness, Clarity, Correctness and Depth. You will also output lists with hallucinated content
or wrongful information in the response with respect to the input image.

##Query: {instruction}

##Output: {answer}

## Evaluate
### Aspects
- Helpfulness: Rate the response based on how well it addresses the user query about the image and provides a relevant answer. A score of 5
indicates the answer fully aids the user, while a 1 suggests it offers little to no help.
- Clarity: Rate the response based on how well-structured it is, with ideas presented in a clear and coherent manner. A high score of 5 means
the answer is clear and logically structured, while a 1 suggests a disjointed or confusing reply.
 - Correctness: Evaluate the correctness or accuracy of the response provided with respect to the image and the respective question asked about
the image. A perfect 5 indicates the response is entirely correct and accurate, while a 1 suggests it has significant errors or has not
provided an answer to the question asked at all.
 - Depth: Determine the level of detail and thoroughness in the response. A score of 5 means the answer delves deeply into the aspects of the
input image for answering the question, while a 1 indicates it barely scratches the surface.

### Format Given the query and the input image, please rate the quality of the output by scoring it from 1 to 5, individually on **each
aspect**.
- 1: strongly disagree
- 2: disagree
- 3: neutral
- 4: agree
- 5: strongly agree

 You are also asked to output all the hallucinated content or wrongful information in the response with respect to the input image.

Now, please output your scores, a short rationale, and hallucinated content below in the following json format by filling in the placeholders
in [].
  { 'helpfulness': { 'reason': '[your rationale]', 'score': '[score from 1 to 5]' }, 'clarity': { 'reason': '[your rationale]', 'score':
'[score from 1 to 5]' }, 'correctness': { 'reason': '[your rationale]', 'score': '[score from 1 to 5]' }, 'depth': { 'reason': '[your
rationale]', 'score': '[score from 1 to 5]' }, 'engagement': { 'reason': '[your rationale]', 'score': '[score from 1 to 5]' },
'hallucinated_phrases': {'reason': '[elaborate reasoning of different hallucinations]', 'tokens': '[comma separated list of exact phrases from
the response that are hallucinated]'},

Only return the json and nothing else.
```

Figure 16: Prompt used to evaluate generations from cognitive benchmarks in Table 2.

# H  DATASET DETAILS

**VaLLu** We propose VaLLu benchmark which is sourced from Oven, MMMU, MMC, MathVista, HallusionBench, MATH-Vision and MME. This dataset is licensed under all the licenses of the original benchmarks that it was sourced from.

**AMBER** AMBER (Wang et al., 2023a) is a benchmark for evaluating hallucinations in both the generative task and discriminative task of MLLMs. It provides comprehensive coverage of evaluations for various types of hallucination, including existence, attribute, and relation. Licensed under Apache-2.0 License.

```
# Evaluation Prompt
     ↳   Please act as an impartial judge and evaluate the quality of an image description provided by an AI agent with
respect to an input image and an input query about the image. You will rate the quality of the response on a single
aspect, which is Informativeness. Precisely, I will provide you with an input image, a cognitive query about the input
image, and a description of the input image. The description is what an AI agent visualizes or recognizes in the input
image. You need to judge how informative the description is with respect to responding correctly to the input cognitive
query.

##Query: {image}

##Output: {prediction}

##Evaluate
### Aspects
- Informativeness: Evaluate the informativeness of the image description. A perfect 5 indicates the information is
entirely correct and accurate, while a 1 suggests it has significant errors.

### Format Given the query, please rate the quality of the output by scoring it from 1 to 5 , individually on **each
aspect**.
- 1: strongly disagree
- 2: disagree
- 3: neutral
- 4: agree
- 5: strongly agree

Now, please output your scores and a short rationale below in a json format by filling in the placeholders in []: {
'informativeness': { 'reason': '[your rationale]', 'score': '[score from 1 to 5]' }}"
```

Figure 17: GPT evaluation prompt for evaluating informativeness of image descriptions generated by LVLMs.

```
# Text-only Evaluation Prompt

Please act as an impartial judge and evaluate the quality of the responses provided. You will rate the quality of the output on multiple
aspects, such as Helpfulness, Clarity, Correctness, Depth, and Engagement.

##Query: {instruction}

##Output: {answer}

## Evaluate
### Aspects
- Helpfulness: Rate the response based on how well it addresses the users query and provides a relevant solution. A score of 5 indicates the
answer fully aids the user, while a 1 suggests it offers little to no help.
- Clarity: Rate the response based on how well-structured it is, with ideas presented in a clear and coherent manner. A high score of 5 means
the answer is clear and logically structured, while a 1 suggests a disjointed or confusing reply.
 - Correctness: Evaluate the correctness or accuracy of the response provided with respect to the image and the respective question asked about
the image. A perfect 5 indicates the response is entirely correct and accurate, while a 1 suggests it has significant errors or has not
provided an answer to the question asked at all.
 - Depth: Determine the level of detail and thoroughness in the response. A score of 5 means the answer delves deeply into the topic, while a 1
indicates it barely scratches the surface.
 - Engagement: Assess how engaging and natural the response sounds in a conversational context. A high score of 5 reflects a response that
feels engaging and human-like in its tone, while a 1 indicates a robotic or boring reply. ,

### Format Given the query, please rate the quality of the output by scoring it from 1 to 5 , individually on **each aspect**.
- 1: strongly disagree
- 2: disagree
- 3: neutral
- 4: agree
- 5: strongly agree

Now, please output your scores and a short rationale below in a json format by filling in the placeholders in []: { 'helpfulness': { 'reason':
'[your rationale]', 'score': '[score from 1 to 5]' }, 'clarity': { 'reason': '[your rationale]', 'score': '[score from 1 to 5]' },
'correctness': { 'reason': '[your rationale]', 'score': '[score from 1 to 5]' }, 'depth': { 'reason': '[your rationale]', 'score': '[score from
1 to 5]' }, 'engagement': { 'reason': '[your rationale]', 'score': '[score from 1 to 5]' } }
```

Figure 18: Prompt used to evaluate generations for rephrased instructions on benchmarks like VaLLu, MMMU, MathVista, and SynthDoG using GPT-4-Turbo as described in Section 4.2.

**MMMU** MMMU (Yue et al., 2024) benchmark evaluates multimodal models on multi-discipline tasks and demands college-level subject knowledge and deliberate reasoning. It consists of 11.5k questions, spanning 30 subjects and 183 subfields, focusing on advanced perception and reasoning on domain-specific knowledge. Licensed under Apache-2.0 License.

**MMC** MMC (Liu et al., 2023a) is a human-annotated benchmark comprising 9 distinct tasks to evaluate LVLM's reasoning capabilities to comprehend visual charts. It comprises tasks such as chart information extraction, chart reasoning, contextual chart understanding, etc. It also offers two evaluation methods: GPT4 free-format general ability evaluation and multiple-choice QA format chart understanding. Licensed for research purposes.

**MathVista** MathVista (Lu et al., 2023) is a robust mathematical reasoning evaluation benchmark that consists of challenging tasks that require fine-grained, deep visual recognition and compositional reasoning. It consists of 6141 examples derived from 31 multimodal datasets involving mathematics. Licensed under CC-BY-SA-4.0.

**MATH-Vision** MATH-Vision (Wang et al., 2024a) dataset is a collection of 3,040 high-quality mathematical problems with visual contexts sourced from real math competitions. The dataset spans 16 distinct mathematical disciplines and is graded across 5 levels of difficulty to evaluate the mathematical reasoning abilities of Multimodal Large Language Models(MLLM). Licensed under MIT License.

**HallusionBench** HallusionBench (Guan et al., 2023) is a comprehensive benchmark designed for the evaluation of image-context reasoning. The benchmark presents significant challenges to advanced large visual-language models (LVLMs), such as GPT-4V(ision), Gemini Pro Vision, Claude 3, and LLaVA- 1.5, by emphasizing nuanced understanding and interpretation of visual data. The benchmark comprises 346 images paired with 1129 questions, all meticulously crafted by human experts. Licensed under BSD-3-Clause.

**LlavaBench** LlavaBench (Liu et al., 2023c) is a challenging evaluation suite designed to assess visual-language alignment and instruction-following capabilities of large visual-language models. It consists of two benchmarks: LLaVA-Bench (COCO), which includes 30 randomly selected images from COCO-Val-2014 paired with 90 questions covering conversational, descriptive, and complex reasoning tasks, and LLaVA-Bench (In-the-Wild), featuring 24 diverse images from various domains, including memes, paintings, and sketches, with 60 questions.

**MM-Vet** MM-Vet (Yu et al., 2023) is an evaluation benchmark designed to assess large multimodal models on complex multimodal tasks. It defines six core vision-language capabilities—Recognition, Knowledge, OCR, Spatial Awareness, Language Generation, and Math—and examines their integration across 16 emergent tasks. MM-Vet employs an LLM-based evaluator for open-ended outputs, providing unified scoring across diverse question types and answer styles. The benchmark offers insights into the capabilities of different large multimodal model paradigms and models beyond simple performance ranking.

**MME** Multimodal large language model Evaluation Benchmark (MME) (Fu et al., 2023) measures perception and cognition abilities on a total of 2 tasks(Perception and Congition) and 14 subtasks(Existence, Count, Position, Color, Poster, Celebrity, Scene, Landmark, Artwork, OCR, Commonsense Reasoning, Numerical Calculation, Text Translation and Code Reasoning). To prevent data leakage from use of public datasets for evaluation, the annotations of instruction-answer pairs are all manually designed.

**Oven** Open-domain Visual Entity Recognition (OVEN) (Hu et al., 2023) is a task where a model needs to link an image onto a Wikipedia entity with respect to a text query. OVEN challenges models to select among six million possible Wikipedia entities, making it a general visual recognition benchmark with the largest number of labels. Licensed under MIT License.

**SynthDoG** SynthDoG (Kim et al., 2022) is created by generating synthetic data which consists of various components: background, document, text, and layout. The background image is sampled from ImageNet, and the document is sampled from the collected paper photos. The text is sampled from Wikipedia. Layout is constructed using randomly stacks grids. The english variant of the dataset consists of 65.5k training and 500 validation entries. Licensed under MIT License.

## I  MODEL DETAILS

**LLaVa-v1.** LLaVa-v1 (Liu et al., 2023c)[2] is an open-source LVLM trained by fine-tuning LLaMA/Vicuna (13B) on GPT-generated multimodal instruction-following data. It is initially trained on 558K filtered image-text pairs from LAION/CC/SBU and then, finetuned on 80K GPT-generated multimodal instruction-following data. Licensed under CC BY-SA 4.0 DEED.

**LLaVa-1.5.** LLaVa-1.5 (Liu et al., 2023b)[3] builds on LLaVA-v1's architecture by replacing linear projection design with a two-layer MLP. It also includes additional academic-task-oriented VQA datasets for VQA, OCR, and region-level perception, to enhance the model's multimodal capabilities. Licensed under CC BY-SA 4.0 DEED.

**LLaVa-1.6.** LLaVa-1.6 (Liu et al., 2024c)[4] re-uses the pretrained connector of LLaVA-1.5 and is instruction tuned on 158K GPT-generated multimodal data, 500K academic-task-oriented VQA data, 50K GPT-4V data and 40K ShareGPT data. LLaVa-1.6 has improved reasoning, OCR, and world knowledge. Licensed under CC BY-SA 4.0 DEED.

---

[2]https://huggingface.co/liuhaotian/llava-llama-2-7b-chat-lightning-lora-preview
[3]https://huggingface.co/liuhaotian/llava-v1.5-7b
[4]https://huggingface.co/liuhaotian/llava-v1.6-vicuna-7b

**mPLUG-Owl2.** mPLUG-Owl2 (Ye et al., 2023)[5] is the first Multi-modal Large Language Model that performs well in both pure-text and multi-modal scenarios. mPLUG-Owl2 is pretrained 400 million image-text pairs from: Conceptual Captions (CC3M/CC12M), COCO, Laio-en, COYO, DataComp and for instruction tuning 5 types of data are used: image captioning, mage question answering, region-aware QA, multi-modal instruct data and text-only instruct. Licensed under MIT License.

**InternLM-X.** InternLM-X (Zhang et al., 2023)[6] is a vision-language large model based on InternLM2 (Cai et al., 2024) for advanced text-image comprehension and composition. InternLM-X is pre-trained on 1.1 billion images alongside 77.7 billion text tokens, including both public datasets and in-house concept data followed by supervised fine-tuning using multi-task training and instruction tuning. Licensed under Apache-2.0 License.

**CogVLM.** CogVLM (Wang et al., 2023b)[7] is a powerful open-source visual language model. CogVLM-17B has 10 billion vision parameters and 7 billion language parameters. CogVLM is pre-trained using 1.5B text-image pairs and instruction tuned on various visual question-answering datasets. CogVLM achieves state-of-the-art performance on 10 classic cross-modal benchmarks. Licensed under Apache-2.0 License.

**Llama-2-7b.** Llama-2-7b (Touvron et al., 2023)[8] is an open source fine-tuned generative text model with 7 billion parameters. It is an auto-regressive language model that uses an optimized transformer architecture. The fine-tuned version use supervised fine-tuning (SFT) and reinforcement learning with human feedback (RLHF) for helpfulness and safety. Licensed under Llama 2 community license agreement.

**Vicuna-7b-v1.5.** Vicuna-7b-v1.5 (Zheng et al., 2023)[9] is a 7b parameter open-sourced model and fine-tuned from Llama 2 with supervised instruction fine-tuning and is an auto-regressive language model based on the transformer architecture. The training data is around 125K conversations from ShareGPT. Licensed under Apache-2.0 License.

**InternLM2-7B.** InternLM-7B (Cai et al., 2024)[10] InternLM2 is a open-source 7 billion parameter chat model with a 200k context window and achieves good performance in the"Needle-in-a-Haystack" test(on long-context tasks like LongBench and L-Eval) and is trained using Conditional Online RLHF (COOL RLHF). Licensed under Apache-2.0 License.

**CogVLM2.** CogVLM2 (Hong et al., 2024)[11] CogVLM2 is a new generation of visual language models based on Meta-LLaMA-3-8B-Instruct, offering significant improvements over previous versions. It supports 8K content length, image resolutions up to 1344x1344, and excels in both English and Chinese. CogVLM2 has achieved state-of-the-art performance on benchmarks like TextVQA, DocVQA, and MM-Vet. The open-source models, including CogVLM2 and CogVLM2-Video, are available on GitHub, contributing to advancements in image and video understanding. Licensed under Apache-2.0 License.

**Qwen2-VL.** Qwen2-VL (Wang et al., 2024b)[12] Qwen2-VL is the latest iteration of the Qwen-VL model, featuring state-of-the-art performance in image and video understanding tasks. It excels in processing images of various resolutions and aspect ratios and supports long-form video understanding for tasks like question answering and content creation. Key architectural updates include Naive Dynamic Resolution and Multimodal Rotary Position Embedding (M-ROPE), improving its multimodal processing capabilities. Available in 2, 7, and 72 billion parameter variants. Licensed under Apache-2.0 License.

---

[5]https://huggingface.co/MAGAer13/mplug-owl2-llama2-7b

[6]https://huggingface.co/internlm/internlm-xcomposer2-vl-7b

[7]https://huggingface.co/THUDM/cogvlm-chat-hf

[8]https://huggingface.co/meta-llama/Llama-2-7b-chat-hf

[9]https://huggingface.co/lmsys/vicuna-7b-v1.5

[10]https://huggingface.co/internlm/internlm2-chat-7b

[11]https://huggingface.co/THUDM/cogvlm2-llama3-chat-19B

[12]https://huggingface.co/Qwen/Qwen2-VL-7B-Instruct

## J    BASELINE DETAILS

**VCD.** Visual Contrastive Decoding (VCD) (Sicong Leng et al., 2023) is a training-free method that contrasts output distributions of original/unaltered inputs and distorted visual inputs. VCD hypothesizes statistical bias and unimodal priors as the two essential causes of object hallucinations and reduces the dependence on them. This results in the generated content being closely grounded to visual inputs resulting more accurate outputs. Licensed under Apache-2.0 License.

**OPERA.** Over-trust Penalty and a Retrospection-Allocation (OPERA) (Huang et al., 2023b) is a training-free approach that hypothesizes that hallucinations are closely tied to the knowledge aggregation patterns in the self-attention matrix and that generation of new tokens by models is majorly dependent on a few summary tokens(over-trust issue) but not all of the previous tokens. This methodology introduces a penalty term on the model logits during beam-search decoding to mitigate the over-trust issue and implements a rollback strategy to retrospect the presence of summary tokens. Licensed under MIT License.

**Woodpecker.** Woodpecker (Yin et al., 2023b) is a training-free method that identifies and corrects hallucinations in the generated text. Unlike other training-free methodologies, Woodpecker works after the model generation process. This method works by creating a visual knowledge base of Question-Answer(QA) pairs which contain object-level and attribute-level claims about the input image followed by an LLM correcting hallucinations.

**LRV.** LRV (Liu et al., 2024a), or Large-scale Robust Visual Instruction tuning, proposes a dataset of visual instructions generated by GPT4, covering 16 vision-and-language tasks with open-ended instructions and answers. Each instance has positive instruction samples and negative instructions for more robust visual instruction tuning. As a result, a model trained on this dataset proves to be robust to hallucinations. Licensed under BSD-3-Clause license.

**LURE.** LVLM Hallucination Revisor (LURE) (Zhou et al., 2023b) attributes object hallucination to three factors: co-occurrence, uncertainty, and object position, and develops an object hallucination revisor which converts potential hallucinations into accurate outputs. This method first creates a hallucination dataset by making modifications to the original data and trains the object hallucination revisor on this dataset. This method post-hoc rectifies object hallucinations in LVLMs by reconstructing less hallucinatory descriptions.

**PAI.** Pay Attention to Image (PAI) (Liu et al., 2024d) is a training-free algorithm designed to reduce hallucinations in LVLMs. PAI intervenes during the inference process to make it more image-centric by amplifying attention to image tokens in the self-attention layers of LVLMs. By adjusting attention weights and subtracting logits of text-only inputs from multi-modal inputs, PAI mitigates the issue of "text inertia," where outputs are overly influenced by context text. This method enhances image representation during generation, leading to more accurate, visually grounded outputs without requiring additional training or external tools.

**HALC.** HALC (Chen et al., 2024a) is a novel decoding algorithm designed to mitigate object hallucinations (OH) in LVLMs. It integrates a robust auto-focal grounding mechanism to correct hallucinated tokens locally and a specialized beam search algorithm to reduce OH globally while preserving text generation quality. HALC can be easily integrated into any LVLM without additional training, addressing all three types of OH—existence, attribute, and relationship. Extensive experiments show that HALC outperforms state-of-the-art methods across multiple benchmarks.

## K    ADDITIONAL DETAILS

### K.1    LOGIT SPACE FOR VR AND COGNITIVE PROMPTS

From our in-depth analysis of the logit space, we see that language hallucinations in LVLMs have a unique characteristic – the probability of the most confident token in the vocabulary marginally decreased from the probability of the same token in the base LLM when prompted with the same prior context (the most confident token by the LVLM will also be the most confident token by the language-only base LLM as the token on a language hallucination). Thus, decoding-based mitigation

methods like OPERA and VCD benefit from this unique property, and contrasting tokens help mitigate hallucinations.

## K.2 Qualitative Comparison of Greedy vs VCD vs VDGD Decoding

In this subsection, we illustrate several instances (figures 20, 22, 23, 21, 24) from VaLLu and compare their responses for LLaVa-1.5 with Greedy, VCD, and VDGD Decoding.

**Question:** Baxter Company has a relevant range of production between 15,000 and 30,000 units. The following cost data represents average variable costs per unit for 25,000 units of production. If 15,000 units are produced, what is the variable cost per unit?

| | Average Cost per Unit |
|---|---|
| Direct materials | $10 |
| Direct labor | 9 |
| Indirect materials | 3 |
| Fixed manufacturing overhead | 6 |
| Variable manufacturing overhead | 2 |
| Fixed selling and administrative expenses | 8 |
| Variable sales commissions | 14 |

**Question:** Where is it typically found?

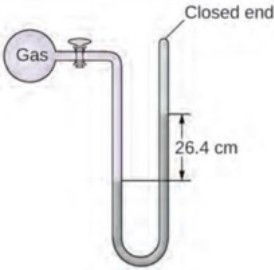

**Greedy Decoding:** To find the variable cost per unit, we need to calculate the total variable cost and divide it by the number of units produced. \n Total variable cost = (Direct materials*Direct labor) + (Indirect materials*Fixed manufacturing overhead) + (Variable manufacturing overhead*Number of units). \n Total variable cost = ($10 * $9) + ($3 * $6) + ($2 * 15,000) = $30,118. \n Variable cost per unit = Total variable cost/Number of units=30,118/15,000 = $2.00. Therefore, if 15,000 units are produced, the variable cost per unit is $2.00.

**Greedy Decoding:** The image shows a wasp on a yellow flower.

**VDGD Decoding:** To find the variable cost per unit, we need to calculate the total variable cost and divide it by the number of units produced. \n Total variable cost = (Direct labor cost + Indirect materials cost + Variable manufacturing overhead cost + Variable sales commissions cost)*Number of units. \n Total variable cost = ($9 + $3 + $2 + $14)*25,000 = $950,000. \n Variable cost per unit = Total variable cost/Number of units = $950,000/15,000 = $63.33. \n Therefore, if 15,000 units are produced, the variable cost per unit would be $63.33.

**VDGD Decoding:** The wasp is typically found in gardens, meadows, and other natural habitats where it can find flowers to feed on. It may also be found near human settlements if the environment provides suitable nesting sites and food sources.

**Description:** The image is a table that lists the average cost per unit for various production-related expenses. The table has two columns: one for the type of expense and another for the average cost per unit. The types of expenses listed are direct materials, direct labor, indirect materials, fixed manufacturing overhead, variable manufacturing overhead, fixed selling and administrative expenses, and variable sales commissions. In\nThe average cost per unit for each expense is as follows:\n- Direct materials: $10\n- Direct labor: $9\n- Indirect materials: $3\n- Fixed manufacturing overhead: $6\n- Variable manufacturing overhead: $2\n- Fixed selling and administrative expenses: $8\n- Variable sales commissions: $14\n\nThere are no other objects or text in the image.

**Description:** The image depicts a wasp perched on a yellow flower. The wasp is brown and black in color, with its wings spread out as it sits on the flower. There are several other yellow flowers in the background, creating a vibrant scene.

Figure 19: Qualitative example for VaLLu benchmark also showing the descriptions.

**Question:** The pressure of a sample of gas is measured at sea level with a closed-end manometer. The liquid in the manometer is mercury. Determine the pressure of the gas in torr.

**Greedy Decoding:** The image shows a closed-end manometer with a silver tube and a mercury-filled gauge. The manometer is measuring the pressure of a gas, specifically, the pressure of a gas at sea level. The tube is connected to the manometer, and the pressure is indicated by the height of the mercury in the tube. The manometer is measuring the pressure in torr, which is a unit of pressure. The image provides a clear visual representation of the manometer and its function in measuring gas pressure.

**VCD Decoding:** The pressure of a sample of gas is measured at sea level with a closed-end manometer filled with mercury. The manometer has a scale that measures the pressure in units of torr. The manometer is connected to the gas sample through a metal tube that has a closed end to prevent the mercury from leaking out. The manometer is placed in a vacuum chamber to minimize external influences on the measurement. The pressure reading is shown on the manometer scale, which is typically a logarithmic scale that allows for a wide range of pressure measurements to be shown in a compact space. The manometer is a useful tool for measuring the pressure of gases in various applications, such as scientific research and industrial processes.

**VDGD Decoding:** The image shows a closed-end manometer with a metal tube containing mercury liquid, measuring the pressure of a gas at sea level. The tube is connected to a metal pipe with a scale on it, indicating the pressure in torr units. The manometer is placed on a white background, making it the main focus of the image. The pressure of the gas is 26.4 torr.

Figure 20: Qualitative example 1 for VaLLu benchmark.

**Question:** What is the net sales figure for Stanley Black and Decker?

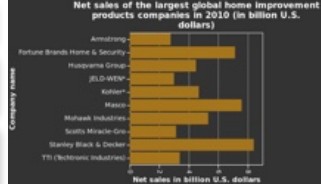

**Greedy Decoding:** The net sales figure for Stanley Black and Decker is $10.5 billion.

**VCD Decoding:** The net sales figure for Stanley Black and Decker is $1.5 billion in 2010.

**VDGD Decoding:** The net sales figure for Stanley Black and Decker is $13.3 billion in 2010.

Figure 21: Qualitative example 2 for VaLLu benchmark.

**Question:** Dose response data was collected during the preclinical testing of four drugs for the treatment of acute heart failure. Which drug studied was the most efficacious?

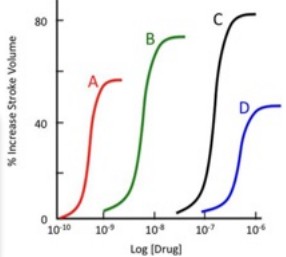

**Greedy Decoding:** Based on the dose-response data, the most efficacious drug studied was the one with the highest efficacy. In the image, there are four different colored lines representing the efficacy of the four drugs. The line with the highest efficacy is the one that has the most significant effect on the outcome. This information can be used to determine which drug is the most effective in treating acute heart failure.

**VCD Decoding:** The most efficacious drug studied was the one with the highest log dose (log D). Log dose is a measure of the concentration of a substance in the blood relative to the concentration in the liver, which is an important factor in determining the efficacy of a drug. In this case, the log dose for the green line is higher than the log dose for the other lines, indicating that the green line represents the most effective drug among the four studied for the treatment of acute heart failure.

**VDGD Decoding:** Based on the image, the most efficacious drug studied was D. The graph shows that the dose-response relationship for D was the steepest, indicating that the drug had the highest efficacy at lower doses. This suggests that D may have a more potent effect on acute heart failure treatment compared to the other three drugs, A, B, and C.

Figure 22: Qualitative example 3 for VaLLu benchmark.

**Question:** Three identical rectangles are put together to form rectangle ABCD, as shown in the figure below. Given that the length of the shorter side of each of the smaller rectangles 5 feet, what is the area in square feet of rectangle ABCD?

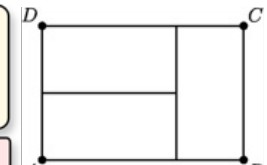

**Greedy Decoding:** The area of rectangle ABCD is 12 square feet.

**VCD Decoding:** To find the area of the rectangle formed by the three identical rectangles, we need to find the length and width of each rectangle and then multiply them together. In the image, the length of the smaller side of each rectangle is given as 5 feet. Since there are three identical rectangles, we can find the total length by adding the lengths together: Length = (3 x 5 feet) = 15 feet. Now we need to find the width of each rectangle. Since the length is the same for all three rectangles, we can find the width by dividing the length by the number of rectangles: Width = Length / Number of Rectangles = 15 feet / 3 = 5 feet. Now we have the dimensions for each rectangle: length is 15 feet and width is 5 feet. To find the area of the rectangle formed by these three identical rectangles, we multiply the length and width together: Area = Length x Width = 15 feet x 5 feet = 75 square feet. So the area in square feet of the rectangle formed by the three identical rectangles is 75 square feet.

**VDGD Decoding:** Given that the length of the shorter side of each of the smaller rectangles is 5 feet, the area in square feet of rectangle ABCD can be calculated by multiplying the length of the longer side by the width of the shorter side. The length of the longer side is 10 feet, and the width of the shorter side is 5 feet. Therefore, the area of rectangle ABCD is 50 square feet.

Figure 23: Qualitative example 4 for VaLLu benchmark.

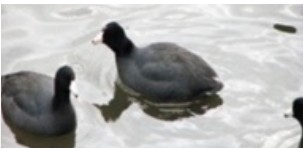

**Question:** Where can the animal be found?

**Greedy Decoding:** The animal, a duck, can be found in a body of water, specifically a lake.

**VCD Decoding:** The animal can be found in the wild, specifically in the tropical rainforests of Central and South America. It is also found in some parts of Africa and Asia. The animal is typically found in areas with plenty of water and vegetation, such as rivers and swamps. It is also known to be found in some parts of the United States and Canada, but these populations are generally small and isolated. The animal is also found in some parts of the world where it is kept in captivity, such as zoos and wildlife parks.

**VDGD Decoding:** The animal can be found in a body of water, possibly a lake or a pond, surrounded by other birds.

Figure 24: Qualitative example 5 for VaLLu benchmark.

### K.3 NOISY EXAMPLES IN EXISTING DATASETS

To maintain a high quality of the benchmark and eliminate noisy examples, we performed a manual review of the existing datasets. This was crucial to ensure that the data used in our paper was reliable. It was also essential for the validity of our research findings. We removed questions that did not have a definite ground truth answer. We also made sure that the questions were answerable using a combination of general knowledge and reasoning. This filtering was done by the paper authors and has been approved by our institution's Institutional Review Board (IRB). We share a few examples (figures: 25, 26, 27, 28, 29, 30, 31, 32) of noise in the existing datasets below:

**Question:** You are a consultant to a large manufacturing corporation that is considering a project with the following net after-tax cash flows (in millions of dollars).

| Company | $1 Discount Store | Everything $5 |
|---|---|---|
| Forecasted return | 12% | 11% |
| Standard deviation of returns | 8% | 10% |
| Beta | 1.5 | 1.0 |

**Sanity Rationale:** The question asks for the net present value of a project and provides specific financial parameters such as project beta, risk-free rate, and expected market return. However, the image provided shows information about two different companies, including their forecasted returns, standard deviations of returns, and betas, which are unrelated to the specific project's cash flows mentioned in the question. The image neither shows the net after-tax cash flows for the project in question nor any relevant data to perform a net present value calculation based on the project's described financial conditions. Thus, the image does not support answering the question adequately.

Figure 25: Noisy example 1 for MMMU benchmark.

**Question:** Fig. 8.15 shows the distribution of 12 spot heights with a regular 20 m spacing covering a rectangular area which is to be graded to form a horizontal plane having an elevation of 10.00 m. Calculate the volume of the earth.

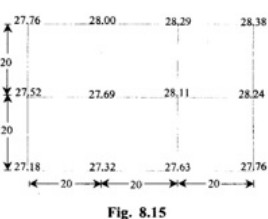

**Sanity Rationale:** The image provides the elevations at various points and the distances between these points, but it does not provide enough information to calculate the volume of earth needed to grade the area to a horizontal plane at an elevation of 10.00 m. The calculation of volume would require additional information such as the method of interpolation between the spot heights and the specific earthwork calculation techniques to be used. Since the question cannot be definitively answered solely with the provided image and general knowledge, the response is no.

Figure 26: Noisy example 2 for MMMU benchmark.

**Question:** What is the current employment rate in the UK's travel and tourism sector?

**Sanity Rationale:** The question inquires about the current employment rate in the UK's travel and tourism sector. The provided image, however, shows data from 2019 and does not specify whether the numbers represent rates or absolute job counts, nor does it give any indication of the employment rate. Additionally, the image is historical and may not accurately reflect the current employment situation. Therefore, the image does not provide the necessary information to answer the question about the current employment rate.

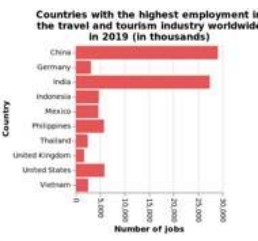

Figure 27: Noisy example 1 for MMC benchmark.

**Question:** Did the population of 40-59 year olds change from 2016 to 2020?

**Sanity Rationale:** The image provided is a solid color block without any distinguishable data, axes, or markers that would allow for the evaluation of population changes over time. Therefore, it is impossible to answer the question about the change in the population of 40-59 year olds from 2016 to 2020 using this image.

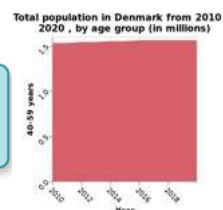

Figure 28: Noisy example 2 for MMC benchmark.

**Question:** Hint: Please answer the question requiring an integer answer and provide the final value, e.g., 1, 2, 3, at the end. Question: What is the age gap between these two people in image? (Unit: years)

**Sanity Rationale:** The question asks for the age gap between two people shown in the image, but it's impossible to accurately determine their ages just by looking at the photograph. Age estimation from a photograph without additional information such as the date the photograph was taken or personal details about the individuals is highly speculative. Therefore, the question does not meet the condition of having a definite, ground truth answer.

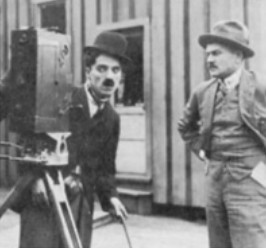

Figure 29: Noisy example 1 for MathVista benchmark.

**Question:** Hint: Please answer the question requiring a floating-point number with two decimal places and provide the final value, e.g., 1.23, 1.34, 1.45, at the end. Question: In the figure, what is the magnitude of the potential difference across the $20\Omega$ resistor? Answer in unit of W (3 sig.fig.).

**Sanity Rationale:** The question asks for the magnitude of the potential difference across a resistor using a unit of 'W' which stands for watts, a unit of power, not potential difference. The correct unit for potential difference should be volts (V). Therefore, the question contains a fundamental error in its request for a unit, making it impossible to answer as stated.

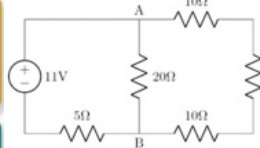

Figure 30: Noisy example 2 for MathVista benchmark.

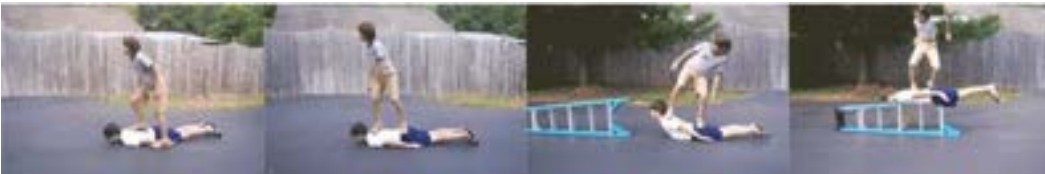

**Question:** According to the positive sequence of the images, are they skating to right? Answer in one word.

**Sanity Rationale:** The question asks whether the individuals in the image are skating to the right, based on the sequence of images. However, the images do not show any skating activity. The sequence depicts two individuals performing a stunt or trick where one person runs and jumps over another lying on the ground. There are no skates visible, and the activity shown does not involve skating. Therefore, the question is not answerable from the image as it assumes an incorrect context and activity that is not depicted.

Figure 31: Noisy example 1 for HallusionBench benchmark.

**Question:** If angle BAF equals to angle CAF, is CF equal to BF? Please answer yes or no.

**Sanity Rationale:** Based on the image provided, the question asks whether CF is equal to BF given that angle BAF equals angle CAF. The image shows a triangle ABC with a point F on line segment BC and other lines and points within the triangle. However, the image does not provide any explicit information or annotations (like length markings or angle measurements) that confirm whether line segments CF and BF are equal, nor does the condition of BAF equaling CAF automatically imply that CF equals BF without additional context or geometric properties being specified (such as triangle ABC being isosceles with AB = AC). Therefore, the question cannot be definitively answered just by examining the image and applying basic geometric reasoning.

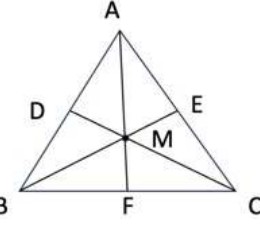

Figure 32: Noisy example 1 for HallusionBench benchmark.

### K.4 ADDITIONAL DETAILS ON ALGORITHM 1

In this Section, we describe our algorithm verbally.

**Token Distribution Analysis.** We perform the token-distribution analysis proposed by (Lin et al., 2023). Specifically, for a given instruction-response-image triplet, the instruction $i = \{i_1, i_2, \cdots\}$ and the image $v$ is first input to the aligned (or fine-tuned) model to obtain its response $r = \{r_1, r_2, \cdots\}$ via greedy decoding. Next, for each position $t$ in the response, a 'context' at this position is defined as to be $x_t = i + \{r_1, \cdots, r_{t-1}\}$. This "context" is then input to the base model (model from which the aligned model was instruction-tuned) to obtain its probability distribution for predicting the next token at position $t$, $P_{\text{base}}$. We then define **Base Rank** as follows: Rank in $P_{\text{base}}$ of the token at $t$ with the maximum $P_{\text{align}}$ value. With the base rank denoted as $\eta$, the unshifted, marginal and shifted tokens are defined as when $(\eta = 1)$, $(1 < \eta \leq 3)$ and $(\eta > 3)$ respectively.

**Obtaining Hallucinated Phrases.** Fig. 12 illustrates the prompt used for AMBER evaluation, where an LVLM is prompted for a description of an input image. In addition to scores, our judge, GPT-4, also returns the exact hallucinated phrases by the LVLM. We divide these hallucinated phrases into 3 types: **Object Hallucinations** – There are hallucinated visual elements which are objects. **Relation Hallucinations** – **Action/Verb Hallucination** – Hallucinated visual elements that are not objects but just actions or verbs, e.g., walking, etc. **Relation Hallucination** – Hallucinated visual elements that define relationships (spatial or other kinds) between objects.

**Filtering visual elements in Hallucinated Phrases.** We prompt LLaMa-3 with the image description output by the LVLM, to extract the phrases from the image description that are visual elements. This ensures filtering out stop-words and all other words that are not visual elements and thus aids our algorithm in getting a precise count of each category of hallucination.

**Causal Categorization of Hallucinations.** Next, based on the Base Rank formulation and the exact hallucinated phrases detected by our judge (GPT-4), we propose an algorithm (Algorithm 1) to automatically categorize identified hallucinations into the 4 different categories. Precisely, for every hallucinated phrase returned by the judge, we first check if the word in the phrase is a visual element. We do this using the list of visual elements output by LLaMa-3. Next, for the first token of each word of the hallucinated phrase, we check the Base Rank of the token.

1. If the Base Rank is 0, we attribute the phrase to be a language hallucination. This is as simple as the phrase was forced to be generated by the base LLM itself without looking at the image.
2. Next, if the Base Rank is more than 0 and the hallucination can be found in the top-k retrieved elements from the IT dataset the model was aligned using, we attribute it to be an IT hallucination (Ghosh et al., 2024). To retrieve top-k image-instruction pairs that are closest to the prompt image, we employ CLIP (Radford et al., 2021) similarity. Specifically, we calculate the similarity between the image embedding of the image associated with the input prompt and the text embedding of the description/response associated with the IT instances. Next, for every prompt, we retrieve the top-25 IT instances and search the hallucinated phrase returned by the judge (in descriptions/responses) using string matching. We show how varying values of $k$ change the final counts for our hallucination categories in Appendix K.5.

3. Next, if the Base Rank is more than 0, the hallucination is not an IT hallucination and the hallucination is a **Relation Hallucination**, we attribute it to be a style hallucination. LVLMs fine-tuned on datasets synthesized from stronger closed-source LVLMs generally learn to mimic their style. One such style is adding more details to an identified object, by relating it to other objects or world knowledge. Our in-depth analysis shows that LVLMs, striving for lengthier, more detailed responses to imitate style, hallucinate and generate wrongful relations for identified objects. Our relation hallucinations returned by the judge precisely capture this, and thus we attribute relation hallucinations with a base rank of more than 0 to be Style hallucinations.

4. Finally, if the Base Rank is more than 0, the hallucination is identified as an Obejct hallucination by the judge, we attribute it to be a Vision Hallucination. These are caused by ineffective recognition of objects in the image (due to low-resolution images, weak encoder, etc.)

### K.5 RESULTS FOR VARIOUS VALUES OF $k$ FOR IT HALLUCINATIONS

Fig. 33 shows that even for higher and lower values of $k$, our hypothesis and claims stay unchanged as changing the value of $k$ does not affect the overall frequency distribution of categories.

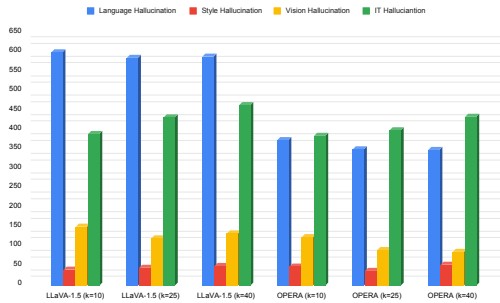

Figure 33: Frequency comparison of hallucination categories for top-$k$=10,25,40

### K.6 ELBOW METHOD FOR LOGIT ANALYSIS

The Knee/Elbow Detection Method operates by plotting the cumulative distribution of the sorted probabilities and seeking the point that maximizes the distance from the line connecting the first and the last points of this cumulative plot. Conceptually, this corresponds to finding the point of maximum curvature on the plot, indicative of the most pronounced change in the distribution's progression. It offers a robust approach by identifying the point in a sorted probability vector where the rate of decrease in probabilities shows a significant bend or 'elbow'. This method is particularly useful when the probabilities are distributed such that the top few are similar and significantly higher than the rest, which then decrease sharply. The method proves to be effective in our experiments with multinomial sampling from a distribution, where it allows us to focus on the probabilities that correspond to the tokens that are most likely to be selected by sampling. This method ensures that the chosen cutoff $k$ dynamically adjusts to the specific characteristics of each probability distribution encountered, aligning with the overarching goal of maintaining sampling relevance and computational efficiency.

### K.7 EXAMPLES OF REPHRASED PROMPTS

To determine whether LVLM hallucinations stem from the limitations of the base LLM or from alignment issues, we ask GPT-4 to rewrite the instructions in prompts in such a way that the base LLM can answer the question without any image input as described in section 4.2. We share a few examples (figures: 34, 35, 36, 37, 38, 39, 40, 41) of rephrased prompts in the existing datasets below:

**Question:** The composition of the Fingroup Fund portfolio is as follows: The fund has not borrowed any funds, but its accrued management fee with the portfolio manager currently totals $30,000. There are 4 million shares outstanding. What is the net asset value of the fund?

| Stock | Shares | Price |
|-------|---------|-------|
| A | 200,000 | $35 |
| B | 300,000 | 40 |
| C | 400,000 | 20 |
| D | 600,000 | 25 |

**Rephrased Instruction:** A portfolio consists of four types of stocks. Stock A has 200,000 shares priced at $35 each, Stock B has 300,000 shares priced at $40 each, Stock C has 400,000 shares priced at $20 each, and Stock D has 600,000 shares priced at $25 each. The fund has not borrowed any funds, but has an accrued management fee of $30,000. There are 4 million shares outstanding. Calculate the net asset value per share for this fund.

Figure 34: Rephrased prompt example for VaLLu benchmark.

**Question:** After school, Maya and Naomi headed to the beach, 6 miles away. Maya decided to bike while Naomi took a bus. The graph below shows their journeys, indicating the time and distance traveled. What was the difference, in miles per hour, between Naomi's and Maya's average speeds?

**Rephrased Instruction:** Maya and Naomi traveled to a destination 6 miles away. Maya biked and completed the journey in 30 minutes. Naomi took a bus and completed the journey in 10 minutes. Calculate the difference, in miles per hour, between Naomi's and Maya's average speeds.

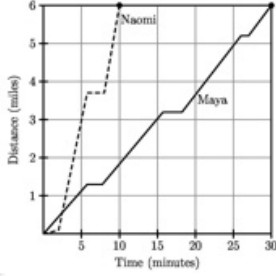

Figure 35: Rephrased prompt example for VaLLu benchmark.

**Question:** If this area was selectively destroyed on both sides, could vomiting be induced by either pharyngeal tickling or gastric distress?

**Rephrased Instruction:** If the area of the brainstem that includes the dorsal motor nucleus of the vagus nerve (marked as X in a typical diagram) and the area postrema (often indicated with an arrow near the fourth ventricle in diagrams) were selectively destroyed on both sides, could vomiting still be induced by pharyngeal tickling or gastric distress?

Figure 36: Rephrased prompt example for MMMU benchmark.

**Question:** Suppose a three-factor model is appropriate to describe the returns of a stock. Information about those three factors is presented in the following:What is the systematic risk of the stock return?

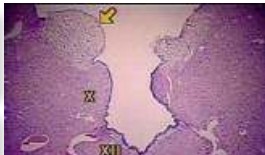

| Factor | β | Expected Value | Actual Value |
|--------|------|----------------|--------------|
| GDP | .0000734 | $17,034 | $17,863 |
| Inflation | −.90 | 2.80% | 2.6% |
| Interest rates | −.32 | 3.70% | 3.5% |

**Rephrased Instruction:** Given a three-factor model with the following factors and their respective beta values: GDP with a beta of 0.0000734, Inflation with a beta of -0.90, and Interest rates with a beta of -0.32, can you calculate the systematic risk of the stock return?

Figure 37: Rephrased prompt example for MMMU benchmark.

**Question:** Hint: Please answer the question requiring an integer answer and provide the final value, e.g., 1, 2, 3, at the end. Question: How many objects are preferred by more than 5 people in at least one category?

**Rephrased Instruction:** Consider a bar graph with five objects labeled as veto, saw, calf, oxen, and fusion, each represented in two categories: grease and picnic. The number of people who prefer each object in each category ranges from 0 to 10. How many objects are preferred by more than 5 people in at least one of these categories?

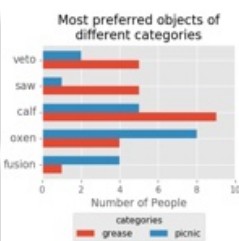

Figure 38: Rephrased prompt example for MathVista benchmark.

**Question:** Hint: Please answer the question requiring a floating-point number with two decimal places and provide the final value, e.g., 1.23, 1.34, 1.45, at the end. Question: what is the difference between the largest and smallest number of cases?

**Rephrased Instruction:** Given the number of cases in thousands for various cities: Quezon City has 106.71 thousand cases, Cavite has 72.34 thousand cases, City of Manila has 66.88 thousand cases, Laguna has 58.31 thousand cases, and Rizal has 55.57 thousand cases. What is the difference between the largest and smallest number of cases, expressed as a floating-point number with two decimal places?

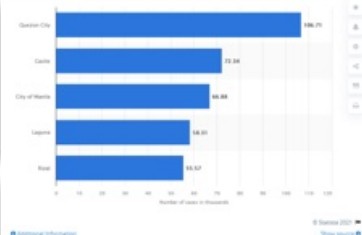

Figure 39: Rephrased prompt example for MathVista benchmark.

**Question:** which year were these orange objects first used in BCE?

**Rephrased Instruction:** In which year were terracotta roof tiles first used in BCE?

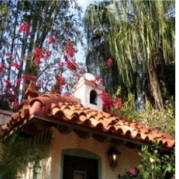

Figure 40: Rephrased prompt 1 for SynthDoG benchmark.

**Question:** This animal can be mostly found in which region?

**Rephrased Instruction:** In which region can the Ugandan Kob, a type of antelope, be mostly found?

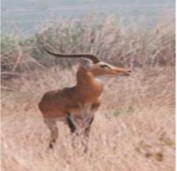

Figure 41: Rephrased prompt 2 for SynthDoG benchmark.

## K.8 COMPUTE INFRASTRUCTURE

All our analysis, inference and baseline experiments are conducted on a node of 4 NVIDIA RTX A6000 GPUs, with 128GB RAM and 10 CPU cores. The evaluations are conducted using gpt-4-turbo-2024-04-09 model.

## L ADDITIONAL RESULTS

### L.1 COMPARISON OF VARIOUS LVLMS ON REPHRASED PROMPTS

Table 5 provides quantitative values for rephrased prompts without image input for various LVLMs, detailed in section 4.2 of the main paper.

| Dataset | Model | Helpfulness | Clarity | Factuality | Depth | Engagement |
|---------|-------|-------------|---------|------------|-------|------------|
| VaLLu | LLaVA-v1.5-t | 3.05 | 4.02 | 2.83 | 2.70 | **3.85** |
| | mPLUG-Owl2-t | 3.70 | 4.15 | 3.14 | 3.11 | 3.24 |
| | InternLM-X-t | **4.28** | **4.58** | **3.66** | **3.62** | 3.57 |
| MMMU | LLaVA-v1.5-t | 1.97 | 3.25 | 1.69 | 1.87 | **3.25** |
| | mPLUG-Owl2-t | 2.48 | 3.05 | 1.63 | 2.22 | 2.64 |
| | InternLM-X-t | **3.51** | **3.93** | **2.43** | **2.98** | 3.01 |
| MathVista | LLaVA-v1.5-t | 2.53 | 3.58 | 2.21 | 2.10 | **3.44** |
| | mPLUG-Owl2-t | 3.56 | 4.05 | 3.04 | 2.52 | 2.79 |
| | InternLM-X-t | **4.39** | **4.65** | **3.99** | **2.91** | 2.95 |
| Oven | LLaVA-v1.5-t | 3.38 | 4.48 | 2.95 | 2.32 | 3.35 |
| | mPLUG-Owl2-t | 3.62 | 4.56 | 2.99 | 2.47 | 3.37 |
| | InternLM-X-t | **3.80** | **4.58** | **3.76** | **2.89** | **3.52** |

Table 5: Performance comparison of various LVLMs on rephrased prompts without images as described in Section 4.2

### L.2 COMPARISON OF VARIOUS LVLMS ON VALLU BENCHMARK WITH NO IMAGE INPUT.

Table 6 provides comparison of various LVLMs on VaLLU benchmark when prompted with no image but image descriptions are provided.

| Methodology | LLaVA-v1 | LLaVA-v1.5 | LLaVA-v1.6 | mPLUG-Owl2 | InternLM-X | CogVLM |
|-------------|----------|------------|------------|------------|------------|--------|
| Vanilla-greedy | 1.67 | 1.78 | 2.07 | 1.76 | 2.02 | 2.23 |
| VCD | 1.31 | 1.29 | 1.40 | 1.42 | 2.04 | 2.21 |
| OPERA | 1.65 | 1.43 | 2.29 | 1.91 | 2.30 | 2.44 |
| Woodpecker | 1.77 | 1.85 | 2.29 | 2.02 | 2.41 | 2.60 |

Table 6: Comparison of LVLMs performance when prompted without an image but image descriptions are provided on VaLLu benchmark. The values correspond to the "Factuality" metric.

### L.3 COMPARISON OF VARIOUS LVLMS ON IMAGE DESCRIPTION

Table 7 provides quantitative values for faithful image description generations for various LVLMs on multiple benchmarks, detailed in section 4.2 of the main paper.

### L.4 COMPARISON OF LVLMS ON DIFFERENT CAPABILITIES ON THE PROPOSED VALLU BENCHMARK

Table 8 provides a comparison of LVLMs on different capabilities on the proposed VaLLu benchmark.

| Dataset | Model | Helpfulness | Clarity | Factuality | Depth | Engagement |
|---|---|---|---|---|---|---|
| AMBER | LLaVA-v1 | 3.79 | 4.46 | 3.2 | 3.27 | 0.96 |
| | LLaVA-1.5 | 4.29 | 4.73 | 3.91 | 3.44 | 1.26 |
| | LLaVA-1.6 | 4.30 | 4.78 | 3.85 | 4.29 | 1.27 |
| | mPLUG-Owl2 | 4.02 | 4.58 | 3.48 | 3.27 | 0.71 |
| | InternLM-X | 4.59 | 4.86 | 4.37 | 4.00 | 1.40 |
| | CogVLM | 4.68 | 4.93 | 4.49 | 3.96 | 1.28 |
| | GPT-4-Turbo | **4.89** | **4.96** | **4.79** | **4.64** | **1.50** |
| OCR | LLaVA-v1 | 1.11 | 2.09 | 1.11 | 1.14 | 0.47 |
| | LLaVA-1.5 | 1.07 | 2.24 | 1.17 | 1.15 | 0.48 |
| | LLaVA-1.6 | 2.51 | 2.92 | 2.63 | 1.99 | 1.11 |
| | mPLUG-Owl2 | 1.21 | 1.15 | 1.16 | 1.10 | 0.64 |
| | InternLM-X | 1.11 | 1.12 | 1.12 | 1.03 | 0.67 |
| | CogVLM | 3.12 | 3.04 | 2.68 | 2.53 | 1.47 |
| | GPT-4-Turbo | **4.15** | **4.02** | **3.78** | **3.98** | **1.67** |
| MMMU | LLaVA-v1 | 1.77 | 2.73 | 1.50 | 1.65 | 0.46 |
| | LLaVA-1.5 | 2.45 | 3.34 | 2.00 | 2.23 | 0.59 |
| | LLaVA-1.6 | 3.54 | 3.98 | 2.95 | 3.39 | 0.99 |
| | mPLUG-Owl2 | 2.94 | 3.64 | 2.56 | 2.33 | 0.82 |
| | InternLM-X | 3.60 | 4.10 | 3.24 | 3.15 | 1.05 |
| | CogVLM | 4.03 | 4.38 | 3.57 | 3.43 | 1.03 |
| | GPT-4-Turbo | **4.86** | **4.92** | **4.73** | **4.56** | **1.15** |
| MathVista | LLaVA-v1 | 1.95 | 2.84 | 1.95 | 1.85 | 0.76 |
| | LLaVA-1.5 | 2.57 | 3.47 | 2.23 | 2.36 | 0.98 |
| | LLaVA-1.6 | 3.68 | 4.07 | 3.17 | 3.48 | 1.12 |
| | mPLUG-Owl2 | 3.14 | 3.85 | 2.78 | 2.46 | 1.03 |
| | InternLM-X | 3.91 | 4.32 | 3.45 | 3.35 | 1.23 |
| | CogVLM | 4.20 | 4.52 | 3.78 | 3.66 | 1.24 |
| | GPT-4-Turbo | **4.54** | **4.97** | **4.12** | **4.10** | **1.35** |
| MMC | LLaVA-v1 | 1.81 | 2.90 | 1.51 | 1.75 | 0.56 |
| | LLaVA-1.5 | 2.66 | 3.49 | 2.21 | 2.42 | 0.71 |
| | LLaVA-1.6 | 4.09 | 4.32 | 3.50 | 3.90 | 1.09 |
| | mPLUG-Owl2 | 3.63 | 4.15 | 3.32 | 2.60 | 0.97 |
| | InternLM-X | 3.72 | 4.17 | 3.28 | 3.25 | 1.10 |
| | CogVLM | 4.27 | 4.58 | 3.91 | 3.92 | **1.32** |
| | GPT-4-Turbo | **4.81** | **4.89** | **4.57** | **4.49** | 1.12 |

Table 7: Performance comparison of various LVLMs on popular benchmarks for *image description*. While AMBER has real-world scenes, others do not. Experiment described in Section 3.2.

| Methodology | VP | VP+Info-Seek | VP+Info-Seek+Reasoning | VP+Reasoning |
|---|---|---|---|---|
| LLaVA-v1 | 1.62 | 2.60 | 1.67 | 1.70 |
| LLaVA-v1.5 | 2.59 | 3.18 | 1.50 | 1.76 |
| LLaVA-v1.6 | 3.46 | 3.68 | 1.67 | 2.15 |
| mPLUG-Owl2 | 2.56 | 3.23 | 1.67 | 1.94 |
| InternLM-X | 3.75 | 3.90 | 2.33 | 2.56 |
| CogVLM | 3.00 | 3.37 | 1.25 | 2.08 |

Table 8: Comparison of LVLMs on different capabilities on the proposed VaLLu benchmark for the "Factuality" metric.

## L.5 Comparison of post-truncation probability statistics on AMBER and MMMU benchmarks

Table 9 provides post-truncation probabilities for all LVLMs on AMBER and MMMU, this is an extension of the statistic shown in section 4.3 of the main paper.

| Dataset | Model | k | Variance | Range | Avg. |
|---------|-------|-----|----------|-------|------|
| AMBER | LLaVA-v1 | 4.19 | 0.05 | 0.31 | 0.23 |
| | LLaVA-v1.5 | 3.91 | 0.07 | 0.39 | 0.29 |
| | LLaVA-v1.6 | 3.78 | 0.08 | 0.43 | 0.32 |
| | mPLUG-Owl2 | 3.65 | 0.06 | 0.36 | 0.28 |
| | InternLM-X | 3.56 | 0.09 | 0.44 | 0.36 |
| | CogVLM | 3.16 | 0.91 | 0.47 | 0.38 |
| MMMU | LLaVA-v1 | 3.68 | 0.08 | 0.39 | 0.27 |
| | LLaVA-v1.5 | 3.53 | 0.09 | 0.41 | 0.23 |
| | LLaVA-v1.6 | 3.40 | 0.09 | 0.42 | 0.24 |
| | mPLUG-Owl2 | 3.32 | 0.10 | 0.44 | 0.26 |
| | InternLM-X | 3.26 | 0.14 | 0.46 | 0.28 |
| | CogVLM | 3.16 | 0.19 | 0.47 | 0.25 |

Table 9: Post-truncation probability statistics for various models using the elbow method. All values are computed across all hallucinated tokens in each dataset.

## L.6 Comparison of VaLLu with Mitigation Baselines

Table 2 provides quantitative values for comparison of VDGD and other baselines on the VaLLu benchmark, detailed in section 5.3 of the main paper.

## L.7 Comparison of MMMU with Mitigation Baselines

Table 11 provides quantitative values for comparison of VDGD and other baselines on the MMMU benchmark, detailed in section 3.2 of the main paper.

## L.8 Comparison of MathVista with Mitigation Baselines

Table 12 provides quantitative values for comparison of VDGD and other baselines on the MathVista benchmark, detailed in section 3.2 of the main paper.

## L.9 Comparison of Oven with Mitigation Baselines

Table 13 provides quantitative values for comparison of VDGD and other baselines on the Oven benchmark, detailed in section 3.2 of the main paper.

## M Computational Efficiency

Table 14 presents a computational analysis of various methods evaluated on LLaVA 1.5 using a 48GB GPU, comparing their average inference time and computational cost measured in teraFLOPs (T). While the VDGD method requires more computation than the baseline methods, the increase is modest and within a reasonable range when considering the significant performance gains it offers. Specifically, VDGD has an average inference time of 2.5 seconds and a computational cost of 11.9 teraFLOPs, which is a manageable increase compared to the Vanilla-greedy method's 1.3 seconds and 9.3 teraFLOPs. Similarly, the combination of VDGD with a Small Captioning Model Ramos et al. (2023) demonstrates acceptable computational demands with an inference time of 2.1 seconds and 11.4 T. These results suggest that the enhanced performance of VDGD methods is achieved with computational requirements that are well within practical limits, making it a viable option.

| Model | Methodology | Helpfulness | Clarity | Factuality | Depth | Engagement |
|---|---|---|---|---|---|---|
| LLaVA-v1 | Vanilla-greedy | 2.61 | 3.69 | 1.95 | 1.92 | 1.25 |
| | Vanilla-sampling | 2.64 | 3.73 | 1.86 | 1.94 | 1.27 |
| | VCD | 1.70 | 2.55 | 1.47 | 1.81 | **1.59** |
| | OPERA | 2.60 | 3.75 | 2.04 | 1.89 | 1.30 |
| | Woodpecker | **2.68** | **3.81** | 2.01 | 1.95 | 1.28 |
| | LRV | 2.64 | 3.78 | 1.98 | 1.94 | 1.36 |
| | LURE | 2.60 | 3.54 | 2.03 | 1.89 | 1.40 |
| | **VDGD (ours)** | 2.38 | 3.39 | **2.16** | **2.46** | 1.34 |
| LLaVA-1.5 | Vanilla-greedy | 2.64 | 3.72 | 2.03 | 2.03 | 1.25 |
| | Vanilla-sampling | 2.57 | 3.65 | 2.01 | 1.92 | 1.22 |
| | VCD | 1.94 | 2.99 | 1.55 | 1.65 | 1.05 |
| | OPERA | 2.73 | 3.74 | 2.05 | 2.04 | 1.22 |
| | Woodpecker | 2.74 | 3.87 | 2.05 | 2.03 | 1.29 |
| | LRV | 2.76 | 3.82 | 2.10 | 2.03 | 1.28 |
| | LURE | 2.72 | 3.76 | 2.03 | 2.05 | 1.27 |
| | **VDGD (ours)** | **2.97** | **3.91** | **2.64** | **2.27** | **1.54** |
| LLaVA-1.6 | Vanilla-greedy | 3.08 | 3.81 | 2.63 | 2.67 | 1.43 |
| | Vanilla-sampling | 3.10 | 3.85 | 2.64 | 2.63 | 1.40 |
| | VCD | 2.01 | 2.96 | 1.80 | 2.10 | 1.20 |
| | OPERA | 3.10 | 4.01 | 2.62 | 2.60 | 1.42 |
| | Woodpecker | 3.16 | 4.00 | 2.67 | 2.58 | 1.44 |
| | LRV | 3.15 | 3.98 | 2.65 | 2.62 | 1.39 |
| | LURE | 3.11 | 4.02 | 2.64 | 2.64 | 1.48 |
| | **VDGD (ours)** | **3.51** | **4.04** | **3.16** | **2.87** | **1.58** |
| mPLUG-Owl2 | Vanilla-greedy | 2.73 | 3.64 | 2.20 | 1.96 | 1.28 |
| | Vanilla-sampling | 2.72 | 3.60 | 2.18 | 1.91 | 1.24 |
| | VCD | 2.16 | 2.95 | 1.80 | 1.85 | 1.26 |
| | OPERA | 2.92 | 3.96 | 2.26 | 2.08 | 1.34 |
| | Woodpecker | 2.89 | 3.90 | 2.23 | 2.02 | 1.24 |
| | LRV | 2.90 | 3.94 | 2.19 | 2.05 | 1.28 |
| | LURE | 2.92 | 3.89 | 2.24 | 2.06 | 1.32 |
| | **VDGD (ours)** | **3.14** | **3.98** | **2.72** | **2.22** | **1.50** |
| InternLM-X | Vanilla-greedy | 3.11 | 3.98 | 2.66 | 2.34 | 1.47 |
| | Vanilla-sampling | 3.13 | 4.04 | 2.70 | 2.41 | 1.56 |
| | VCD | 2.56 | 3.26 | 2.32 | 2.19 | 1.41 |
| | OPERA | 3.14 | 4.20 | 2.65 | 2.32 | 1.46 |
| | Woodpecker | 3.13 | 4.12 | 2.60 | 2.28 | 1.41 |
| | LRV | 3.06 | 4.19 | 2.59 | 2.31 | 1.45 |
| | LURE | 3.12 | 4.10 | 2.64 | 2.40 | 1.49 |
| | **VDGD (ours)** | **3.57** | **4.37** | **3.45** | **2.65** | **1.68** |
| CogVLM | Vanilla-greedy | 3.25 | 4.19 | 2.82 | 2.43 | 1.55 |
| | Vanilla-sampling | 3.26 | 4.21 | 2.83 | 2.40 | 1.55 |
| | VCD | 2.80 | 3.62 | 2.63 | 2.10 | 1.58 |
| | OPERA | 3.40 | 4.27 | 2.85 | 2.46 | 1.57 |
| | Woodpecker | **3.44** | **4.30** | 2.91 | 2.45 | 1.59 |
| | LRV | 3.42 | 4.26 | 2.88 | 2.42 | 1.54 |
| | LURE | 3.39 | 4.20 | 2.78 | 2.41 | 1.52 |
| | **VDGD (ours)** | 3.33 | 4.15 | **3.01** | **2.51** | **1.60** |
| GPT-4-Turbo | - | **4.02** | **4.44** | **3.63** | **2.94** | **1.59** |

Table 10: Performance comparison of various LVLMs on VaLLu benchmark. This is an extension of results shown in Table 2

| Model | Methodology | Helpfulness | Clarity | Factuality | Depth | Engagement |
|---|---|---|---|---|---|---|
| LLaVA-v1 | Vanilla-greedy | 1.81 | 2.79 | 1.36 | 1.51 | 1.00 |
| | Vanilla-sampling | 1.82 | 2.82 | 1.38 | 1.56 | 1.06 |
| | VCD | 1.70 | 2.67 | 1.22 | 1.40 | 0.95 |
| | OPERA | 1.88 | 2.87 | 1.45 | 1.61 | 1.07 |
| | Woodpecker | 1.89 | 2.88 | 1.48 | 1.60 | 1.06 |
| | LRV | 1.83 | 2.85 | 1.40 | 1.57 | 1.08 |
| | LURE | 1.91 | 2.90 | 1.52 | 1.63 | 1.09 |
| | **VDGD (ours)** | **2.09** | **2.99** | **1.64** | **1.74** | **1.14** |
| LLaVA-1.5 | Vanilla-greedy | 1.86 | 2.90 | 1.47 | 1.70 | 1.09 |
| | Vanilla-sampling | 1.83 | 2.88 | 1.42 | 1.67 | 1.08 |
| | VCD | 1.70 | 2.52 | 1.34 | 1.58 | 0.97 |
| | OPERA | 2.18 | 3.08 | 1.62 | 1.75 | 1.22 |
| | Woodpecker | 2.15 | 2.93 | 1.57 | 1.63 | 1.18 |
| | LRV | 2.16 | 2.88 | 1.60 | 1.67 | 1.16 |
| | LURE | 2.03 | 2.87 | 1.46 | 1.71 | 1.15 |
| | **VDGD (ours)** | **2.30** | **3.17** | **1.85** | **1.86** | **1.20** |
| LLaVA-1.6 | Vanilla-greedy | 1.85 | 2.67 | 1.54 | 1.80 | 0.97 |
| | Vanilla-sampling | 1.80 | 2.63 | 1.50 | 1.78 | 0.96 |
| | VCD | 1.68 | 2.57 | 1.36 | 1.68 | 0.94 |
| | OPERA | 1.86 | 2.70 | 1.57 | 1.84 | 0.99 |
| | Woodpecker | 1.90 | 2.74 | 1.63 | 1.89 | 1.04 |
| | LRV | 1.88 | 2.72 | 1.61 | 1.85 | 1.02 |
| | LURE | 1.92 | 2.73 | 1.65 | 1.86 | 1.03 |
| | **VDGD (ours)** | **2.11** | **2.89** | **1.81** | **1.92** | **1.05** |
| mPLUG-Owl2 | Vanilla-greedy | 2.24 | 3.12 | 1.58 | 1.65 | 1.16 |
| | Vanilla-sampling | 2.25 | 3.12 | 1.59 | 1.66 | 1.14 |
| | VCD | 2.10 | 2.98 | 1.42 | 1.56 | 1.04 |
| | OPERA | 2.35 | 3.23 | 1.67 | 1.74 | 1.18 |
| | Woodpecker | 2.28 | 3.17 | 1.61 | 1.69 | 1.15 |
| | LRV | 2.34 | 3.24 | 1.66 | 1.72 | 1.14 |
| | LURE | 2.32 | 3.21 | 1.63 | 1.67 | 1.19 |
| | **VDGD (ours)** | **2.48** | **3.36** | **1.82** | **1.78** | **1.23** |
| InternLM-X | Vanilla-greedy | 2.95 | 3.71 | 2.14 | 2.39 | 1.63 |
| | Vanilla-sampling | 2.97 | 3.76 | 2.17 | 2.45 | 1.64 |
| | VCD | 2.76 | 3.54 | 2.03 | 2.26 | 1.57 |
| | OPERA | 3.09 | 3.86 | 2.20 | 2.43 | 1.66 |
| | Woodpecker | 3.07 | 3.83 | 2.19 | 2.46 | 1.68 |
| | LRV | 3.11 | 3.88 | 2.24 | 2.49 | 1.70 |
| | LURE | 3.10 | 3.89 | 2.22 | 2.48 | 1.71 |
| | **VDGD (ours)** | **3.21** | **4.11** | **2.57** | **2.78** | **1.79** |
| CogVLM | Vanilla-greedy | 2.42 | 3.34 | 1.69 | 1.83 | 1.22 |
| | Vanilla-sampling | 2.44 | 3.37 | 1.68 | 1.84 | 1.23 |
| | VCD | 2.22 | 3.15 | 1.53 | 1.68 | 1.17 |
| | OPERA | 2.56 | 3.47 | 1.76 | 1.88 | 1.27 |
| | Woodpecker | 2.57 | 3.47 | 1.78 | 1.90 | 1.26 |
| | LRV | 2.58 | 3.49 | 1.80 | 1.92 | 1.25 |
| | LURE | 2.54 | 3.42 | 1.75 | 1.85 | 1.22 |
| | **VDGD (ours)** | **2.65** | **3.56** | **1.92** | **1.97** | **1.29** |
| GPT-4-Turbo | - | **4.15** | **4.53** | **3.66** | **2.99** | **1.95** |

Table 11: Performance comparison of various LVLMs on MMMU benchmark (only questions tagged with open-ended generation). This is an extension of results shown in Figure 7.

## N  VDGD WITH BEAM SEARCH

VDGD can be seamlessly applied to beam search decoding without modifying its core methodology. Before a set of tokens is selected for each beam, VDGD reweights the logit space of the current

| Model | Methodology | Helpfulness | Clarity | Factuality | Depth | Engagement |
|---|---|---|---|---|---|---|
| LLaVA-v1 | Vanilla-greedy | 3.74 | 4.54 | 2.23 | 2.28 | 1.42 |
| | Vanilla-sampling | 3.72 | 4.56 | 2.24 | 2.27 | 1.43 |
| | VCD | 3.54 | 4.32 | 2.05 | 2.07 | 1.36 |
| | OPERA | 3.80 | 4.63 | 2.29 | 2.35 | 1.50 |
| | Woodpecker | 3.78 | 4.60 | 2.26 | 2.30 | 1.47 |
| | LRV | 3.76 | 4.64 | 2.30 | 2.38 | 1.49 |
| | LURE | 3.74 | 4.59 | 2.25 | 2.28 | 1.46 |
| | **VDGD (ours)** | **3.92** | **4.78** | **2.52** | **2.42** | **1.54** |
| LLaVA-1.5 | Vanilla-greedy | 3.65 | 4.38 | 2.36 | 2.01 | 1.32 |
| | Vanilla-sampling | 3.68 | 4.39 | 2.38 | 2.04 | 1.36 |
| | VCD | 3.38 | 4.09 | 2.35 | 1.87 | 1.10 |
| | OPERA | 3.74 | 4.46 | 2.26 | 1.93 | 1.32 |
| | Woodpecker | 3.46 | 4.53 | 2.34 | 1.97 | 1.38 |
| | LRV | 3.53 | 4.42 | 2.39 | 1.96 | 1.27 |
| | LURE | 3.33 | 4.41 | 2.37 | 1.91 | 1.29 |
| | **VDGD (ours)** | **3.84** | **4.61** | **2.56** | **2.15** | **1.45** |
| LLaVA-1.6 | Vanilla-greedy | 3.72 | 4.38 | 2.50 | 2.34 | 1.50 |
| | Vanilla-sampling | 3.69 | 4.32 | 2.45 | 2.28 | 1.48 |
| | VCD | 3.54 | 4.11 | 2.34 | 2.18 | 1.35 |
| | OPERA | 3.80 | 4.47 | 2.59 | 2.46 | 1.57 |
| | Woodpecker | 3.78 | 4.42 | 2.56 | 2.43 | 1.53 |
| | LRV | 3.75 | 4.40 | 2.54 | 2.42 | 1.52 |
| | LURE | 3.79 | 4.41 | 2.52 | 2.41 | 1.53 |
| | **VDGD (ours)** | **3.90** | **4.53** | **2.79** | **2.51** | **1.58** |
| mPLUG-Owl2 | Vanilla-greedy | 3.79 | 4.43 | 2.48 | 2.08 | 1.33 |
| | Vanilla-sampling | 3.80 | 4.45 | 2.51 | 2.12 | 1.34 |
| | VCD | 3.62 | 4.27 | 2.35 | 1.98 | 1.24 |
| | OPERA | 3.85 | 4.52 | 2.57 | 2.20 | 1.40 |
| | Woodpecker | 3.84 | 4.56 | 2.60 | 2.24 | 1.42 |
| | LRV | 3.82 | 4.54 | 2.59 | 2.22 | 1.41 |
| | LURE | 3.86 | 4.58 | 2.61 | 2.25 | 1.45 |
| | **VDGD (ours)** | **3.99** | **4.65** | **2.76** | **2.36** | **1.52** |
| InternLM-X | Vanilla-greedy | 4.02 | 4.69 | 2.84 | 2.35 | 1.55 |
| | Vanilla-sampling | 4.01 | 4.68 | 2.83 | 2.37 | 1.54 |
| | VCD | 3.82 | 4.37 | 2.66 | 2.10 | 1.44 |
| | OPERA | 4.13 | 4.76 | 2.92 | 2.42 | 1.59 |
| | Woodpecker | 4.07 | 4.72 | 2.88 | 2.40 | 1.58 |
| | LRV | 4.10 | 4.73 | 2.89 | 2.40 | 1.56 |
| | LURE | 4.14 | 4.75 | 2.91 | 2.43 | 1.60 |
| | **VDGD (ours)** | **4.28** | **4.89** | **3.12** | **2.54** | **1.66** |
| CogVLM | Vanilla-greedy | 3.72 | 4.39 | 2.76 | 2.63 | 1.44 |
| | Vanilla-sampling | 3.73 | 4.38 | 2.74 | 2.61 | 1.42 |
| | VCD | 3.58 | 4.27 | 2.64 | 2.50 | 1.37 |
| | OPERA | 3.76 | 4.45 | 2.82 | 2.69 | 1.47 |
| | Woodpecker | 3.75 | 4.44 | 2.79 | 2.64 | 1.45 |
| | LRV | 3.80 | 4.51 | 2.86 | 2.70 | 1.49 |
| | LURE | 3.82 | 4.53 | 2.88 | 2.73 | 1.52 |
| | **VDGD (ours)** | **3.94** | **4.64** | **2.99** | **2.80** | **1.60** |
| GPT-4-Turbo | - | **4.05** | **4.75** | **3.17** | **2.24** | **1.61** |

Table 12: Performance comparison of various LVLMs on MathVista benchmark (only questions tagged with open-ended generation). This is an extension of results shown in Figure 7.

logit based on the KL-Divergence between the current logit and the image description logits. This process is repeated at every decoding step. Another important point to note is that VDGD operates independently of prior response tokens, relying solely on the fixed image description tokens, which

| Model | Methodology | Helpfulness | Clarity | Factuality | Depth | Engagement |
|---|---|---|---|---|---|---|
| LLaVA-v1 | Vanilla-greedy | 3.12 | 4.41 | 2.44 | 2.19 | 1.23 |
| | Vanilla-sampling | 3.14 | 4.42 | 2.45 | 2.20 | 1.26 |
| | VCD | 2.98 | 4.10 | 2.20 | 2.07 | 1.16 |
| | OPERA | 3.20 | 4.48 | 2.50 | 2.22 | 1.28 |
| | Woodpecker | 3.21 | 4.49 | 2.46 | 2.21 | 1.25 |
| | LRV | 3.25 | 4.51 | 2.51 | 2.23 | 1.29 |
| | LURE | 3.20 | 4.47 | 2.49 | 2.20 | 1.28 |
| | **VDGD (ours)** | **3.42** | **4.59** | **2.78** | **2.30** | **1.30** |
| LLaVA-1.5 | Vanilla-greedy | 3.25 | 4.41 | 2.66 | 2.61 | 1.19 |
| | Vanilla-sampling | 3.24 | 4.40 | 2.65 | 2.58 | 1.10 |
| | VCD | 2.82 | 4.10 | 2.44 | 2.39 | 1.20 |
| | OPERA | 3.22 | 4.40 | 2.49 | 2.58 | 1.18 |
| | Woodpecker | 3.26 | 4.42 | 2.46 | 2.64 | 1.22 |
| | LRV | 3.25 | 4.38 | 2.52 | 2.59 | 1.24 |
| | LURE | 3.24 | 4.35 | 2.51 | 2.60 | 1.21 |
| | **VDGD (ours)** | **3.47** | **4.56** | **2.84** | **2.51** | **1.80** |
| LLaVA-1.6 | Vanilla-greedy | 3.43 | 4.41 | 3.13 | 3.31 | 1.35 |
| | Vanilla-sampling | 3.40 | 4.38 | 3.10 | 3.27 | 1.30 |
| | VCD | 3.26 | 4.19 | 2.96 | 3.13 | 1.20 |
| | OPERA | 3.49 | 4.48 | 3.20 | 3.38 | 1.40 |
| | Woodpecker | 3.45 | 4.45 | 3.21 | 3.35 | 1.42 |
| | LRV | 3.42 | 4.40 | 3.18 | 3.34 | 1.39 |
| | LURE | 3.38 | 4.36 | 3.18 | 3.28 | 1.36 |
| | **VDGD (ours)** | **3.64** | **4.71** | **3.49** | **3.42** | **1.45** |
| mPLUG-Owl2 | Vanilla-greedy | 3.23 | 4.42 | 2.86 | 2.48 | 1.25 |
| | Vanilla-sampling | 3.24 | 4.47 | 2.84 | 2.50 | 1.24 |
| | VCD | 3.07 | 4.21 | 2.67 | 2.40 | 1.17 |
| | OPERA | 3.29 | 4.52 | 2.92 | 2.58 | 1.27 |
| | Woodpecker | 3.27 | 4.50 | 2.90 | 2.52 | 1.25 |
| | LRV | 3.23 | 4.49 | 2.88 | 2.50 | 1.24 |
| | LURE | 3.26 | 4.51 | 2.91 | 2.51 | 1.28 |
| | **VDGD (ours)** | **3.34** | **4.64** | **3.15** | **2.70** | **1.32** |
| InternLM-X | Vanilla-greedy | 3.28 | 4.52 | 3.35 | 2.59 | 1.37 |
| | Vanilla-sampling | 3.27 | 4.53 | 3.33 | 2.57 | 1.35 |
| | VCD | 3.08 | 4.24 | 3.10 | 2.44 | 1.30 |
| | OPERA | 3.37 | 4.65 | 3.44 | 2.65 | 1.43 |
| | Woodpecker | 3.38 | 4.63 | 3.45 | 2.66 | 1.44 |
| | LRV | 3.40 | 4.69 | 3.47 | 2.70 | 1.46 |
| | LURE | 3.36 | 4.64 | 3.42 | 2.68 | 1.42 |
| | **VDGD (ours)** | **3.54** | **4.78** | **3.68** | **2.79** | **1.49** |
| CogVLM | Vanilla-greedy | 3.58 | 4.51 | 3.35 | 2.76 | 1.38 |
| | Vanilla-sampling | 3.59 | 4.50 | 3.37 | 2.78 | 1.41 |
| | VCD | 3.36 | 4.32 | 3.21 | 2.58 | 1.30 |
| | OPERA | 3.68 | 4.60 | 3.42 | 2.84 | 1.45 |
| | Woodpecker | 3.70 | 4.58 | 3.45 | 2.85 | 1.42 |
| | LRV | 3.65 | 4.57 | 3.40 | 2.82 | 1.40 |
| | LURE | 3.72 | 4.62 | 3.46 | 2.83 | 1.41 |
| | **VDGD (ours)** | **3.89** | **4.82** | **3.59** | **2.89** | **1.49** |
| GPT-4-Turbo | - | **4.15** | **4.79** | **3.89** | **3.25** | **1.67** |

Table 13: Performance comparison of various LVLMs on Oven benchmark. This is an extension of results shown in Figure 7.

remain constant even during beam search. As a result, VDGD can be directly integrated into beam search decoding to achieve similar improvements in reducing hallucinations. Table 15 shows the result of VDGD with beam search on MMMU dataset.

| Method | Average Inference Time (s) | FLOPs (T) |
|---|---|---|
| Vanilla-greedy | 1.3 | 9.3 |
| VCD | 1.9 | 10.2 |
| VDGD + Small Captioning Model | 2.1 | 11.4 |
| VDGD | 2.4 | 11.9 |

Table 14: Comparison of average inference time and computational cost (in teraFLOPs) among VDGD and baseline methods evaluated on LLaVA 1.5 using a 48GB GPU.

| Benchmark | LLaVA-v1 | LLaVA-1.5 |
|---|---|---|
| Vanilla-greedy | 1.26 | 1.35 |
| Vanilla-sampling | 1.27 | 1.44 |
| VDD | 1.34 | 1.52 |
| OPERA | 1.30 | 1.43 |
| Woodpecker | 1.32 | 1.44 |
| LRV | 1.29 | 1.49 |
| LURE | 1.31 | 1.47 |
| PAI | 1.42 | 1.39 |
| HALC | 1.40 | 1.54 |
| VDGD | 1.42 | 1.62 |
| VDGD + Beam Search | 1.39 | 1.58 |

Table 15: Performance comparison of VDGD w/ beam search for LLaVA-v1 and LLaVA-1.5

## O    FURTHER DISCUSSION ON RELATED WORKS

Recent advances in understanding hallucinations in Large Vision-Language Models (LVLMs) provide critical insights into addressing the persistent challenge of hallucination in multimodal systems. Bai et al. (2024) highlight the growing importance of mitigating hallucinations in LVLMs, particularly in tasks involving complex interactions between visual and textual data. Their comprehensive survey dissects hallucinations into granular categories, such as object attributes and relations, and identifies factors contributing to hallucinations across data, model architecture, and inference stages. Liu et al. (2023c) similarly focus on categorizing and analyzing hallucinations, emphasizing the importance of detailed evaluation metrics and benchmarks tailored to measure both generative and discriminative capabilities of LVLMs. They propose frameworks for systematically assessing hallucination symptoms and mitigating them through refined model alignment and enhanced multimodal representation. These studies underscore the necessity of bridging gaps in current LVLM evaluation methods and mitigation strategies by addressing unique challenges such as data quality, alignment module efficiency, and decoding mechanisms. Our proposed VDGD model aligns with these efforts by introducing a robust framework to reduce hallucinations, leveraging state-of-the-art insights while contributing novel mitigation pathways. This connection to the broader context enriches our understanding and opens pathways for integrating these findings into future iterations of VDGD and related LVLM frameworks.

