# OpenReview forum: "Visual Description Grounding Reduces Hallucinations and Boosts Reasoning in LVLMs"
_ICLR.cc/2025/Conference — ICLR 2025 Poster_

### Official Review · Reviewer_4B1M · 2024-10-23

**Soundness:** 3
**Presentation:** 3
**Contribution:** 3
**Rating:** 8
**Confidence:** 4

**Summary:**

The paper presents a comprehensive and extensive analysis on object hallucination. It proposes VDGD, a method that generates descriptions first, which are then used as prompts for a second inference. During decoding, KLD is calculated with the pre-generated descriptions to identify highly deviant candidates. The authors curate several datasets and introduce the VaLLu benchmark for a comprehensive hallucination evaluation.

**Strengths:**

### S1. The paper is well-structured and comprehensive, providing a smooth overall flow.

### S2. The analysis is in-depth and offers interesting insights.

### S3. The VaLLu benchmark has the potential to serve as a comprehensive benchmark for evaluating hallucinations in models.

**Weaknesses:**

### W1. While the idea is simple and effective, a significant drawback is the latency. The proposed method requires generating long descriptions even for short responses (e.g., in Fig. 8, a single token output would typically be very fast in a baseline method, but the proposed method is much slower). Therefore, it is important to include a latency analysis (e.g., average inference time, throughput) compared to simpler decoding-based hallucination mitigation methods [1,2,3] and baseline LVLMs, especially since many recent training-free methods perform a single inference.
[1] Don’t Miss the Forest for the Trees: Attentional Vision Calibration for Large Vision Language Models, Arxiv 2024.
[2] Seeing is Believing: Mitigating Hallucination in Large Vision Language Models via CLIP-Guided Decoding, Arxiv 2024.
[3] RITUAL: Random Image Transformations as a Universal Anti-hallucination Lever in LVLMs, Arxiv 2024.

### W2. Using the description from the first inference as a prompt for the second inference appears to act like a form of Chain-of-Thought (CoT) prompting. The authors should explicitly discuss how VDGD relates to or differs from Visual CoT methods [4,5,6], and compare several aspects (e.g., performance, computational requirements, applicability to different types of tasks). Comparisons or references to recent Visual CoT methods will strengthen the paper.
[4] Compositional Chain-of-Thought Prompting for Large Multimodal Models, CVPR 2024.
[5] Beyond Embeddings: The Promise of Visual Table in Visual Reasoning, EMNLP 2024.
[6] Visual Evidence Prompting Mitigates Hallucinations in Multimodal Large Language Models, OpenReview.

### W3. While each individual analysis is interesting, the overall flow feels somewhat disjointed. The paper presents a categorization of hallucinations and provides extensive explanations for each category, but there are no corresponding experimental results or analysis showing how the proposed method specifically addresses or improves each of these hallucination categories. This lack of connection between the categorization and the method’s effectiveness on each type of hallucination weakens the coherence of the paper. The authors should include a specific analysis or set of experiments demonstrating how VDGD performs on each category of hallucination they've identified. This would help tie together the theoretical framework and the practical application of their method.

### W4. While the in-depth analysis is appreciated, the paper sometimes feels overloaded with content, which can distract from the core focus. At times, it is difficult to follow, and the connection between earlier sections and the methodology feels weak. The dense content also limits the space for method-related experiments, with only one experiment table included in the main paper. Most experiments have been relegated to the appendix, suggesting the need for better content management.

**Questions:**

### Q1. While this does not affect my score, I believe the terms “perception” and “recognition” should be interchanged in the paper. Perception refers to the basic observation of visuals, while recognition is a more complex process based on what has been perceived. However, the paper appears to use these terms in reverse, which could cause some confusion for readers.

Ref: https://en.wikipedia.org/wiki/Visual_perception

---

> ### Author Response · Authors · 2024-11-21
> **Response to of Official Review of Submission4948 by Reviewer 4B1M (1/3)**
>
> Dear Reviewer 4B1M,
>
> We thank you for your thorough review and constructive feedback. We have tried to address each of your concerns point by point in the rebuttal.
>
> ### Weaknesses
>
> > Weakness 1 (About additional latency)
>
> **Ans.** Thank You for the comment. We would like to make a few points in our support:
>
> - Improving baseline LVLMs typically requires: (i) high-quality datasets with cognitive prompts, which are challenging to collect; (ii) better architectures; and (iii) re-training the LVLM. Our proposed method eliminates the need for any of these and introduces a training-free technique that makes the first attempt at improving reasoning on cognitive prompts in LVLMs.
> - Furthermore, our approach aligns with existing training-free hallucination mitigation techniques (e.g., VCD, LURE, PAI, HALC, etc.) in both motivation and methodology, though the algorithm itself is novel.
> - Most existing methods (including [1,2,3] mentioned by the reviewer) address hallucinations by exploiting the high confidence of language hallucinations in language-only or corrupted image scenarios. While this is effective for mitigating object hallucinations, we demonstrate in Section 3 that these techniques are inadequate for cognitive prompts. Consequently, VDGD employs a distinct mechanism tailored to cognitive prompts, which, while slightly more computationally intensive, is necessary for addressing this unique challenge.
> We would like to show a quick computational analysis:
>
> Computational Analysis on LLaVA 1.5 with 48GB GPU (on MMAU):
>
> | Method | Average Inference Time (s)  | FLOPs (T) | Performance |
> |---|---|---|---|
> | Vanilla-greedy | 1.3 | 9.3 | 1.35 |
> | VCD  | 1.9  | 10.2 | 1.52 |
> | CLIP guided decoding  | 2.6  | 12.6 | 1.45 |
> | RITUAL  | 2.0  | 10.2 | 1.52 |
> | VDGD + Small Captioning Model [1] | 1.8 | 10.4 | 1.58 |
> | VDGD| 2.3 | 11.9  | 1.62 |
>
> As we cam see, VDGD only leads to a slight increase computational complexity but also leads to substantial improvement in performance, especially on cognitive prompts (all other methods are meant for alleviating object hallucinations). We further show that VDGD is competitive with a small captioning model (which is not the LVLM) and the key is to generate an image caption that captures the visual elements. We hope you find this new results insightful. This shows that efficient captioning models also has the potential effectively reduce VDGD complexity.
>
> **We have cited this in the revised version of our paper and added a Discussion in the Appendix N to discuss the differences with these methods.**
>
> > Weakness 2 (About comparison with CoT methods)
>
> **Ans.** Thank You for the comment. We provide a comparison below with the cited methods:
>
> - [4] Compositional Chain-of-Thought Prompting for Large Multimodal Models, CVPR 2024.
> Constructs a scene graph by asking the LVLM to look at the image and the given question and to identify three main properties: objects, object attributes and object relationships. The scene graph is then passed to the LVLM with the query again to get the final response. This method primarily solves compositionality in real world scenes. However the proposed method would be ineffective when the input does not contain real world objects, for example find the peak in a stock price graph.
>
> - [5] Beyond Embeddings: The Promise of Visual Table in Visual Reasoning, EMNLP 2024.
> Proposes a Visual Table, which is trained on GPT4V generated data. The Visual Table generates hierarchical descriptions of visual scenes, featuring a scene description and multiple object centric descriptions covering categories, attributes, and knowledge. The input image is first passed to the Visual Table Generator and then the image query and generated data from VT are passed to the LVLM to get output. This method achieves performance improvement in real-world and non-real world scenes. The disadvantages are this method is not training free and requires significant computational resources.
>
> - [6] Visual Evidence Prompting Mitigates Hallucinations in Multimodal Large Language Models
> Proposes using a small visual model which acts as an object detection model. The model also generates object description, object locations and object relations. The generated data along with input image and user query is passed to the LVLM. This paper suffers from the same disadvantages of CCoT where it is ineffective with non-real world scenes and datasets like MMMU, MathVista etc.
>
> **We have cited this in the revised version of our paper and added a Discussion in the Appendix N to discuss the differences with these methods.**
>
> We also provide a comparison of results below:

---

> ### Author Response · Authors · 2024-11-21
> **Response to of Official Review of Submission4948 by Reviewer 4B1M (2/3)**
>
> **Continued**
> | Methodology                   | Avg Inference Time(s) | FLOPs (T) | Effectiveness in Real-world Scene Datasets (AMBER, MMBENCH)  | Effectiveness in Non real-world scene datasets (MMMU, MathVista, MMVET) | Training Free |
> |-------------------------------|-----------------------|-----------|--------------------------------------------------------------|-------------------------------------------------------------------------|---------------|
> | CCoT                          |                   2.7 |      12.3 | Y                                                            | N                                                                       | Y             |
> | Visual Table                  |                   1.9 |      10.8 | Y                                                            | Y                                                                       | N             |
> | VEP                           |                   1.7 |      10.3 | Y                                                            | N                                                                       | Y             |
> | VDGD + Small Captioning Model |                   2.1 |      11.4 | Y                                                            | Y                                                                       | Y             |
> | VDGD                          |                   2.4 |      11.9 | Y                                                            | Y                                                                       | Y             |
> > Weakness 3 (About the results on the defined categories)
>
> **Ans.** Thank you for your question. We would like to clarify a potential misunderstanding here. The newly defined hallucination categories pertain to image descriptions and not cognitive QAs. These methods cannot be directly applied to categorize hallucinations in cognitive QAs – for example, instruction-response pairs in typical visual instruction tuning datasets do not contain cognitive prompts. Additionally, as shown in Fig. 6, unlike in image description tasks which have a mix of various hallucinations presented in Section 4, hallucinations in cognitive prompts are dominated by language hallucinations are predominant in. **Thus, hallucinations for cognitive prompts cannot be effectively categorized into our mentioned types which is why we do not show results on the same.**
>
> **But why is this analysis required here?** To clarify, our analysis and categorization aim to investigate what is needed to mitigate hallucinations in cognitive prompts, thereby improving reasoning. Section 3 highlights that current mitigation techniques fail to address hallucinations in cognitive prompts. Section 4 categorizes hallucinations and suggests that existing methods, such as logit correction, are inherently tailored to mitigate language hallucinations. Finally, Section 5 demonstrates that (i) for image descriptions, hallucinated tokens are high-confidence single tokens in the logit space and can often be corrected by language-only models with the same prefix; (ii) in cognitive prompts, hallucinations arise from a top-$k$ probability space dominated by low-confidence tokens, making current methods ineffective (lines 70-75).
>
> Building on these findings, VDGD addresses this challenge by grounding response generation to image descriptions, boosting the confidence of correct tokens and mitigating hallucinations more effectively.
>
> > Weakness 4 (About the overall flow and the requirement of the analysis)
>
> **Ans.** Thank you for your observations. We would like to take this opportunity to clarify the flow of information in our paper. Our work is structured to first investigate what is required to effectively mitigate hallucinations for cognitive prompts and thereby improve reasoning.
>
> - **Section 3** explores why current mitigation techniques fail to address hallucinations in cognitive prompts. We conduct experiments using various hallucination mitigation methods, LVLMs, and datasets, demonstrating that existing approaches only reduce hallucinations in image description-based prompts, specifically for natural scenes.
>
> - **Section 4** delves into the reasons behind this limitation. We categorize the types of hallucinations observed and highlight that most existing methods are tailored to address *language hallucinations*. For example, techniques like logit correction target highly confident tokens in language-only or premature model outputs, which may inherently restrict their effectiveness to language hallucinations due to their design choices.
>
> (continued in next part)

---

> > ### Author Response · Authors · 2024-11-21
> > **Response to of Official Review of Submission4948 by Reviewer 4B1M (3/3)**
> >
> > - **Section 5** validates this phenomenon further, showing that: (i) for image description prompts, hallucinated tokens typically appear as single high-confidence tokens in the logit space, which align with outputs of language-only models for the same prefix and can be corrected effectively; (ii) for cognitive prompts, however, hallucinated tokens are distributed across a set of low-confidence tokens with similar probabilities (illustrated in lines X), making them harder to mitigate with current methods.
> >
> > **It was crucial to have all these observations to motivate VDGD.**
> >
> > *Finally, many of our experimental results are presented as figures, with the corresponding quantitative values provided in the Appendix. We have ensured that no primary experiments have been relegated to the Appendix; only ablation studies and supplementary, nice-to-have analyses are included there. We hope the reviewer will consider the structure of our paper following this rebuttal.*

---

> ### Author Response · Authors · 2024-11-23
> **Request to review the rebuttal**
>
> Dear reviewer 4B1M,
>
> Thank you for taking the time to review our paper. We have addressed your concerns in our submitted response and provided a revised version of the paper. As the rebuttal period is nearing its conclusion, we kindly request you to review our rebuttal and share any additional comments or concerns you may have. Thank you once again for your valuable feedback!
>
> Best,
> Authors of Submission4948

---

> ### Comment · Reviewer_4B1M · 2024-11-23
>
> I would like to thank the authors for their detailed response. I have carefully reviewed the rebuttal and appreciate your patience while awaiting my feedback. I am willing to consider raising the score if the following points are addressed:
>
> ## W1.
> I remain unconvinced by the latency analysis presented. Considering that most LVLM benchmarks typically require only a short binary answer (e.g., “Yes” or “No”) or a multiple-choice response (e.g., “A, B, C, D”), the output generation involves merely a single token. While VCD necessitates two forward passes to contrast probability distributions derived from the original and distorted images, it similarly requires generating only a single token for the final response (with the constraint of answering in one word, such as “please answer in one word”).
>
> In contrast, the proposed method appears to have significantly higher latency. It requires pre-generating detailed image descriptions, which can span hundreds of tokens, followed by a prefill phase involving predefined prompts and the generated descriptions before arriving at the final output. This multi-step process is likely to result in considerably longer latency.
>
> To strengthen the argument, I would suggest including a detailed latency breakdown for each step of the pipeline. Additionally, clarification on the “small captioning model” mentioned is needed, as its specifics have not been provided.
>
> ## W2.
> While the authors have stated that the listed methods are cited in the revised version, this does not appear to be the case upon review. Furthermore, I believe that the visual CoT methods discussed are highly relevant to the proposed approach and should be included in the main paper.
>
> Ideally, the proposed method should also be compared with these methods more comprehensively. However, given the constraints of the review timeframe, I understand that a full comparison may not be feasible. At the very least, these related methods should be properly discussed in the main paper to provide necessary context and establish their relevance to the proposed approach.
>
> Thank you once again for your efforts. I look forward to your clarifications on these points.

---

> ### Author Response · Authors · 2024-11-23
> **Response to Official Comment by Reviewer 4B1M**
>
> Dear Reviewer 4B1M,
>
> Thank You for the time in reading our rebuttal. We are glad we could clarify most of your concerns. We would like to take this opportunity to respond to the further weaknesses mentioned by you in your response to our rebuttal.
>
> > W1 (About breakdown of latency)
>
> **Ans.** Thank You for your question. We would like to present a 3-way comparison of latency on MMMU and VaLLu for LLaVa-1.5. Before we go ahead with our analysis, we request you to note two important points:
>
> - For the input prompt, beyond the length of the text prompt, **the vision encoder adds 576 tokens to the total context length** (in case of LLaVa 1.5 which employs CLIP ViT-L/336px)
> - We request you to note that VDGD is comprised on all open-ended generation responses. MMMU on the other hand has 689 open-ended generation questions and not all single word answers. Finally, most models do not respond in just one word and sometimes add additional context to the response (e.g., Yes, ....)
> - We also would like to re-iterate that a majority of results and analysis in our paper are focused towards evaluating and improving open-ended generations in LVLMs as they more closely reflect ideal real-world interactions with LVLMs.
>
>
> **Vanilla Greedy Sampling**
>
> - MMMU: Single Forward Pass: Time: ~1.3 seconds TFlops: 9.3
> - VaLLu: Single Forward Pass: Time: ~1.6 seconds TFlops: 11.0
>
> **VCD** (VCD analysis also reflects a majority of the logit correction methods proposed in literature)
>
> - MMMU: Two consecutive forward passes for each token (assuming both copies are in the same GPU): Time: ~1.9 seconds TFlops: 14.6 for LVLM w/ image and 5.4 for LVLM w/o image (average ~10.2)
>
> - VaLLu: Two consecutive forward passes for each token (assuming both copies are in the same GPU): Time: ~2.5 seconds TFlops: 18.9 for LVLM w/ image and 8.9 for LVLM w/o image (average ~13.6)
>
> **Please note that as output length increases in real-world open-ended generations, VCD-like logit correction provides to be much more expensive than VDGD as the extended context length for each new token requires two passes now**
>
> **VDGD**
>
> - MMMU: First Forward Pass for Caption: Time: ~1 seconds TFlops: 9.3 (same as above greedy as image tokens take most processing and response tokens contribute minimally)
> - MMMU: Second Forward Pass for Response Generation Time: ~1.4 seconds TFlops: 14.5
> - VaLLu: First Forward Pass for Caption: Time: ~1 seconds TFlops: 9.3
> - VaLLu: Second Forward Pass for Response Generation Time: Time: ~1.4 seconds TFlops: 14.5
>
> **Please note that for VDGD, our implementation (codebase in Reproducibility section of our paper) saves model logits during caption generation which is used directly by the model for response generation. Adding it to context during generation did not lead to score change.**
>
> **VDGD with small captioning model**
>
> - MMMU: First Forward Pass for Caption: Time: ~0.4 seconds TFlops: 6.3
> - MMMU: Second Forward Pass for Response Generation Time: ~1.4 seconds TFlops: 14.5
> - VaLLu: First Forward Pass for Caption: Time: ~0.4 seconds TFlops: 6.3
> - VaLLu: Second Forward Pass for Response Generation Time: Time: ~1.4 seconds TFlops: 14.5
>
> Thus, we would like to conclude with 3 points:
> - We hypothesize that the input token processing is responsible for a majority of the compute. This has also been discussed in prior art. In comparison, the captions generated are around ~30-50 tokens (only <10% additional tokens)
> - In real-world cases with open-ended generations, VDGD is competitive in terms of compute taken for responding with other hallucination mitigation techniques proposed in literature.
> - VDGD can be implemented in a variety of methods. For example, a small captioning model, or the logits can be saved prior to generation (which was done in our case for benchmark result generation)
> - **We also acknowledge in our rebuttal that VDGD iadds computational overhead. However, we expect the overhead to keep getting lower with advancements in captioning models.**
>
> Finally, for the small model, we employ: SmallCap: Lightweight Image Captioning Prompted with Retrieval Augmentation (CVPR 2023)
>
> Thank You again and we hope we have responded to your queries!
>
> > W2 (About discussion with visual CoT methods)
>
> **Ans.** We have just updated our paper with these addition to the main paper! We have cited your mentioned works and added a short discussion in lines 101-107 of the revised version of our paper.
>
> Thank You again for your time. Your feedback is invaluable to us in improving the quality of our paper.

---

> ### Comment · Reviewer_4B1M · 2024-11-24
>
> Thank you for your prompt response.
>
> # W1.
> I am still not fully convinced by the latency analysis provided. While I understand that most models are not inherently designed to respond in a single word (e.g., “Yes” or “No”), it is possible to constrain their outputs (using "please answer in a single word") to such a format when the task allows for binary or multiple-choice responses. This can be achieved through careful prompting to ensure concise outputs in scenarios where single-word answers are appropriate.
>
> [VCD]
> - forward pass with original image: (# image tokens = 576 + # prompt length = 30~50) -> # output length = $N$
> - forward pass with distorted image: (# image tokens = 576 + # prompt length = 30~50) -> # output length = $N$
>
> [VDGD]
> - first forward pass: (# image tokens = 576 + # prompt length = 30~50) -> # output length = $M$ (description length)
> - second forward pass: (# image tokens = 576 + #  prompt length = $M$ + 30~50) -> # output length = $N$
>
> In the case of constrained decoding, $N = 1$ for VCD (e.g., binary or multiple-choice questions). While I recognize that this may not always apply to open-ended or detailed questions, VCD remains more efficient in its average-case scenario due to the shorter input/output requirement. In contrast, VDGD involves generating longer descriptions, with $M$ often far exceeding $N$. Additionally, VDGD incurs the extra cost of prefilling $M$ additional tokens during the second forward pass, which adds to the overall latency.
>
> I am particularly curious about how the latency of the first forward pass (caption generation) for VDGD compares to that of the second forward pass (response generation). This discrepancy likely depends on the configuration of `max_token_length` and warrants further clarification.
>
> As a result, I find the following statement from the rebuttal unconvincing:
> `“VCD-like logit correction proves to be much more expensive than VDGD as the extended context length for each new token requires two passes now.”`
>
> Also, the “small captioning model” referenced in the paper should be properly cited.

---

> > ### Author Response · Authors · 2024-11-24
> > **Reply to Official Comment by Reviewer 4B1M**
> >
> > Thank You for the time in reading our rebuttal.
> >
> > > Also, the “small captioning model” referenced in the paper should be properly cited.
> >
> > **Ans.** We have updated the paper with this citation.
> >
> > > As a result, I find the following statement from the rebuttal unconvincing: “VCD-like logit correction proves to be much more expensive than VDGD as the extended context length for each new token requires two passes now.”
> >
> > **Ans.** We apologize for any misunderstanding here. Our statement was made for a non-benchmark real-world condition case. Let us assume in a typical case non-benchmark and real-world use case the average response output length $N$ =1000 tokens.
> >
> > Thus, in this case, the FLOPs (or compute) taken by VCD is higher (we acknowledge that for one-word benchmark cases VCD always proves to be more efficient -- **this why our analysis and experiments in our paper also focus on free-form open-ended generation where context actually plays crucial information, e.g., step-by-step thinking):
> >
> > [VCD]
> >
> > - forward pass with original image: (# image tokens = 576 + # prompt length = 30$\approx$50) --> # output length = $N$
> > - forward pass with distorted image: (# image tokens = 576 + # prompt length = 30$\approx$50) --> # output length = $N$
> >
> > [VDGD]
> >
> > - first forward pass: (# image tokens = 576 + # prompt length = 30$\approx$50) -> # output length = $M$ (description length)
> > - second forward pass: (# image tokens = 576 + # prompt length = $M$ + 30$\approx$50) -> # output length = $N$
> >
> > This is why in our previous rebuttal message we say **as the extended context length** -- by which we mean as the context length increases beyond a certain point (non-benchmark cases).
> >
> > > I am particularly curious about how the latency of the first forward pass (caption generation) for VDGD compares to that of the second forward pass (response generation).
> >
> > **Ans.** Captions generated by captioning models are ideally 30-50 tokens and do not come close to `max_token_length`. In most real-world cases, the number of caption tokens is almost always smaller than the number of response tokens (also for more recent step-by-step thinking methods like o1) . For standard benchmarks, the number of caption tokens is almost always larger than response tokens. Additionally, as mentioned earlier, small captioning models can ease the caption generation latency.
> >
> > **We would like to note that we have already acknowledged the computational requirements in the Limitations section of our paper. However, we believe the significant performance improvements achieved, combined with the training-free nature of VDGD, provide a strong justification for the increased inference-time computation.**

---

> ### Comment · Reviewer_4B1M · 2024-11-24
>
> Thank you for the detailed response throughout the discussion.
> My concerns are mostly resolved.
>
> I am increasing my score to 8.

---

> > ### Author Response · Authors · 2024-11-25
> > **Thank You!**
> >
> > Thank You for you response and increasing our score. We really appreciate it!
> >
> > Best,
> > Authors of Submission4948

---

### Official Review · Reviewer_cozw · 2024-11-02

**Soundness:** 2
**Presentation:** 3
**Contribution:** 2
**Rating:** 6
**Confidence:** 4

**Summary:**

This paper addresses the problem of hallucinations in LVLMs. Strictly speaking, the authors provide a way to understand the root cause of such hallucinations in LVLMs. They claim that existing approaches for hallucination mitigation only focus on the visual recognition aspect of the model, and do not dive further into understanding whether the model actually has cognitive skills, thus failing to mitigate hallucinations properly from such models. The authors first conduct a study to investigate the various causes of hallucinations in LVLMs. Then they introduce VDGD, a training-free method to mitigate said hallucinations from an LVLM, and finally propose VaLLu, a benchmark to evaluate cognitive reasoning capabilities of LVLMs.

**Strengths:**

1) The paper is well written and nicely presented.
2) The authors present a classification of different types of halluciantions
3) The authors recognize the gap between visual recognition capability of a model and ability to utilize it for cognitive tasks
4) The authors propose a training-free strategy to mitigate hallucinations

**Weaknesses:**

1) Sections 3.1 and 3,2 are not that informative about the failure of hallucination techniques on cognitive tasks. The claim in 3.1 that all methods boost performance on AMBER but not on other datasets is weak since even on AMBER, the relative performance increase is small, which is the same for other datasets as well. This goes against the claims of these two sections.

2) In section 3.3, in the algorithm, firstly, currently it says that it is a language hallucination if base rank is less than 0. How is a rank less than zero? I think it should be language hallucination if it does not fall inside the visual elements of the response. Moreover, I think you are using GPT-4 vision, and not just GPT-4 for this? Also, the visual content extraction itself will have hallucinations due to use of llama-3. Further, pushing everything first to IT hallucination definitely skews the outputs of visual and style hallucinations, it can be both. And so the results showing huge IT hallucination compared to the other two is misleading.

3) Experiment on 4.1 showing fall of rank as length of text prompt increases is nice, but it is also bound to happen since the textual context is getting added. This does not definitively prove that no image context is being attended to. Also, the rank difference between the two datasets is just 1.

4) The gpt-type metric the authors propose is claimed to have high correlation with the human responses. But in appendix we see that the other correlations are also quite high, with a normal benchmarks having a 0.92 correlation compared to author's 0.96. This marginal difference is not significant enough to claim for a new type of evaluation protocol.

**Questions:**

1. In Sections 3.1 and 3.2, you mention that existing hallucination mitigation techniques improve performance on AMBER but not on other datasets. Could you provide quantitative results for this to show this?

2. In Section 3.3, the algorithm indicates that a base rank less than zero signifies a language hallucination. Since ranks are typically non-negative, could you explain how a rank can be less than zero in this context?

3. The algorithm seems to classify instances as information transfer (IT) hallucinations first, which might influence the distribution of hallucination types. How do you ensure that this approach doesn't skew the results, particularly the higher incidence of IT hallucinations compared to visual and style hallucinations?

4. In Section 4.1, you observe a decline in rank as the text prompt lengthens, suggesting reduced attention to image context. Given that longer prompts naturally introduce more textual context, how do you differentiate between the model's reliance on textual versus visual information in this scenario?

5. Your proposed GPT-based evaluation metric shows a correlation of 0.96 with human responses, while existing benchmarks have a correlation of 0.92. Considering this marginal difference, what advantages does your metric offer over traditional evaluation protocols?

---

> ### Author Response · Authors · 2024-11-21
> **Response to Official Review of Submission4948 by Reviewer cozw (1/2)**
>
> We thank you for your thorough review and constructive feedback. We have tried to address each of your concerns point by point in the rebuttal.
>
> ### Questions
>
> > Question 1 (about quantitive scores)
>
> **Ans.** Thank You for your question. We provide the scores below:
>
> | Model                | Score |
> |----------------------|-------|
> | LLaVA-1.5 (vanilla)  | 3.91  |
> | VCD                 | 4.16  |
> | OPERA               | 4.09  |
> | Woodpecker          | 4.21  |
> | LURE                | 4.06  |
> Caption: Scores for Amber.
>
> | Model                | Score |
> |----------------------|-------|
> | LLaVA-1.5 (vanilla)  | 2.00  |
> | VCD                 | 1.79  |
> | OPERA               | 1.71  |
> | Woodpecker          | 1.93  |
> Caption: Scores for MMMU.
>
> | Model                | Score |
> |----------------------|-------|
> | LURE                | 1.87  |
> | LLaVA-1.5 (vanilla)  | 2.23  |
> | VCD                 | 2.21  |
> | OPERA               | 2.27  |
> | Woodpecker          | 2.19  |
> | LURE                | 2.25  |
> Caption: Scores for MathVista.
>
> | Model                | Score |
> |----------------------|-------|
> | LLaVA-1.5 (vanilla)  | 2.21  |
> | VCD                 | 2.15  |
> | OPERA               | 2.15  |
> | Woodpecker          | 2.13  |
> | LURE                | 2.21  |
> Caption: Scores for MMMU.
>
>
> | Model                | Score |
> |----------------------|-------|
> | LLaVA-1.5 (vanilla)  | 1.17  |
> | VCD                 | 1.11  |
> | OPERA               | 1.23  |
> | Woodpecker          | 1.19  |
> | LURE                | 1.27  |
> Caption: Scores for Donut.
>
> > Question 2 (about  base rank in language hallucination)
>
> **Ans.** Thank you for the question. We would like to clarify a potential misunderstanding. As you correctly noted, base ranks (BR) are non-negative. In Section 3.3, Algorithm 1 specifies that a BR = 0 signifies a language hallucination, not a BR < 0.
>
> To clarify further: if BR > 0, it is not a language hallucination; otherwise (i.e., BR = 0), it is classified as a language hallucination. There is no case where BR < 0 in this context.
>
> > Question 3 (About the classification of IT instances)
>
> **Ans.** Thank you for the question. We would like to address your concern by highlighting three key points that demonstrate the robustness of our method:
>
> - IT hallucinations originate explicitly from the information transfer (IT) stage and can be directly matched with entries in the IT dataset. As noted in lines Section 3.3 of our paper, this makes IT hallucinations the most explicit and identifiable type, with strong evidence for their cause.
>
> - Appendix K.5 provides ablations for various values of *k*, showing that while the count of IT hallucinations may vary slightly with *k*, the counts of other hallucination types remain stable. This demonstrates that categorizing IT hallucinations first does not impact the classification of visual or style hallucinations.
>
> - The consistent count of IT hallucinations across different  *k* values reinforces that IT hallucinations are not misclassified or inflated due to other causes. By categorizing them first, we ensure accurate and unbiased classification of other hallucination types.
>
> > Question 4 (Section 4.1 length of text prompts)
>
> **Ans.** Thank you for the question. First, we would like to clarify a potential misunderstanding: the X-axis in Figure 6 represents the token position in the response, not the prompt length (as stated in the caption and lines 300-309). The figure shows a sharp decline in Base Rank after the second token, after which the curve flattens, with an average Base Rank between 0 and 1 throughout the rest of the response. Since the first two tokens are likely style tokens (e.g., This, The, etc [1]), they contain minimal content for the model to leverage. This sharp drop and sustained low Base Rank demonstrate that the model predominantly relies on language priors rather than attending to the image, a phenomenon we term the alignment gap.
>
> While longer prompts naturally introduce more textual context, the model is not expected to rely heavily on earlier response tokens during auto-regressive generation. For example, in tasks like image description generation (e.g., AMBER), a model should ground its responses primarily in the image context rather than earlier response tokens. The observed pattern further highlights the misalignment between visual grounding and the model’s reliance on language priors, which we address in our work.

---

> > ### Author Response · Authors · 2024-11-21
> > **Response to Official Review of Submission4948 by Reviewer cozw (2/2)**
> >
> > > Question 5 (About GPT metric)
> >
> > **Ans.** Thank You for the question. Our motivation is similar to [1,2] and several other works from literature. We mention them as follows:
> >
> > - Fine-Grained Evaluation: Unlike existing benchmarks that often rely on coarse-grained or binary scoring, our metric provides a detailed and mutli-aspect assessment of responses by capturing subtle differences in factual correctness, reasoning, and alignment with the prompt. This leads to a more precise evaluation of LVLM capabilities.
> > - Consistency Across Benchmarks: Traditional evaluation protocols often employ benchmark-specific metrics that vary in robustness and focus. Our unified GPT-based evaluation ensures consistency and comparability across diverse benchmarks, reducing variability introduced by differing metrics.
> > - Penalization of Hallucinations: Our evaluation prompt explicitly penalizes hallucinated responses, a critical issue in LVLMs that traditional metrics may overlook that only judge correctness but do not penalize hallucinations. This provides a more accurate assessment of the model’s ability to generate factual and reliable outputs.
> > - Better suited for open-ended generations: A wealth of analysis in our paper is made on open-ended generations from LVLMs -- which is a more ideal an real-world case (than MCQs) of interacting with humans. Prior benchmarks adopt traditional string matching due to the presence of MCQs. Automated LLM-as-a-judge evaluation provides better evaluation for open-ended generations [1].
> >
> > **The high correlation with traditional metrics shows that our proposed metric rewards the model correctly when it responds with accurate responses. However, as mentioned earlier, beyond this our metric also penalizes on hallucinations and provides other benefits.**
> >
> > ### References
> > [1] Ghosh, Sreyan, et al. "A Closer Look at the Limitations of Instruction Tuning." arXiv preprint arXiv:2402.05119 (2024).
> > [2] Yu, Weihao, et al. "Mm-vet: Evaluating large multimodal models for integrated capabilities." arXiv preprint arXiv:2308.02490 (2023).

---

> > ### Comment · Reviewer_cozw · 2024-11-25
> >
> > Thank you for the rebuttal. Most of my concerns have been addressed. However, currently my main issue is:
> >
> > > **Question 1:**
> >
> > The scores for both Amber and DONUT increase in a similar manner. This is why I believe the claim that the improvement is significant only for Amber may not be entirely accurate. Considering this, I feel such improvement may not serve as a strong basis for motivation.

---

> ### Author Response · Authors · 2024-11-23
> **Request to review the rebuttal**
>
> Dear reviewer cozw,
>
> Thank you for taking the time to review our paper. We have addressed your concerns in our submitted response and provided a revised version of the paper. As the rebuttal period is nearing its conclusion, we kindly request you to review our rebuttal and share any additional comments or concerns you may have. Thank you once again for your valuable feedback!
>
> Best,
> Authors of Submission4948

---

> > ### Author Response · Authors · 2024-11-24
> > **Request to review the rebuttal**
> >
> > Dear reviewer cozw,
> >
> > Thank you for taking the time to review our paper. We have addressed your concerns in our submitted response and provided a revised version of the paper. As the rebuttal period is nearing its conclusion, we kindly request you to review our rebuttal and share any additional comments or concerns you may have. Thank you once again for your valuable feedback!
> >
> > Best,
> > Authors of Submission4948

---

> > > ### Author Response · Authors · 2024-11-25
> > > **Request to review the rebuttal**
> > >
> > > Dear reviewer cozw,
> > >
> > > Thank you for taking the time to review our paper. We have addressed your concerns in our submitted response and provided a revised version of the paper. As the rebuttal period is nearing its conclusion, we kindly request you to review our rebuttal and share any additional comments or concerns you may have. Thank you once again for your valuable feedback!
> > >
> > > Best,
> > > Authors of Submission4948

---

> ### Author Response · Authors · 2024-11-25
> **Response to Official Comment by Reviewer cozw**
>
> Thank You for reviewing our response and the rebuttal. Your feedback is very important for improving the quality of our paper.
>
> We would like to highlight a possible overlook: For various methods, AMBER has much higher performance improvement than DONUT in the Tables provided in our rebuttal:
>
> - For Woodpecker, AMBER has an improvement of +0.30 , but DONUT only has +0.02
> - For Opera, AMBER has an improvement of +0.18 , but DONUT only has +0.05
> - For VCD, AMBER has an improvement of +0.25, but DONUT has -0.06 (performance decrease)
> - For LURE, AMBER has an improvement of +0.15, but on DONUT has +0.10. We acknowledge that the gains are almost similar **only in LURE** (AMBER still has +0.05 improvement higher than DONUT). We hypothesize that LURE is a fine-tuning method and not a *training-free* method.
>
> The relative gap in gains are substantial and is correlated with all other results presented in our paper.
>
> --------
> **Additional Thoughts**.
>
> All results are averaged across 3 runs.
>
> Additionally, both benchmarks are visual element recognition benchmarks -- while AMBER is visual element recognition for natural scenes, DONUT is visual element recognition for document understanding (or OCR). **Scores on DONUT are also overall lower than AMBER.** None of them are cognitive QA benchmarks, which is the main focus of our analysis from Section 3.1. We would like to highlight two additional points:
>
> - Our main motivation for this analysis is that current mitigation techniques are algorithmically built for mitigating only visual element hallucinations in natural scenes. Our hypothesis is detailed in lines 270 - 278 of our paper. An intriguing example is Woodpecker which depends on object recognition and grounding (not applicable in real-world scenes) and VCD where logit correction works due to extended descriptions with high confidence generated by the LVLM.
>
> - Our motivation of VDGD is not grounded to this analysis. VDGD is motivated by the fact that *current hallucination mitigation techniques do not perform well on mitigating hallucinations in cognitive prompts*. This analysis can be seen in Section 3.1 where truly none of the hallucination mitigation techniques improve on cognitive QA benchmarks. Our analysis in Section 3.2 is just to further strengthen our analysis in Section 3.1 and investigate the cause for lower performance on cognitive QA benchmarks which ideally do not have real-world scenes.
>
> ------
> We respectfully hope you can consider our response and we are happy we were able to resolve your other queries.
>
>
> Best,
> Authors of Submission4948

---

> > ### Author Response · Authors · 2024-11-26
> > **Request to review to the response**
> >
> > Dear Reviewer cozw,
> >
> > Thank You for your time in reviewing our paper and the rebuttal. Your feedback in invaluable to us in improving the quality of our paper.
> >
> > In response to your last comment on the performance of AMBER vs DONUT in Section 3.1, we have highlighted a potential misunderstanding or overlook in reply to that comment. We respectfully request you to to please review our response and let us know if your concern has been addressed. We are also more than happy to address any other concern you have!
> >
> > Best,
> > Authors of Submission4948

---

> > ### Comment · Reviewer_cozw · 2024-11-27
> >
> > Most of my concerns have been addressed. I will upgrade my score to 6.

---

### Official Review · Reviewer_fYgL · 2024-11-03

**Soundness:** 3
**Presentation:** 4
**Contribution:** 3
**Rating:** 6
**Confidence:** 4

**Summary:**

This paper focuses on the cognitive reasoning aspect of hallucination in large vision language models. Through a set of analysis and experiments, it demonstrates that the core blocker of this issue is the difficulty of linking recognized visual concepts to the internal knowledge of LLM. Therefore, the paper further proposes a simple method that per-appendes the image description to the text instruction as the full instruction so that the model can better leverage its reasoning capacity. Evaluation shows that this method can achieve consistent performance improvement on reasoning-focused benchmarks.

**Strengths:**

1. The problem of cognitive reasoning in hallucination is interesting and seems to be under-explored in previous works.
2. The analysis is sufficient, solid, and easy to follow, yielding several interesting insights.
3. The proposed method, although appears to be very simple, is based on the analysis on "language grounding" in previous sections, which has a reasonable motivation of such design.
4. The method demonstrates consistent improvements across benchmarks.

**Weaknesses:**

1. This method seems to be limited in science domain, e.g., chart understanding and reasoning. The underlying assumption of the method is that the image can be sufficiently described by texts. It might hold true for science images, e.g., one can easily describe a chart by enumerating all the involved data or simply transforming the chart figure into a table. However, for natural scenes with complex object categories, attributes, and relations, it is almost impossible to fully represent the image with texts. The evaluated benchmarks seems to be focused on such kind of data.
2. Based on my first point, I may suspect that the essential reason of the performance improvement comes from that chart figure is more intuitive for human eyes while text descriptions of data is more suitable for LLM to understand. It may has little relation with **cognitive reasoning**.
3. Also, based on my first point, we'd better not simply regard such science data as reasoning, there can be other forms of reasoning in natural scenes according to some related works [1].
4. The analysis, though informative, takes too much space, and it may have overlap with previous works [2]. For example, categorization of hallucination types in this paper is essentially based on the **cause** of hallucination, which has been discussed in previous works.
5. Moreover, the the experiments and investigation of the proposed method seems to be limited. It is better to involve more ablation studies.
6. The related works is somehow limited. I understand it might be constrained by the space, but it's important to review and discuss related works about hallucination, reasoning, benchmarks, and so on [2] [3].

I will put my initial score as 5 and I hope the authors can resolve my concerns.

[1] Lai et. al. LISA: Reasoning Segmentation via Large Language Model

[2] Bai et. al. Hallucination of Multimodal Large Language Models: A Survey

[3] Liu et. al. A Survey on Hallucination in Large Vision-Language Models

**Questions:**

Please see weaknesses part.

---

> ### Author Response · Authors · 2024-11-21
> **Response to Official Review of Submission4948 by Reviewer fYgL (1/2)**
>
> We thank you for your thorough review and constructive feedback. We have tried to address each of your concerns point by point in the rebuttal.
>
> # Weaknesses
>
> > Weakness 1 (About reasoning on data with natural scenes)
>
> **Ans.** Thank you for the insightful comment. We acknowledge that VDGD assumes visual content can be sufficiently represented in text. While this may seem more suited for structured scientific images like charts and tables, our evaluation in Table 2 also demonstrates that VDGD is effective across a range of non-scientific image benchmarks, such as MMVet, LLaVA-Bench, and Oven. VDGD leverages the LVLM's ability to identify key visual elements—such as objects, attributes, and their relationships—which ensures accurate descriptions. This capability allows VDGD to generalize effectively, reducing hallucinations in both structured scientific images and complex, unstructured natural images.
>
> **Additionally, recent advanced and foundational models (e.g., Qwen 2 Vision) have shown superior capabilities in capturing details in natural scene images. Finally, we as image captioning methods keep improving and their capabilities to capture every minute visual element improves, we hypothesize that the performance of VDGD will keep improving.**
>
> > Weakness 2 (About definition of cognitive reasoning)
>
> **Ans.** Thank you for the comment. We would like to clarify a potential misunderstanding. We would first to reiterate the VDGD process:
> - As discussed in Section 4 of our paper, the *top-k* space of hallucinated tokens in responses to cognitive prompts (those requiring reasoning and knowledge) is dominated by low-confidence and equally likely tokens, a result of the alignment gap (lines 340-342). This finding is unique to cognitive prompts, and VDGD is specifically designed to address this issue.
> - By grounding responses to image descriptions, VDGD increases the confidence of the correct token. This improvement is crucial during auto-regressive generation, as it prevents hallucination snowballing and enables accurate responses. VDGD operates on a simple yet intuitive principle, similar to how humans write down observations from an image to guide reasoning tasks.
>
> **To summarize**, VDGD does not merely prepend descriptions to improve reasoning. Instead, it leverages grounding to boost token confidence, ensuring more accurate responses. Finally, as by VDGD’s performance on benchmarks like MMVet [4], LLaVa-Bench [5] and Oven [6] which involves reasoning beyond simple data representation.
>
> We also state that" *VDGD can be seen as analogous to how humans think – “Much like humans, who often write down their observations of an image in their own words and refer back to them while tackling complex reasoning tasks*.
>
>
> > Weakness 3 (About other forms of reasoning)
>
> **Ans.** Thank you for the comment. Our work focuses broadly on cognitive prompts, which require reasoning or knowledge to generate responses, and not on specific types of reasoning—this term has been consistently used throughout our paper. Benchmarks such as MMVet, LLaVA-Bench, and Oven include non-scientific instances that include cognitive prompts (or require reasoning). As stated earlier, if the description captures the details of the image, VDGD effectively guides generation to reduce hallucinations. Additionally, benchmarks like LISA [7], which involve segmentation tasks, and instances in the mentioned benchmarks demonstrate reasoning beyond scientific contexts.
>
> > Weakness 4 (Comparison with [2] on hallucination categories)
>
> **Ans.** Thank you for the question! Upon reviewing [2] in detail, we find that while [2] categorizes hallucinations by type (e.g., object, attribute, and relation hallucinations, as referenced in lines 237-266 of our paper) and attributes their causes to broader factors like data, architecture, or connection modules, it does not categorize hallucinations based on their decoding-time origin or analyze and link their causes to behaviors in the logit space—a key focus of our work.
> Our contribution lies in proposing a novel approach that first categorizes hallucinations by type and then by their specific cause, including decoding-time factors and logit-space dynamics. This perspective is critical for understanding and mitigating hallucinations, as demonstrated in our analysis and findings.
>
> > Weakness 5 (About more ablations)
>
> **Ans.** Thank you for your comment. The Appendix of our paper contains extensive ablation studies. For instance, Table 3 presents key ablations for VDGD, while Tables 5–9 include additional experiments:
> - Scores of LVLMs on rephrased prompts without images (Appendix L.1)
> - Performance on the VaLLu benchmark when only image descriptions are provided (Table 3)

---

> > ### Author Response · Authors · 2024-11-21
> > **Response to Official Review of Submission4948 by Reviewer fYgL (2/2)**
> >
> > **continued**
> >
> > - Evaluations on popular image description benchmarks (Appendix L.3)
> > - Capability-specific analysis on VaLLu using the "Factuality" metric (Appendix L.2)
> > - Post-truncation probability statistics across models using the elbow method. These studies provide a comprehensive investigation of VDGD’s effectiveness (Appendix L.5)
> >
> > If you have any more suggestions, please let us know and we would be happy to compare and provide more ablation studies.
> >
> > - Weakness 6 (About more related work)
> >
> > **Ans.** Thank You for your suggestion. We have cited these papers in the revised version of our paper and added discussion in Appendix O.
> >
> > ### References
> >
> > [1] Lai et. al. LISA: Reasoning Segmentation via Large Language Model.
> > [2] Bai et. al. Hallucination of Multimodal Large Language Models: A Survey.
> > [3] Liu et. al. A Survey on Hallucination in Large Vision-Language Models.
> > [4] Yu, Weihao, et al. "Mm-vet: Evaluating large multimodal models for integrated capabilities." arXiv preprint arXiv:2308.02490 (2023).
> > [5] Liu, Haotian, et al. "Visual instruction tuning." Advances in neural information processing systems 36 (2024).
> > [6] Hu, Hexiang, et al. "Open-domain visual entity recognition: Towards recognizing millions of wikipedia entities." Proceedings of the IEEE/CVF International Conference on Computer Vision. 2023.
> > [7] Lai et. al. LISA: Reasoning Segmentation via Large Language Model.

---

> ### Author Response · Authors · 2024-11-23
> **Request to review the rebuttal**
>
> Dear reviewer fYgL,
>
> Thank you for taking the time to review our paper. We have addressed your concerns in our submitted response and provided a revised version of the paper. As the rebuttal period is nearing its conclusion, we kindly request you to review our rebuttal and share any additional comments or concerns you may have. Thank you once again for your valuable feedback!
>
> Best,
> Authors of Submission4948

---

> > ### Author Response · Authors · 2024-11-24
> > **Request to review the rebuttal**
> >
> > Dear reviewer fYgL,
> >
> > Thank you for taking the time to review our paper. We have addressed your concerns in our submitted response and provided a revised version of the paper. As the rebuttal period is nearing its conclusion, we kindly request you to review our rebuttal and share any additional comments or concerns you may have. Thank you once again for your valuable feedback!
> >
> > Best,
> > Authors of Submission4948

---

> > > ### Comment · Reviewer_fYgL · 2024-11-24
> > >
> > > I thank the authors for the detailed response and appreciate your patience. After reading the rebuttal, I would like to raise my score to 6.
> > >
> > > However, there are still some concerns regarding the involvement of captions that I hope the authors can address in the revised paper.
> > >
> > > 1. As the authors mentioned, strong foundation models can provide more reliable captions. This is kind like the method is transferring the capability of a strong model to a weak model. The authors should explicitly discuss this limitation and the convoluted relations between the actual LVLM and the captioning model.
> > > 2. It is better to quantitatively investigate the effect of the captions. E.g., using ground truth captions if applicable or comparing captions generated by different models. This can help understand the role and potential limitation of using captions.

---

> ### Author Response · Authors · 2024-11-24
> **Response to Official Comment by Reviewer fYgL**
>
> Dear Reviewer fYgL,
>
> Thank You for your time in reviewing and responding to our rebuttal. We are grateful and we are confident your comments and suggestions can help us improve the quality of our paper.
>
> At your request, we have made 2 major additions and uploaded a revised version of our paper:
>
> 1. We have added evaluation results on the VaLLu benchmark for (i) VDGD + GPT-4 captions (a stronger captioning model) and (ii) VDGD + [1] as a captioning model (a smaller but more compute efficient captioning model). We present the results below and have added these rows to Table 3 in the Appendix with other ablations:
>
> | Benchmark     | LLaVA-v1 | LLaVA-1.5 | LLaVA-1.6 | mPLUG-Owl2 | InternLM-X | CogVLM |
> |---------------|----------|-----------|-----------|------------|------------|--------|
> | **VDGD** *(ours)*                        | 2.16     | 2.64     | 3.16     | 2.72     | 3.45     | 3.01     |
> | **VDGD (+) GPT-4 Captions**              | **2.31** ± 0.04 | **2.91** ± 0.6  | **3.37** ± 0.02 | **2.97** ± 0.05 | **3.65** ± 0.02 | **3.44** ± 0.06 |
> | **VDGD (+) SmallCap Captions**           | 2.06 ± 0.06 | 2.38 ± 0.08 | 3.00 ± 0.03 | 2.43 ± 0.09 | 3.23 ± 0.08 | 2.95 ± 0.04 |
> | **VDGD (-) VCD**                         | 2.08 ± 0.09 | 2.43 ± 0.15 | 3.01 ± 0.08 | 2.54 ± 0.07 | 3.26 ± 0.12 | 2.95 ± 0.06 |
>
>
> We also show VDGD (-) VCD results  for comparison (VDGD without VCD-based decoding for caption generation. As we can see,
> VDGD shows notable performance gains with GPT-4 captions, highlighting the impact of high-quality captions. It also performs competitively with smaller captioning models (SmallCap -- https://arxiv.org/abs/2209.15323), suggesting future improvements in small and better captioning models can further enhance VDGD’s effectiveness and efficiency.
>
> **The results were accumulated as efforts to your questions and a similar question by Reviewer 4B1M.**
>
> 2. We have explanation to two places in our paper for the same: (i) lines 509 - 511 in the main paper and (ii) lines 816 - 822 in the Appendix where we have thoroughly explained this phenomena.
>
> We thank you again and request you to please let us know if we can clarify anymore of your concerns. We appreciate your willingness to raise our score and we look forward to it!
>
> Best,
> Authors of Submission4948

---

> > ### Comment · Reviewer_fYgL · 2024-11-25
> >
> > Thank the authors for the response. I have changed my score to 6.
> >
> > The new experiment demonstrated that GPT-4 can further improve the performance, while SmallCap can worsen the performance. It suggests the method has a strong dependency on the caption quality, which should be discussed in the paper.

---

> > > ### Author Response · Authors · 2024-11-25
> > > **Thank You!**
> > >
> > > Thank You for you response and increasing our score. We really appreciate it!
> > >
> > > We have added the explanation to two places in our paper for the same: (i) lines 509 - 511 in the main paper and (ii) lines 816 - 822 in the Appendix where we have thoroughly explained this phenomena.
> > >
> > > Best,
> > > Authors of Submission4948

---

### Official Review · Reviewer_9PSE · 2024-11-03

**Soundness:** 3
**Presentation:** 3
**Contribution:** 2
**Rating:** 6
**Confidence:** 4

**Summary:**

In this paper, the authors argue that the current LVLMs and hallucination mitigation decoding strategies lack visual perception capabilities. Through extensive analyses, the authors delve into such deficiency across cognitive and real-world benchmarks and categorize more detailed hallucination taxonomies beyond mere object hallucination: Language Hallucination / Vision Hallucinations / Style Hallucinations / IT Hallucinations. At the end, the authors introduce a new decoding strategy, VDGD, which prefixes the model's detailed descriptive response to the given input and truncates grounding of its intermediate responses to refine the final answers.

**Strengths:**

Through section 3 and 4, in this paper, the authors extensively explored hallucination issues across various benchmarks, models, and decoding strategies. The novel taxonomies beyond simple object hallucination are crucial to understand the current problems in hallucination research areas (particularly LVLM).

**Weaknesses:**

Even with the new hallucination categories, and new findings, their approach, VDGD, lacks of analyzing its effectiveness on the new hallucination categories they defined and limited for its computational costs due to successive response generation.

**Questions:**

1. How can hallucinatory results be mitigated using the proposed VDGD in the newly defined hallucination categories, compared to other decoding strategies? Analyses through section 3 and 4 are really intriguing, but the reviewer belives that there is significant gap to bridge such motivation and findings into the VDGD design.


2. The method of VDGD is limited to merely prefixing self-generated model response and relying on the first generated response that the model predicts (ironically this may include a lot of hallucinatory responses-even if limitation section mentioned this). Considering LLMs are more prone to hallucination snowballing rather than error correction, it is unclear where the performance gains are derived from. Unlike original contrastive decoding, VDGD cannot make logit corrections by counteracting with premature models and relies solely on vocabulary truncation.


3. Computational analyses should be conducted such as throughput or latency. Also, can this VDGD be seamlessly applied to beam search decoding? Then, how will be the result comparison?

---

> ### Author Response · Authors · 2024-11-21
> **Response to Official Review of Submission4948 by Reviewer 9PSE (1/2)**
>
> Dear Reviewer 9PSE,
>
> We thank you for your thorough review and constructive feedback. We have tried to address each of your concerns point by point in the rebuttal.
>
> ### Questions
>
> > Question 1 (About VDGD results on newly defined categories)
>
> **Ans.** Thank you for your question. We would like to clarify a potential misunderstanding here. The newly defined hallucination categories pertain to image descriptions and not cognitive QAs. These methods cannot be directly applied to categorize hallucinations in cognitive QAs – for example, instruction-response pairs in typical visual instruction tuning datasets do not contain cognitive prompts. Additionally, as shown in Fig. 6, unlike in image description tasks which have a mix of various hallucinations presented in Section 4, hallucinations in cognitive prompts are dominated by language hallucinations are predominant in. Thus, hallucinations for cognitive prompts cannot be effectively categorized into our mentioned types which is why we do not show results on the same.
>
> **But why is this analysis required here?** To clarify, our analysis and categorization aim to investigate what is needed to mitigate hallucinations in cognitive prompts, thereby improving reasoning. Section 3 highlights that current mitigation techniques fail to address hallucinations in cognitive prompts. Section 4 categorizes hallucinations and suggests that existing methods, such as logit correction, are inherently tailored to mitigate language hallucinations. Finally, Section 5 demonstrates that (i) for image descriptions, hallucinated tokens are high-confidence single tokens in the logit space and can often be corrected by language-only models with the same prefix; (ii) in cognitive prompts, hallucinations arise from a top-$k$ probability space dominated by low-confidence tokens, making current methods ineffective (lines 70-75).
>
> Building on these findings, VDGD addresses this challenge by grounding response generation to image descriptions, boosting the confidence of correct tokens and mitigating hallucinations more effectively.
>
> > Question 2 (About source of gains for VDGD and if VDGD goes through snowballing effect)
>
> **Ans.** Thank you for the question. To address the first part: we agree that hallucinations in image descriptions can lead to hallucinated responses in cognitive questions. Lines 430-431 state that for all results in Table 1, VDGD uses image descriptions generated via the VCD method. VCD mitigates language hallucinations in image descriptions by correcting low-confidence tokens, which are often hallucinated. As shown in Table 3 (Appendix), using VDGD with vanilla decoding instead of VCD for image descriptions results in a performance drop. This highlights the importance of combining VDGD with effective language hallucination mitigation methods like VCD.
> VDGD is complementary to existing mitigation techniques. While methods like VCD effectively address language hallucinations in image descriptions, VDGD leverages these corrected descriptions to improve reasoning by grounding the generation process. It is important to note that VDGD is designed with a distinct motivation. Unlike methods that rely on logit corrections to counter low-confidence tokens, VDGD explores other aspects of the logit space, focusing on grounding through image descriptions. This approach allows VDGD to address reasoning tasks effectively, as demonstrated in our results.
>
> Additionally, Figure 1 illustrates that while VCD addresses language hallucinations, it does not mitigate other hallucination types. VDGD complements such techniques by ensuring that corrected descriptions are used for reasoning, leading to overall performance improvements.
>
> > Question 3 (About computational analysis and applicability to beam search decoding)
>
> **Ans.** Thank you for your suggestion. We have added below a comparison for latency and compute efficiency. We have also added this in the revised version of our paper in Appendix M.
>
> Computational Analysis on LLaVA 1.5 with 48GB GPU (on MMMU):
>
> | Method | Average Inference Time (s)  | FLOPs (T) | Score |
> |---|---|---|--|
> | Vanilla-greedy | 1.3 | 9.3 | 1.35 |
> | VCD  | 1.9  | 10.2 | 1.52 |
> | VDGD + Small Captioning Model [1] | 1.8 | 10.4 | 1.58 |
> | VDGD| 2.3 | 11.9  | 1.62 |
>
> We also show that VDGD is competitive with a small captioning model (which is not the LVLM) and the key is to generate an image caption that captures the visual elements. We hope you find this new results insightful. This shows that efficient captioning models can effectively reduce VDGD complexity.

---

> > ### Author Response · Authors · 2024-11-21
> > **Response to Official Review of Submission4948 by Reviewer 9PSE (2/2)**
> >
> > **continued**
> >
> > Thank you for the question about beam search decoding. Yes, VDGD can be seamlessly applied to beam search decoding without modifying its core methodology. Before a set of tokens is selected for each beam, VDGD reweighs the logit space of the current logit based on the KL-Divergence between the current logit and the image description logits. This process is repeated at every decoding step. Another important point to note is that VDGD operates independently of prior response tokens, relying solely on the fixed image description tokens, which remain constant even during beam search. As a result, VDGD can be directly integrated into beam search decoding to achieve similar improvements in reducing hallucinations.
> >
> > On your request, below we show results of VDGD applied on beam search for LLaVA-v1 and LLaVA1.5 on MMMU:
> >
> > | Benchmark          | LLaVA-v1 | LLaVA-1.5 |
> > |--------------------|----------|-----------|
> > | Vanilla-greedy     | 1.26     | 1.35      |
> > | Vanilla-sampling   | 1.27     | 1.44      |
> > | VDD                | 1.34     | 1.52      |
> > | OPERA              | 1.30     | 1.43      |
> > | Woodpecker         | 1.32     | 1.44      |
> > | LRV                | 1.29     | 1.49      |
> > | LURE               | 1.31     | 1.47      |
> > | PAI                | 1.42     | 1.39      |
> > | HALC               | 1.40     | 1.54      |
> > | VDGD               | 1.42     | 1.62      |
> > | VDGD + Beam Search | 1.39     | 1.58      |
> >
> > We did not include them in the initial version as we did not find enough papers in LVLM literature that employ beam search decoding and much of literature in hallucination mitigation also only show results with greedy and sampling based decoding. We have added this analysis to the Appendix N of the revised version of our paper.
> >
> > ### References
> >
> > [1] https://arxiv.org/abs/2209.15323

---

> ### Author Response · Authors · 2024-11-23
> **Request to review the rebuttal**
>
> Dear reviewer dSa3,
>
> Thank you for taking the time to review our paper. We have addressed your concerns in our submitted response and provided a revised version of the paper. As the rebuttal period is nearing its conclusion, we kindly request you to review our rebuttal and share any additional comments or concerns you may have. Thank you once again for your valuable feedback!
>
> Best,
> Authors of Submission4948

---

> > ### Author Response · Authors · 2024-11-24
> > **Request to review the rebuttal**
> >
> > Dear reviewer dSa3,
> >
> > Thank you for taking the time to review our paper. We have addressed your concerns in our submitted response and provided a revised version of the paper. As the rebuttal period is nearing its conclusion, we kindly request you to review our rebuttal and share any additional comments or concerns you may have. Thank you once again for your valuable feedback!
> >
> > Best,
> > Authors of Submission4948

---

> > > ### Author Response · Authors · 2024-11-25
> > > **Request to review the rebuttal**
> > >
> > > Dear reviewer dSa3,
> > >
> > > Thank you for taking the time to review our paper. We have addressed your concerns in our submitted response and provided a revised version of the paper. As the rebuttal period is nearing its conclusion, we kindly request you to review our rebuttal and share any additional comments or concerns you may have. Thank you once again for your valuable feedback!
> > >
> > > Best,
> > > Authors of Submission4948

---

> > > > ### Comment · Reviewer_9PSE · 2024-11-25
> > > >
> > > > Thank you for the detailed rebuttal.
> > > > The reviewer carefully reads other reviewes and discussion and keeps the current score.
> > > >
> > > > Several concerns to address as future directions:
> > > >
> > > > 1. In addition to my initial question, the reviewer wonders if the current VDGD can indeed mitigate sub-categorized hallucinations in AMBER and MMLU, as shown in Fig. 5. Including the VDGD results in those figures would more clearly show the method's effectiveness. (but the reviewer also understands the lack of hallucination-specialized benchmarks.)
> > > >
> > > > 2. Regarding the computational analysis, do the authors include the total time for generating responses from VCD, or report only the inference time for the decoding phase of VDGD (as the authors mentioned, VDGD relies on the responses generated by VCD)?
> > > >
> > > > -- additional --
> > > >
> > > > To be honest, (even with updated Table 3 in appendix), the reviewer cannot agree of the advantage for the possible conjunction decoding strategy with VDGD when using outputs from other captioners. The reviewer believes that hallucination issues should be addressed through self-correction (if we use GPT-4V captioner, why not just use GPT-4V for the final answer?).

---

> > > > > ### Author Response · Authors · 2024-11-25
> > > > > **Response to Official Comment by Reviewer 9PSE**
> > > > >
> > > > > Thank You for your time in reviewing our response. We respect your scoring decision. We would like to highlight and respond to your questions.
> > > > >
> > > > > > if we use GPT-4V captioner, why not just use GPT-4V for the final answer?
> > > > >
> > > > > **Ans.** Thank You for your question! This particular experiment is not a part of our main experiments or analysis and it was just to show that *better captions can improve VDGD response quality and further reduce hallucinations.*, as mentioned in lines 509 - 511 in the main paper and lines 816 - 822 in the Appendix. This experiment was carried out on request of Reviewer fYgL who asked us to compare how varying qualities of captions can affect VDGD response. This is also the reason why we did not employ GPT-4V captions in VDGD results in Table 2 of our main paper.
> > > > >
> > > > > We acknowledge that employing GPT-4 itself can perform very well and the results for the same are also provided in Table 7,11,12, and 13 in the Appendix.
> > > > >
> > > > > > the reviewer cannot agree of the advantage for the possible conjunction decoding strategy with VDGD when using outputs from other captioners
> > > > >
> > > > > **Ans.** Thank You for your question! We did not want to show any advantage of the possible conjunction of VDGD with other captioners. This experiment was carried out on request of Reviewer fYgL who asked us to compare how varying qualities of captions can affect VDGD response. *Additionally, we show that a small captioning model can perform competitively and can reduce VDGD computational overhead for the first caption generation stage (instead of using the large LVLM with VCD itself).*
> > > > >
> > > > > > Regarding the computational analysis, do the authors include the total time for generating responses from VCD, or report only the inference time for the decoding phase of VDGD (as the authors mentioned, VDGD relies on the responses generated by VCD)?
> > > > >
> > > > > **Ans.** Thank You for the question! This includes the total time for generating responses from VCD. We would like to highlight that a majority of the processing time is involved in processing the image tokens (576 in number). In comparison, the captions are ideally 30-50 tokens and add minimal time overhead, both for VCD output generation and VDGD input processing.
> > > > >
> > > > > > In addition to my initial question, the reviewer wonders if the current VDGD can indeed mitigate sub-categorized hallucinations in AMBER and MMLU, as shown in Fig. 5. Including the VDGD results in those figures would more clearly show the method's effectiveness. (but the reviewer also understands the lack of hallucination-specialized benchmarks.)
> > > > >
> > > > > **Ans.** Thank You for the question!  Comparing VDGD on Amber is unfair as VDGD already relies on image captions, which is the core task of AMBER. About MMMU and other cognitive benchmarks, we would like to iterate from our Rebuttal response to Question 1:  *The newly defined hallucination categories are meant for to image descriptions by LVLMs and not response to cognitive QAs. These methods cannot be directly applied to categorize hallucinations in cognitive QAs.* For our response to Question 1 in the rebuttal, we also provide examples and the reason why we include that in our main paper.
> > > > >
> > > > > We respect your decision for scoring. We just wanted to highlight and respond to the you clarifying any potential misunderstandings. Please let us know if you have further queries we can address!
> > > > >
> > > > > Best,
> > > > > Authors of Submission4948

---

### Meta-Review · Area_Chair_docw · 2024-12-25

**Metareview:**

The submission addresses the problem of hallucination in large vision-language models. It identifies that a key contributor to hallucination is the difficulty of associating visual concepts with the internal knowledge of large language models. The authors introduce VDGD, a training-free approach to mitigate hallucination by adding image description to the text instruction for the inputs of VLMs. They also introduce a new benchmark to measure "cognitive reasoning" capabilities of such models. After rebuttal, the submission received three borderline accepts (6) and one accept (8). The AC agrees with the consensus reached by the reviewers, and recommends acceptance of the submission to ICLR 2025. The AC especially appreciates the detailed analysis on the sources of hallucination in VLMs, and encourages the authors to integrate all valuable reviewer feedback in the final version.

**Additional Comments On Reviewer Discussion:**

After rebuttal discussion, concerns from 4B1M and cozw have been addressed. There were remaining concerns that:
- (fYgL) The method depends on the image captioning quality. This should be clarified in the final draft.
- Additionally, the AC believes that most of the questions raised by 9PSE in their last message were adequately addressed by the authors.

---

### Decision · Program_Chairs · 2025-01-22

Accept (Poster)